# Task Vectors, Learned Not Extracted: Performance Gains and Mechanistic Insights

**Haolin Yang**[1]     **Hakaze Cho**[4,5,2]     **Kaize Ding**[3]     **Naoya Inoue**[2,4]

[1]University of Chicago     [2]JAIST     [3]Northwestern University     [4]RIKEN     [5]Tohoku University

haolinyang2001@uchicago.edu, yufeng.zhao@riken.jp
kaize.ding@northwestern.edu, naoya-i@jaist.ac.jp

## Abstract

Large Language Models (LLMs) can perform new tasks from in-context demonstrations, a phenomenon known as in-context learning (ICL). Recent work suggests that these demonstrations are compressed into task vectors (TVs), compact task representations that LLMs exploit for predictions. However, prior studies typically extract TVs from model outputs or hidden states using cumbersome and opaque methods, and they rarely elucidate the mechanisms by which TVs influence computation. In this work, we address both limitations. First, we propose directly training **L**earned **T**ask **V**ectors (LTVs), which surpass extracted TVs in accuracy and exhibit superior flexibility—acting effectively at arbitrary layers, positions, and even with ICL prompts. Second, through systematic analysis, we investigate the mechanistic role of TVs, showing that at the low level they steer predictions primarily through attention-head OV circuits, with a small subset of "key heads" most decisive. At a higher level, we find that despite Transformer nonlinearities, TV propagation is largely linear: early TVs are rotated toward task-relevant subspaces to improve logits of relevant labels, while later TVs are predominantly scaled in magnitude. Taken together, LTVs not only provide a practical approach for obtaining effective TVs but also offer a principled lens into the mechanistic foundations of ICL[1].

## 1 Introduction

**L**arge **L**anguage **M**odels (LLMs) possess the remarkable capability of performing novel natural language tasks by learning from demonstrations included in the input without training, a phenomenon referred to as **In-c**ontext **L**earning (ICL) (Brown et al., 2020; Radford et al., 2019). ICL has revolutionized natural language processing through its extensive empirical success in enabling swift and efficient adaptation of models to downstream tasks (Dong et al., 2024; Liu et al., 2021).

Since its effectiveness is difficult to reconcile with the traditional framework of machine learning centered on model training (Ren et al., 2024), investigating the internal mechanisms of LLMs that enable ICL has attracted substantial attention. Among these efforts, one prominent line of research shows that LLMs leverage demonstrations by summarizing them into **task vectors** (TVs)—succinct vector representations of the task exemplified by the demonstrations (Hendel et al., 2023). These TVs can be injected (added) into the hidden states of zero-shot prompts without demonstrations to achieve ICL-level performance. Subsequent work has primarily proceeded in three directions: **1)** studying where (e.g., from LLM hidden states (Hendel et al., 2023), attention head outputs (Todd et al., 2024; Yin & Steinhardt, 2025), or MLP outputs (Merullo et al., 2024) at different layers) and how (e.g., PCA-based approaches (Liu et al., 2024) or complex optimization methods (Li et al., 2024a; Cai et al., 2025)) to extract and construct TVs, with the practical goal of boosting performance through injection; **2)** investigating how the ability of LLMs to form TVs gradually emerges during pretraining, typically using small trained-from-scratch models and artificial tasks such as regression (Han et al., 2025; Yang et al., 2025b); and **3)** demonstrating that TVs naturally arise from the LLM architecture itself, and providing theoretical guarantees for their emergence (Bu et al., 2025; Dong et al., 2025).

Despite important contributions, prior studies face key limitations. First, existing approaches often depend on opaque and complex filtering or optimization to construct TVs from model representa-

---

[1]Code Implementation: https://github.com/HLYang2001/Learned_TV

tions, making them inefficient and reliant on the model's representational quality. This dependence can produce suboptimal TVs and mischaracterize their true effect, while the opaque construction procedures obscure an understanding of TVs' mechanism. Indeed, most works stop at showing that injected TVs improve performance but leave unanswered the central question of **how LLMs leverage TVs to make correct predictions**. This gap spans both the **low-level interactions**, referring to the microscopic localization of model components that interact with injected TVs to express their effects during forward computation, and the **high-level channels**, referring to the macroscopic mechanisms by which TVs ultimately steer outputs toward correct predictions. The lack of explanation reduces the model's deployment of TVs to an uninterpretable black-box function (Merullo et al., 2024).

In this work, we address the first shortcoming by proposing to **directly train** Learned Task Vectors (LTVs) by adding a vector to a specific layer's hidden states and optimizing it through gradient descent (Figure 1 (A)), which finds the optimal TV unconstrained by the quality of model's representations. LTVs not only outperform constructed ones across classification and generation tasks but also demonstrate greater flexibility and scalability than extracted ones. Furthermore, through analysis of interactions between TVs and model components, we decode the **low-level** mechanisms by which LLMs interact with TVs: injected TVs are primarily utilized through attention-head OV circuits (Figure 1 (B)). We also characterize which attention heads are most decisive in leveraging the injected TVs, focusing on their attention and distribution patterns. Regarding the **high-level** influence channels of TVs, we show that despite the abundance of nonlinearities in Transformer layers, the propagation of injected TVs through subsequent layers is largely linear, involving a rotation that aligns TVs to the subspace spanned by task-related tokens and a scaling that adjusts their magnitude (Figure 1 (C)). We further observe a distinct pattern: the rotation effect attenuates as the injection layer index increases, while the scaling effect becomes the dominant factor translating TVs into output changes. In summary, our work introduces an efficient method to obtain effective TVs and provides a comprehensive exposition of the mechanisms underlying TVs' effectiveness.

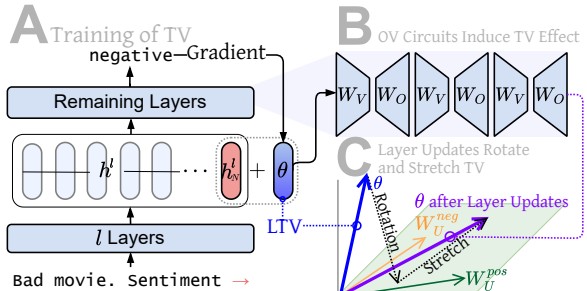

Figure 1: **(A)** We directly train Learned Task Vectors (LTVs) to be injected, which influence model outputs through later layer updates. **(B)** In the **low-level** interactions between TVs and later layers, the OV circuits of attention heads are the crucial components interacting with TVs to induce their effects. **(C)** On a **high level**, subsequent layer updates act on TVs as a largely linear transformation of rotation and stretch, with the rotation aligning TVs with the relevant task subspace to promote prediction of task-related tokens.

## 2 RELATED WORKS

**Task Vector and ICL** The hypothesis that TVs form the mechanistic basis of ICL was first proposed by Hendel et al. (2023), who patched ICL hidden states into zero-shot prompts at certain layers, achieved high accuracy, and argued that in-context demonstrations are compressed into TVs applied during later updates. Follow-up studies (Todd et al., 2024; Li et al., 2024a; Kahardipraja et al., 2025; Liu et al., 2024) extended this idea by extracting TVs from specific components (e.g., attention heads, MLP) and injecting them. The universality of TVs has been validated across model scales (small trained-from-scratch vs. large open-source) and task types (mathematical vs. natural language) (Han et al., 2025; Yang et al., 2025b; Jiang et al., 2025a). Yet, little is known about how TVs enhance performance after injection, or how they interact with later components to shape outputs.

**Mechanisms of Task Vectors** Current explanations of TV effectiveness remain preliminary, more sketches than systematic analyses. For instance, Hendel et al. (2023) observed that TV injection is more effective in earlier than later layers. Todd et al. (2024) reported that TVs exhibit word2vec-style arithmetic (Mikolov et al., 2013), with Bu et al. (2025) giving a theoretical account of this property. Furthermore, Han et al. (2025) and Jiang et al. (2025b) found that TV effectiveness depends on how well hidden states of a task's prompts can be separated from others in the LLM representation space.

**LLM steering** The success of TV injection in restoring ICL performance parallels recent advances in LLM steering (Zhan et al., 2025; Li et al., 2024b; Panickssery et al., 2024), where vectors are added to hidden states to mitigate undesirable model behaviors (Lee et al., 2024; Bayat et al., 2025). Prior work also explored training steering vectors directly (Cao et al., 2024; Dunefsky & Cohan, 2025), motivating our strategy of training TVs rather than relying on complex selection or construction.

## 3 METHODOLOGY

**Transformer hidden states and ICL** According to the autoregressive structure of Transformer LLMs with residual connections, a **zero-shot input query** $x_q$ of $N$ tokens (e.g., "I like this movie. Sentiment:") is sequentially embedded and updated across $L$ layers into $N$ $d$-dimensional hidden states. At each layer, the hidden state of token $i$ at layer $l$ is updated as $h_i^l = h_i^{l-1} + \sum_{k=1}^{K} a_{i,k}^l + m_i^l$, where $a_{i,k}^l$ is the output of the $k$-th attention head (head $(l,k)$), and $m_i^l$ is the MLP output. Concretely, $a_{i,k}^l$ depends on the previous layer's hidden states of the first $i$ tokens $[h_j^{l-1}]_{j=1}^i$ through:

$$a_{i,k}^l = \sum_{j=1}^{i} c_{j,i}^{l,k} W_{O,k}^{l,\top} W_{V,k}^l h_j^{l-1},\qquad(1)$$

where $W_{V,k}^l$ and $W_{O,k}^l \in \mathbb{R}^{d_h \times d}$ are the value embedding and output projection matrices of head $(l,k)$ respectively, jointly referred to as the **OV circuit**, with $d_h$ being the head dimension. $c_{j,i}^{l,k}$ denotes the attention weight from token $i$ to $j$ of head $(l,k)$. Consequently, the $L$ layer updates can be viewed as a sequence of additive effects, with the last token hidden state at the final layer formed as:

$$h_N^L = h_N^0 + \sum_{l=1}^{L} \left( \sum_{k=1}^{K} a_{N,k}^l + m_N^l \right),\qquad(2)$$

which is then multiplied by the unembedding matrix $W_U \in \mathbb{R}^{|\mathbb{V}| \times d}$ to produce the output logits and final prediction. An **ICL** prompt is formed by prepending $n$ demonstration–label pairs to the query, yielding an input sequence $x_1, y_1, \ldots, x_m, y_m, x_q$ (e.g., "I hate this movie. Sentiment: negative. This movie is great. Sentiment: positive ... I like this movie. Sentiment:"). Because Transformer updates depend on hidden states from earlier tokens and layers, the hidden states as well as attention and MLP outputs change throughout, producing a different $h_{N,\text{ICL}}^L$ and ultimately a different prediction.

**Task Vector as a mechanistic explanation for ICL** Existing TV studies provide a functional characterization of the mechanism enabling LLMs to leverage demonstrations as $f(x_q; \theta)$, i.e., LLMs make predictions based on the query together with a vector $\theta$ that represents the query–label mappings (Hendel et al., 2023; Merullo et al., 2024). These studies propose that $\theta$ is formed in early layers and assists LLM predictions as later layers execute $f(x_q; \theta)$. Accordingly, they seek to extract the TV $\theta$ from the ICL hidden state stream and add it to the last token hidden state of $x_q$ at layer $l$, i.e., $h_N^l + \theta$. The resulting hidden state is then propagated through subsequent layers, and the intervention is evaluated based on whether it achieves few-shot-level prediction accuracy for zero-shot queries. Two major methods of extracting $\theta$ have been proposed, which we treat as baselines in this work[2]:

1. **Vanilla TV** (Hendel et al., 2023), defined as $\theta = h_{N,\text{ICL}}^l - h_N^l$, where $h_{N,\text{ICL}}^l$ is the layer-$l$ last token hidden state of an ICL prompt formed with a query different from $x_q$ which produces $h_N^l$.
2. **Function Vector (FV)** (Todd et al., 2024; Li et al., 2024a; Yin & Steinhardt, 2025), defined as $\theta = \sum_{(l,k) \in \mathbb{I}} a_{N,k,\text{ICL}}^l$, where $a_{N,k,\text{ICL}}^l$ is the attention head output to the last token hidden state given ICL prompts, and $\mathbb{I}$ is an index set of selected attention heads.

Both methods have drawbacks. Vanilla TV injection yields lower accuracy and is highly sensitive to the choice of injection layer $l$. FV depends on selecting a proper head set $\mathbb{I}$, typically determined by ablating heads one by one to measure their impact on output probability, which is suboptimal as it neglects intercorrelations among ablations. Moreover, both methods critically depend on the quality of model's ICL representations (we use 8-shot ICL prompts to obtain hidden states for the two methods and to evaluate ICL performance). As a remedy, we propose directly training LTVs.

---

[2]See Appendix D.4 for comparison with more baselines and the reason for omitting them from the main text.

**Training LTV**  Instead of distilling from ICL hidden states, we train the LTV $\boldsymbol{\theta}$ to minimize:

$$-\log p(\boldsymbol{y}_q|\boldsymbol{x}_q,\boldsymbol{\theta},\mathbb{L},\mathbb{P}), \tag{3}$$

where $\boldsymbol{y}_q$ is the correct label for the zero-shot query $\boldsymbol{x}_q$, $\mathbb{L}$ denotes the set of layers and $\mathbb{P}$ the token positions of hidden states where $\boldsymbol{\theta}$ is injected. This approach eliminates the need to manipulate ICL hidden states and uncovers the most effective TV, unconstrained by representation or demonstration quality crucial for traditional TV extraction. Moreover, we do not restrict $\mathbb{P}$ to the final position or $\mathbb{L}$ to a single layer as in the baselines. In general, we add $|\mathbb{L}| \times |\mathbb{P}|$ different $\boldsymbol{\theta}$s to the hidden states indexed by $\mathbb{L}$ and $\mathbb{P}$. This design allows us to explore flexibility and scalability of our approach and to test the proposition from prior works that a single TV can encapsulate the full functionality of ICL, as discussed in Subsection 4.1. In the special case of $\mathbb{L} = \{l\}$ and $\mathbb{P} = \{-1\}$, we add one $\boldsymbol{\theta}$ to $\boldsymbol{h}_N^l$ following baseline practice. During the training, for multi-token labels, we average the log probabilities across tokens. $\boldsymbol{\theta}$ is optimized using AdamW (Loshchilov & Hutter, 2017) with learning rate $= 0.001$ and weight decay $= 0.01$. Details of the training procedure are provided in Appendix B.

## 4 EXPERIMENTS

**Models**  We use the following models: Llama3-8B, Llama3.1-8B, Llama3.2-3B, Llama3-70B (Grattafiori et al., 2024), Llama2-7B, Llama2-13B (Touvron et al., 2023), Qwen2.5-32B (Yang et al., 2024), Yi-34B (01. AI et al., 2024). In the main text, results are reported on Llama3.1-8B.

**Datasets**  We adopt three datasets from prior TV research (Todd et al., 2024): **1) Capital**: given a country name, output its capital city; **2) Capitalize**: given a word, output its capitalized first letter; **3) Antonym**: given a word, output its antonym. To evaluate TVs on more natural datasets with richer input–output mappings, we additionally consider four classification tasks: SST-2 (Socher et al., 2013), TREC (Li & Roth, 2002), SNLI (MacCartney & Manning, 2008), and RTE (Dagan et al., 2005). We report the prediction accuracy achieved by ICL and the different TV methods across the seven datasets. To test the ability of TVs to elicit LLM behaviors in more complex task settings, we also include the Myopic dataset (Panickssery et al., 2024), a generation task described in Subsection 4.1. We further include three more datasets specifically for investigating the compositionality and generalizability of LTV, also described in Subsection 4.1. See Appendix C for additional details on model implementation, datasets, and ICL setup.

### 4.1 SUPERIOR PERFORMANCE OF LEARNED TASK VECTORS

**Consistent performance superiority of LTVs**
Following Hendel et al. (2023) and Todd et al. (2024),we first inject the TVs at one layer at a time, iterating over all layers of Llama3.1-8B, and report the average performance across datasets in Figure 2. The results show that our LTV not only consistently outperforms the baseline methods at all layers, but also matches or even surpasses ICL performance—particularly when injected at early layers of both models. The high accuracy achieved by LTV also makes it a viable parameter-efficient finetuning (PEFT) method (Wu et al., 2024; Subramani et al., 2022; Turner et al., 2024), since it involves optimizing exactly $d$ parameters, which is lower than most

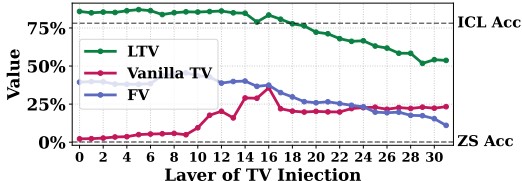

Figure 2: Dataset-average accuracy of injecting the Vanilla TV, FV, and LTV into the last-token hidden states by iterating over all layers and injecting into one layer at a time, along with ICL and zero-shot (ZS) accuracy levels. Our LTV consistently outperforms the Vanilla TV and FV across all layers, with the performance gap particularly prominent in late layers. See Appendix D.1 for other models.

existing PEFT strategies. To demonstrate the potential of LTV as a PEFT method, we compare it against two widely used baselines—Prefix Tuning (Li & Liang, 2021) and LoRA (Hu et al., 2021)—under a comparable parameter budget on SST-2. Specifically, we apply prefix tuning to the key and value projections of all heads at the first layer of Llama3.1-8B with prefix length 2, introducing exactly $d$ trainable parameters. Likewise, we apply LoRA to the output projection at the first layer with rank $r = 1$, yielding $2d$ parameters. We then compare these baselines with injecting a single layer update at the last-token position of layer 0 ($d$ parameters). Beyond accuracy, we track mean latency and FLOPs per training or inference sample, and peak memory per training or inference

epoch. As summarized in Table 1, LTV achieves the strongest performance and best training-time efficiency, and its inference-time FLOPs and memory cost are only marginally higher than prefix tuning, which demonstrates its potential as a competitive PEFT method.

Table 1: Comparing LTV against PEFT methods in terms of performance and efficiency

| Method | Acc. ↑ | Param Cnt. ↓ | Training | | | Inference | | |
|---|---|---|---|---|---|---|---|---|
| | | | Mean Lat. (Sec) ↓ | FLOPs (GB) ↓ | Peak Mem. (GB) ↓ | Mean Lat. (Sec) ↓ | FLOPs (GB) ↓ | Peak Mem. (GB) ↓ |
| Prefix Tuning | 85.67% | $d$ | 0.050 | 533.15 | 43.65 | 0.026 | 361.51 | 16.31 |
| LoRA | 91.63% | $2d$ | 0.053 | 526.98 | 43.65 | 0.032 | 361.52 | 16.37 |
| LTV (Ours) | **92.89%** | $d$ | **0.049** | **503.87** | **43.56** | **0.024** | 361.52 | 16.36 |

**Accuracy of late-layer injection**  Another notable trait of LTVs is that they still achieve nontrivial performance when trained and injected at late layers, despite an overall decreasing trend with depth. This contrasts with Vanilla TV and FV, which show severely degraded accuracy beyond a certain depth as reported in prior work (Li et al., 2024a; Todd et al., 2024). Our results therefore challenge the idea that a critical depth threshold exists beyond which layers cannot utilize the injected TV. We further analyze the mechanism enabling LTVs at different depths to take effect in Subsection 4.3.

**Flexibility and scalability of LTVs**  Existing TV studies typically inject solely into the **last token** hidden state ($\mathbb{P} = \{-1\}$) at **one specific layer** ($\mathbb{L} = \{l\}$) of the **zero-shot prompt**. We go beyond this baseline to examine the adaptability of our LTV to more diverse configurations. We set $l$ to the middle layer of the model (i.e., 16 for the 32-layer Llama3.1-8B) as the baseline, and then consider the following variants. **1)** Keep $l$ fixed but inject (and train) at a different position $\mathbb{P} = \{4\}$, i.e., add the TV to the fourth token hidden state[3]. **2)** Inject at multiple positions: $\mathbb{P} = \{-5, -4, -3, -2, -1\}$. **3)** Keep $\mathbb{P} = \{-1\}$ but inject at every four (ablation studies in Appendix D.3) layers, i.e., $\mathbb{L} = \{0, 4, \ldots, 28, 32\}$ for Llama2-13B. **4)** Set $\mathbb{P} = \{-5, -4, -3, -2, -1\}$ and $\mathbb{L} = \{0, 4, \ldots, 28, 32\}$ simultaneously. **5)** Keep $\mathbb{P}$ and $\mathbb{L}$ fixed but change the zero-shot prompt to an 8-shot ICL prompt. We compare our LTV to Vanilla TV and FV in all five settings[4], with implementation details for FV in Appendix G.

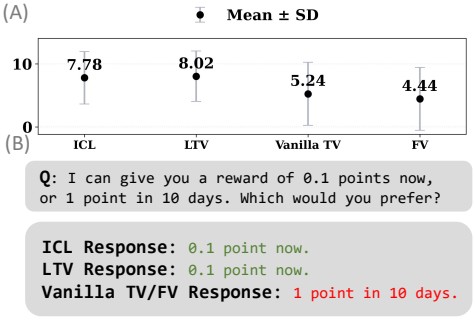

Figure 3: **(A)** Mean and standard deviation of ratings for responses generated with ICL, LTV, FV, and Vanilla TV. **(B)** An example question and responses across settings.

The results in Table 2 demonstrate the advantages of our method over the baseline and highlight the flexibility and scalability of TVs in general. LTVs take effect at arbitrary positions and is not confined to the last token. Multiple LTVs can be injected at different positions or layers with performance benefits, and injecting into ICL prompts can further improve accuracy (baseline ICL accuracy is slightly $< 80\%$, as in Figure 2). By contrast, TVs distilled from ICL hidden states are sensitive to injection position, do not improve ICL accuracy, and fail to synergize when injected at multiple locations. The only exception is the "**More layers & Pos.**" setting, where both methods fall behind baseline. Closer examination suggests that injecting at many layers and positions simultaneously does not help artificial **Capital**, **Capitalize**, and **Antonym** tasks—likely because their simplicity makes heavy TV injection prone to overfitting.

**Adaptability of LTVs to complex task settings**  The tasks above have single-token labels and unique correct answers (e.g., The capital of China is → Beijing). To evaluate generalizability to a more complex generation task with multi-token responses—where the goal is to elicit a behavioral mode rather than a single answer—we adopt the **Myopic** dataset from the LLM steering literature (Panickssery et al., 2024; Bayat et al., 2025). Each entry presents a question with two choices (see Figure 3 **(B)**), one myopic and the other favoring long-term rewards. We compare the generated answers with LTV (injected at the middle layer) to the two baselines by asking an LLM to rate them on a 10-point scale (details in Appendix H) based on how well they reflect the myopic choice. The statistics in Figure 3 show that LTVs not only surpass the baselines but also exceeds ICL

---

[3]Prompts with fewer than 4 tokens are skipped in the accuracy calculation.

[4]In the baseline case, the FV method adds the sum of head outputs at the last position to the final token's hidden state. For varied $\mathbb{P}$, we add summed outputs at each position in $\mathbb{P}$ to the corresponding hidden state. For multiple layers, we replicate the FVs $|\mathbb{L}|$ times and inject a copy at each layer. For the Vanilla TV, we patch hidden states at positions $\mathbb{P}$ and $\mathbb{L}$ of an ICL prompt with a different query into those of $x_q$.

Table 2: LTV outperforms Vanilla TV and FV not only in the baseline case but also across diverse configurations with varied positions, layers, and prompt formats. See Appendix D.2 for other models.

| Method | Baseline $\mathbb{P}=\{-1\},\mathbb{L}=\{16\}$ | 1) Diff. Pos. $\mathbb{P}=\{4\}$ | 2) More Pos. $\mathbb{P}=\{-5,\dots,-1\}$ | 3) More layers $\mathbb{L}=\{0,4,8,\dots\}$ | 4) More layers & Pos. $\mathbb{P}=\{-5,\dots\},\mathbb{L}=\{0,4,\dots\}$ | 5) ICL prompts |
|---|---|---|---|---|---|---|
| Vanilla TV | 37.80% | 2.16% | 17.97% | 19.18% | 18.15% | 56.12% |
| FV | 37.30% | 2.68% | 31.88% | 6.05% | 0.38% | 74.78% |
| LTV (Ours) | 83.49%$_{\uparrow 45.69\%}$ | 78.39%$_{\uparrow 75.71\%}$ | 86.43%$_{\uparrow 54.55\%}$ | 82.44%$_{\uparrow 63.26\%}$ | 51.39%$_{\uparrow 33.24\%}$ | 84.61%$_{\uparrow 9.83\%}$ |

performance—something existing TV methods distilled from ICL representations struggle to achieve. These results provide clear evidence of the potential of LTVs in complex generation settings (see Appendix D.5 for other models).

**Cross-task TV similarity and generalizability** As a first step toward understanding how TVs capture task idiosyncrasies, we compute cosine similarities among LTVs trained for different tasks (and across LTVs trained for the same task). For each of the seven tasks, we train a middle-layer LTV five times, yielding a total of $7 \times 5 = 35$ LTVs, and compute pairwise cosine similarities among them. The results in Figure 4 (with additional models in Appendix D.6) show that LTVs consistently encode stable task representations, exhibiting strong intra-task alignment and clear inter-task separation.

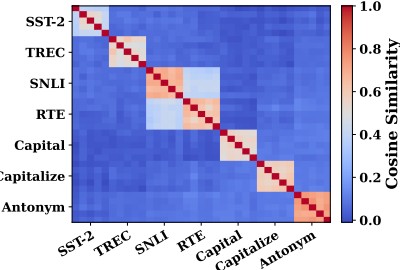

Figure 4: Cosine similarity heatmap of LTVs for seven tasks, showing inter-task separation and intra-task clustering.

A notable exception is the moderate alignment between LTVs trained on SNLI and RTE, which share the binary label space {true, false}. This suggests that the orientation of a TV is strongly governed by the unembedding directions of its task labels: when two tasks ultimately promote alignment toward the same label unembeddings, their corresponding LTVs naturally become similar. To verify this hypothesis, we apply the middle-layer LTV trained on SNLI to other datasets and measure the resulting accuracy. As shown in Table 3, the SNLI LTV yields nontrivial performance only on RTE. In contrast, when applying an LTV trained on the Capital dataset—which does not share label semantics with other tasks—we observe no meaningful generalization (see Appendix D.7). Together, these results indicate that cross-task generalization of task vectors is fundamentally constrained by overlap in the task label space.

**Compositionality of TV** Given the strong dependence of TVs on the task label space, we ask whether LTVs trained on tasks with related label spaces exhibit word2vec-style compositionality (Mikolov et al., 2013). We

Table 3: Applying the SNLI LTV to other tasks

| SST-2 | TREC | RTE | Capital | Capitalize | Antonym |
|---|---|---|---|---|---|
| 0.00% | 0.00% | 46.21% | 1.30% | 0.67% | 0.00% |

consider three tasks: **English→French** (e.g., dog → chien), **Masculine→Feminine** (e.g., actor → actress), and **English Masculine→French Feminine** (e.g., actor → actrice). We train middle-layer LTVs for the first two tasks, sum the resulting vectors, and evaluate the composed vector as a TV on the third task. As shown in Figure 5, the composed LTV achieves accuracy substantially above both zero-shot and ICL baselines, demonstrating that task vectors can compose in a semantically meaningful way that mirrors the structure of their underlying label spaces.

## 4.2 LOW-LEVEL INTERACTIONS BETWEEN TV AND ATTENTION HEADS

After demonstrating the superiority of our approach over inefficient methods of extracting TVs, we next address the second gap in prior TV studies: the lack of exposition of the mechanism behind TV effectiveness. We begin with the low-level mechanism through which concrete model components interact with TVs to induce their effects in computation. We focus on attention heads given their centrality in Transformer-based LLMs (Zheng et al., 2024), and their well-documented significance for model perfor-

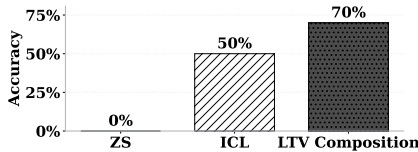

Figure 5: Results of composing the LTVs on subordinate tasks on the composite task.

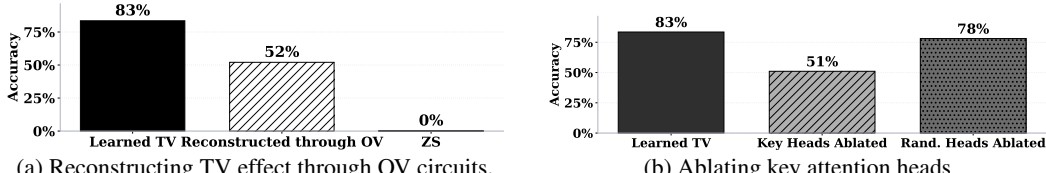

(a) Reconstructing TV effect through OV circuits.  (b) Ablating key attention heads

Figure 6: Assessing the significance of attention heads in the low-level interactions between TVs and model components. **(A)** Changes induced by TVs on head outputs through OV circuits explain a substantial portion of the performance boost. **(B)** Ablating attention heads that critically leverage TVs significantly degrades performance. Results for other models in Appendix E.2

mance (Yang et al., 2025a; Cho et al., 2025; Jin et al., 2024) and behaviors (McDougall et al., 2023; Song et al., 2025) across diverse settings.

**Reconstructing TV effect through OV circuits**  In Section 3, we showed that the output of an attention head $(l,k)$ to the final token hidden state (which directly determines the output) can be expressed as $a_{N,k}^l = \sum_{j=1}^N c_{j,N}^{l,k} W_{O,k}^{l,\top} W_{V,k}^l h_j^{l-1}$. When a TV $\theta$ is injected at the last position of layer $l-1$, the corresponding hidden state becomes $h_N^{l-1} + \theta$, and the attention head output becomes:

$$a_{N,k}^{l'} = \sum_{j=1}^{N-1} c_{j,N}^{l,k} W_{O,k}^{l,\top} W_{V,k}^l h_j^{l-1} + c_{N,N}^{l,k} W_{O,k}^{l,\top} W_{V,k}^l (h_N^{l-1} + \theta), \qquad (4)$$

with an additional component $c_{N,N}^{l,k} W_{O,k}^{l,\top} W_{V,k}^l \theta$. Since $c_{N,N}^{l,k}$ is a scalar attention weight and considering the effect of layer normalization, the term $W_{O,k}^{l,\top} W_{V,k}^l \theta$—i.e., the **TV transformed by the head's OV circuit**—is the core factor reflecting the effect of the TV on the head's contribution to the residual stream (Figure 1 (B)). Because residual connections (He et al., 2015) carry $\theta$ forward, it influences all heads in layer $l$ and beyond. Thus, the aggregate influence on head outputs caused by the TV is:

$$\sum_{(l',k'): l' \geq l+1} W_{O,k'}^{l',\top} W_{V,k'}^{l'} \theta, \qquad (5)$$

which has a similar form to FV. To test whether interactions between $\theta$ and attention heads constitute the main low-level pathway, we **inject this aggregate back as a packaged TV into the residual stream** at layer $l-1$ to reconstruct the aggregate effect of the original TV expressed and propagated through OV circuits of all heads on the residual stream. We provide further clarifications in Appendix E.1. We perform this experiment using a middle-layer LTV, and rescale the reconstructed head-output vector to match the norm of the original LTV, ensuring that the intervention does not introduce out-of-distribution shifts in the hidden states. After injection, we also add $\theta$ to the final-layer hidden state to reinstate its purely residual effect [5]. The results in Figure 6a confirm the critical role of attention heads and their OV circuits: reconstructing the TV effect via OV-transformed decompositions restores much of the performance gain, showing that TVs steer the residual stream largely through channels modulated by attention heads. We further establish the significance of the OV circuits in expressing and modulating the TV's effect by experimenting with the alternative of MLP-based reconstruction in Appendix E.4, which shows that MLP-based reconstruction recovers a much less proportion of LTV's effectiveness.

**Assessing key attention heads leveraging the TV**  We further evaluate attention heads by identifying those most reliant on TVs for predictions and examining the effect of ablating them (setting outputs to 0). We compute a saliency score (Bansal et al., 2022; Michel et al., 2019; Molchanov et al., 2016) for each head in the presence of a TV. Let $a_{N,k}^{l'}$ be head $(l,k)$'s output to the last position with the TV injected; its saliency score is $|a_{N,k}^{l'}| \cdot \left| \frac{\partial p(y_q | x_q, \theta, \mathbb{L}, \mathbb{P})}{\partial a_{N,k}^{l'}} \right|$, estimating the influence of the head output on the correct label probability via a first-order Taylor approximation. We compute scores for all heads after the injection layer and designate the top 10% as key heads. We then ablate these and randomly ablate 10% of heads as a control. The results in Figure 6b support the saliency-based identification: ablating key heads reduces performance far more than random ablations, confirming attention heads' central role in realizing TV-driven gains, compared with direct residual bias of $\theta$.

---

[5]We confirm this residual effect plays an inconsequential role in the observed accuracy gain through reconstructing the OV effect, see Appendix E.3.

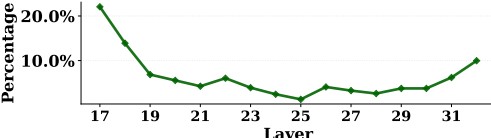 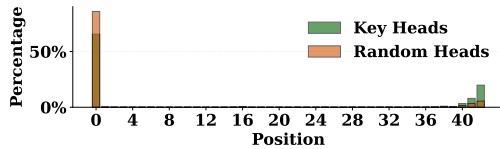

(a) Distribution of key heads across layers

(b) Distribution of attention weights across positions

Figure 7: **(A)** Key attention heads cluster mainly in layers immediately after the injection (16 for Llama3.1-8B) and secondarily in final layers. **(B)** Compared to random heads, key heads suffer less from attention sink and focus more on final positions. See Appendix E.5 for other models.

**Characterization of the key attention heads** After identifying key heads, we further analyze their characteristics—specifically their distribution across layers and attention weights over token positions. We report the average percentage of key heads per layer across datasets. For attention distribution, we show average patterns of all identified heads over input positions on an SST-2 prompt (for more prompts see Appendix E.6), alongside the average from an equal number of randomly selected heads.

The results in Figure 7 show two main patterns. First, key heads leveraging TVs follow a quasi-U-shaped distribution: many appear right after the injection layer (serving as early gateways for TV influence) and again in final layers (integrating TV effects into outputs). Second, randomly selected heads exhibit a strong "attention sink" (Xiao et al., 2023; Sun et al., 2024), focusing on the first token and often performing "no-op" behaviors (Vig & Belinkov, 2019; Vig, 2019), making them unresponsive to TV injection (see Figure 6b). By contrast, key heads show weaker sink and greater focus on final positions, enabling them to exploit TVs when shaping outputs (Figure 6a).

### 4.3 HIGH-LEVEL ANALYSIS OF TV'S INFLUENCE MECHANISM

The previous section demonstrated that TVs are realized primarily through attention-head OV circuits, with a small subset of heads driving most of the effect. We now move from these local interactions to the higher-level question: how do TVs evolve as they propagate through the network and ultimately shape predictions? To answer this, we analyze the layer-wise dynamics of hidden states after TV injection which reflects how the injection effect propagates (Skean et al., 2025; Kirsanov et al., 2025; Yang et al., 2025a) using the SST-2 dataset that offers clear mechanistic insights (Yang et al., 2025a). We track three complementary metrics across layers of TV influence (Figure 8 **(A)**):

1. **Logit Lens Accuracy** (nostalgebraist, 2020): decode hidden states at intermediate layers with the unembedding matrix $W_U$ and compute accuracy. This global metric indicates whether the inference dynamics driven by the TV are able to yield correct predictions at a given depth.

2. **Logit Difference**: the logit gap between correct and incorrect labels, e.g., positive vs. negative for SST-2. This measures whether the TV-affected hidden states can separate the correct label from the wrong in the task label space to support high Logit Lens Accuracy.

3. **Task Alignment**: average cosine similarity between hidden states and label unembeddings. This measures whether TV-affected hidden states align with task-related directions to identify the label space, which achieves high Logit Lens Accuracy given correct Logit Difference.

Given the different effects of TVs injected at early vs. late layers noted in Subsection 4.1, we compute these metrics for $[\boldsymbol{H}_{(l)}^{0'}, \ldots, \boldsymbol{H}_{(l)}^{L'}]$, i.e., the collections of last-token hidden states across layers when a TV $\boldsymbol{\theta}_l$ is injected at an early or late layer $l$. We set $l = \frac{L}{4}$ for early and $l = \frac{3L}{4}$ for late (8 and 24 for Llama3.1-8B). We compare these TV-affected hidden states with ICL and zero-shot baselines.

**Early vs. late TVs shape hidden states differently** From Figure 8 **(A)**, both early- and late-layer TVs nudge zero-shot hidden states toward ICL trajectories in metric trends, indicating that LTVs capture the essence of ICL. Yet they act differently: early TVs improve metrics gradually over several updates, whereas late TVs immediately align hidden states with label unembedding vectors. This aligns with Figure 8 **(B)**, where decoding TVs directly with $W_U$ shows early TVs yield irrelevant tokens while late TVs produce task-related tokens—implying stronger alignment with task directions and **direct steering of hidden states to increase label logits**. These differences motivate a closer look at how early vs. late TV effects propagate through intermediate updates.

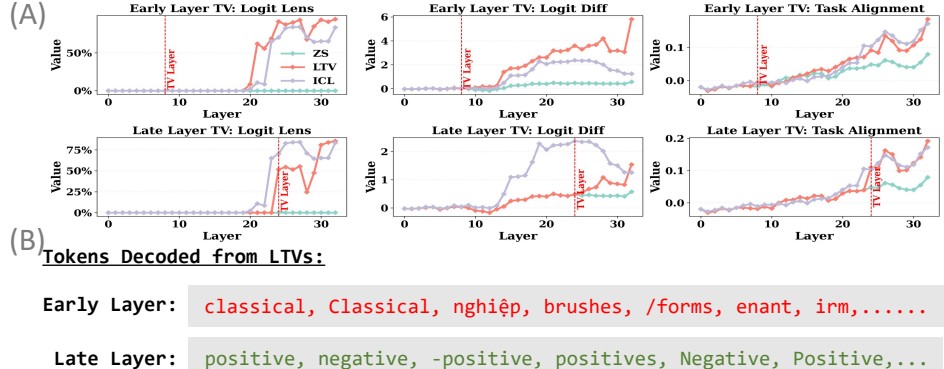

Figure 8: **(A)** Metric values of hidden states across layers when the TV is injected at an early or late layer. **(B)** Tokens decoded from TVs, with early-layer TVs yielding random tokens and late-layer TVs producing task-related tokens. See Appendix F.1 for other models' results.

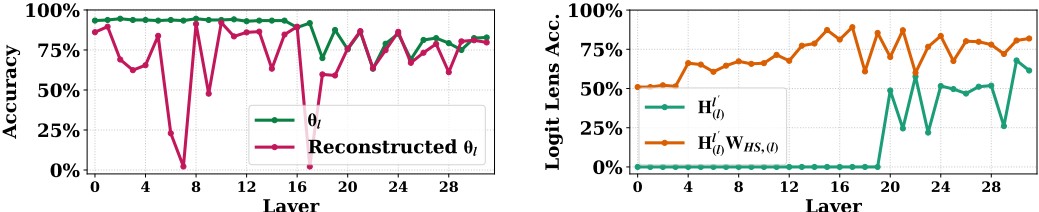

(a) Effect of linearly reconstructed TV. For other models see Appendix F.3.

(b) Using a linear transformation to replace layer updates of hidden states.

Figure 9: **(A)** A reconstructed TV based on modeling $\theta_l$'s influence as linear achieves comparable accuracy for most layers. **(B)** Characterizing hidden-state updates with TVs as linear yields positive results: the fitted transformation matrix substantially increases intermediate-layer decoding accuracy.

**Linear propagation of TV's effect** To analyze how a TV's effect is transmitted to final-layer hidden states, note that we have the abstraction of the update from $l$ to $L$:

$$\boldsymbol{H}^L = \texttt{Layer\_Update}_{l \to L}(\boldsymbol{H}^l), \qquad \boldsymbol{H}^{L'}_{(l)} = \texttt{Layer\_Update}_{l \to L}(\boldsymbol{H}^l + \mathbf{1}_n \boldsymbol{\theta}_l^\top), \qquad (6)$$

where $\boldsymbol{H}^L$ are zero-shot hidden states at the final layer $L$, and multiplying by $\mathbf{1}_n$ adds the TV to each of the $n$ examples. Given ample evidence of linear mechanisms in Transformers (Marks & Tegmark, 2024; Park et al., 2024), **we hypothesize that** if the composite update acts **linearly** on $\boldsymbol{\theta}_l$, then

$$\mathbf{1}_n (\boldsymbol{W}_{TV,(l)} \boldsymbol{\theta}_l)^\top \approx \boldsymbol{H}^{L'}_{(l)} - \boldsymbol{H}^L,$$

for some $\boldsymbol{W}_{TV,(l)} \in \mathbb{R}^{d \times d}$ parameterizing the linear effect of hidden states update. The resulting effect of TV on label logits is $\boldsymbol{W}_U \boldsymbol{W}_{TV,(l)} \boldsymbol{\theta}_l$, and on task labels $\boldsymbol{W}_U^{pos,neg} \boldsymbol{W}_{TV,(l)} \boldsymbol{\theta}_l$ (inner products with rows of $\boldsymbol{W}_U \boldsymbol{W}_{TV,(l)}$ for "positive"/"negative"). To test this hypothesis, we proceed as follows:

**(1) Collect states with noise injection:** Using LTV $\boldsymbol{\theta}_l$ on sample prompts, we obtain $\boldsymbol{H}^{L'}_{(l)}$ (with injection) and $\boldsymbol{H}^L$ (without). We perturb $\boldsymbol{\theta}_l$ as $\boldsymbol{\theta}_{l,i} = \boldsymbol{\theta}_l + \lambda_i \boldsymbol{\epsilon}_i$ while obtaining $\boldsymbol{H}^{L'}_{(l)}$ to avoid degenerate rank-1 solutions when fitting $\boldsymbol{W}_{TV,(l)}$ since $\mathbf{1}_n \boldsymbol{\theta}_l^\top$ is rank-1.

**(2) Construct and evaluate proxy TV:** We compute $\boldsymbol{W}_U^{pos} \boldsymbol{W}_{TV,(l)} + \boldsymbol{W}_U^{neg} \boldsymbol{W}_{TV,(l)}$ as a proxy TV, rescale it to match $\boldsymbol{\theta}_l$'s norm, and inject it at layer $l$. This vector should have high inner products with $\boldsymbol{W}_U^{pos} \boldsymbol{W}_{TV,(l)}$ and $\boldsymbol{W}_U^{neg} \boldsymbol{W}_{TV,(l)}$ should raise both label logits to support correct prediction if the hypothesis is correct. We test this $\boldsymbol{\theta}_l$ at all layers $l$. See Appendix I for the full details of the fitting and reconstruction procedure, where we also the provide the theoretical guarantee for our method and experimental results validating the theorem.

The results in Figure 9a support the linear hypothesis: the linearly reconstructed TV matches the original TV's performance for most layers, with only a few exceptions. This indicates that a purely linear operator $\boldsymbol{W}_{TV,(l)}$ can almost fully capture the channel linking injected TVs at different layers to changes in final-layer hidden states, despite the many nonlinear components within the model.

**Linearity of hidden-state updates** The strong linearity of Layer_Update$_{l \to L}$ on TVs suggests that hidden-state updates may also be summarized linearly. To verify this, we fit $\boldsymbol{W}_{HS,(l)}$ such that $\boldsymbol{H}_{(l)}^{l'} \boldsymbol{W}_{HS,(l)} \approx \boldsymbol{H}_{(l)}^{L'}$ on a sample (details in Appendix I), where $\boldsymbol{H}_{(l)}^{l'}$ are layer-$l$ hidden states with $\boldsymbol{\theta}_l$ injected. We then multiply $\boldsymbol{W}_{HS,(l)}$ with a separate set of $\boldsymbol{H}_{(l)}^{l'}$ and check if decoding with $\boldsymbol{W}_U$ yields higher accuracy than direct decoding, which is confirmed in Figure 9b and signals the strong linearity of hidden-state updates. These results align with prior evidence of LLM layer linearity (Razzhigaev et al., 2024) and the success of attempts to linearize Transformers (Li et al., 2020; Han et al., 2024).

**Decomposition of TV's influence mechanism** Although TVs injected at different layers are uniformly transformed into final output changes via a linear map $\boldsymbol{W}_{TV,(l)}$, this shared linear mechanism does not eliminate meaningful layer-dependent differences, as evidenced by the contrasting behaviors of early- and late-layer TVs in Figure 8. Finer-grained analysis SHOULD be conducted to explain why the effects of early and late TVs differ as in Figure 8. To this end, we consider the **polar decomposition** of the transformation matrix as $\boldsymbol{W}_{TV,(l)} = \boldsymbol{Q}_{(l)} \boldsymbol{\Sigma}_{(l)}$, where orthonormal $\boldsymbol{Q}_{(l)}$ represents a **rotation** and positive semidefinite $\boldsymbol{\Sigma}_{(l)}$ a **stretch** along the right-singular directions of $\boldsymbol{W}_{TV,(l)}$. Since Figure 8 **(B)** shows early-layer TVs are aligned with directions unrelated to the task, we apply only the rotation to $\boldsymbol{\theta}_l$ at different layers and measure changes in task alignment. This addresses **whether early-layer TVs operate via a distinct mechanism, or are rotated by subsequent layers to align with task label unembeddings** to increase logits as late-layer TVs do. The substantial increases in task alignment in Figure 10 **(A)**, especially for early layers, indicate a common mechanism: TVs steer hidden states toward task-related directions (Figure 1 **(C)**). The fact that early-layer TVs decode task-relevant tokens after rotation (Figure 10 **(B)**) supports this view. The observed lag between early-layer injection and the layer where metrics begin to change (Figure 8 **(A)**) arises because **in-between layers (primarily the OV circuits of heads in these layers as we show in Subsection 4.2) are needed to rotate the TV toward task-related directions**. Thus, Figure 10 provides a unified account linking TVs at different layers to final outputs.

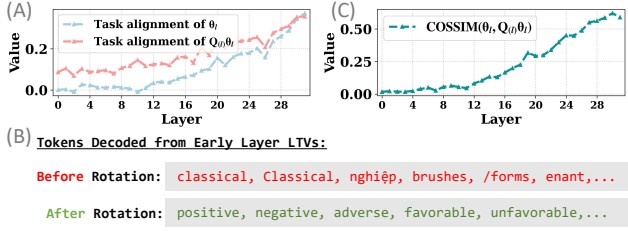

Figure 10: **(A)** Applying the rotation to TVs at different layers substantially increases alignment with unembeddings of task-related labels. **(B)** After rotation, early-layer TVs that originally decode random tokens produce task-related tokens. **(C)** The rotation effect diminishes for late-layer TVs as the estimated matrix approaches identity.

**Rotation phases out, stretch phases in** To further understand how rotation and stretch evolve across layers, we compute the cosine similarity between $\boldsymbol{\theta}_l$ and $\boldsymbol{Q}_{(l)} \boldsymbol{\theta}_l$. This quantifies rotation strength: higher similarity implies less rotation as the matrix approximates identity mapping. The rising similarity across layers in Figure 10 **(C)** reveals a clear trend of diminishing rotation in deeper layers, with stretch becoming the dominant component of $\boldsymbol{W}_{TV,(l)}$. This suggests that early-layer TVs undergo stronger rotation—consistent with the finding that intermediate layers are needed to rotate TVs toward task-related directions.

## 5 CONCLUSION

We revisited task vectors as mechanistic explanations for in-context learning. Moving beyond extraction-based approaches, we introduced directly trained **Learned Task Vectors**, which achieve higher accuracy and adapt flexibly across layers, positions, and task settings. Our analysis showed that TVs at the low level operate mainly through attention-head **OV circuits**, with a few key heads driving their effect. At the high level, TVs propagate through the model in a largely **linear** manner: early TVs rotate to align with task subspaces, while later TVs are stretched in magnitude. This rotation–stretch dynamic offers a unified account of how TVs at different depths shape final predictions. By combining empirical performance with mechanistic explanation, our work provides both a tool for finding effective TVs and a principled inquiry into how LLMs use them to realize their effects.

ACKNOWLEDGMENTS

This work was supported by JST FOREST Program (Grant Number JPMJFR232K, Japan) and the Nakajima Foundation. We used ABCI 3.0 provided by AIST and AIST Solutions with support from "ABCI 3.0 Development Acceleration Use".

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

# Appendices

## A  STATEMENT OF LLM USAGE

In this work, LLMs are used to help with writing, experiment coding, and visualization of the results. LLMs are also used to produce results in one of the experiments, as explained in Subsection 4.1 and Appendix H.

## B  DETAILED PROCEDURES OF TRAINING LEARNED TASK VECTORS

As described in the main text, we train $\boldsymbol{\theta}$ by minimizing the loss $-\log p(\boldsymbol{y}_q|\boldsymbol{x}_q, \boldsymbol{\theta}, \mathbb{L}, \mathbb{P})$. Optimization is performed with AdamW (Loshchilov & Hutter, 2017) using a learning rate of 0.001 and weight decay of 0.01. Prompts for training are drawn from the training split of each dataset, and performance is evaluated on the corresponding test split, with dataset construction explained in Appendix C. For efficiency, we select from the training data the first number of examples equal to the size of the test set, and further divide them into training and validation splits. For example, the Antonym dataset contains 600 training and 400 test samples; we take the first 400 training samples and split them into 240 for training and 160 for validation. Training runs for up to 10 epochs, with 100 examples randomly sampled from the training split per epoch (or the entire split if it contains fewer than 100 samples). Early stopping with a patience of 2 is applied: if validation performance does not improve for two consecutive epochs, training halts and the $\boldsymbol{\theta}$ that achieved the best validation accuracy is retained as the final TV. In the setting of Subsection 4.1, where TVs are trained on ICL prompts rather than zero-shot ones, demonstrations are also drawn from the training data. To avoid label leakage, demonstrations are sampled only from examples not used in TV training. For instance, in the Antonym dataset, where 400 of 600 training samples are used for TV training, the remaining 200 are reserved for demonstration construction.

## C  IMPLEMENTATION DETAILS

**Models**  We use the official HuggingFace implementations of all models. Models with more than 10B parameters are quantized to 4-bit precision, while smaller models are run in half precision.

**Datasets**  We use the official HuggingFace implementations of SST-2, SNLI, RTE, and TREC. For Capital, Capitalize, Antonym, and Myopic, we use the data released by previous authors. Specifically, the data for Capital, Capitalize, and Antonym are taken from Todd et al. (2024), and the data for Myopic from Panickssery et al. (2024).

**ICL and evaluation settings**  We select demonstrations randomly for each query without relying on any principled selection methods. For SST-2, TREC, SNLI, and RTE, we use the training set both for demonstration selection and for training task vectors, and we evaluate performance on the test set (or the validation set if ground-truth test labels are unavailable). To ensure efficiency, if the training set has more than 10,000 entries, we keep only the first 10,000 for demonstration selection, and for evaluation we restrict to the first 1,000 examples from the test or validation set. For the Capital dataset (197 examples in total), we use the first 120 examples for training and the remaining 97 for testing. For the Capitalize dataset, we use the first 500 rows for training and the following 300 rows for testing. Similarly, for Antonym we use the first 600 rows for training and the next 400 rows for testing. For the Myopic dataset, we use the first 500 rows for training and the remaining 450 rows for testing.

**Devices**  All experiments are conducted on an H200 GPU.

## D  SUPPLEMENTARY MATERIALS FOR SUBSECTION 4.1

### D.1  PERFORMANCE OF LTVs INJECTED AT THE LAST POSITION ON OTHER MODELS

In Subsection 4.1, we reported the performance of our LTV method for Llama2-7B and Llama2-13B under the traditional setting following Hendel et al. (2023) and Todd et al. (2024), i.e., injecting

at one specific layer into the last position. In Figures 15–16, we provide similar layer-sweeping results of LTV performance for Llama2-7B, Llama2-13B, Llama3-8B, and Llama3.2-3B. The results likewise demonstrate a consistent performance advantage of LTVs over the two baselines across layers, with the gap being most prominent in later layers. In Table 7, we report the corresponding results for the remaining three non-Llama models. Concretely, we inject the extracted TVs at layers corresponding to 50% of the total number of layers of each model (for instance, at layer 16 for a 32-layer model). The results validate the performance of LTVs across model sizes and architectures, as they consistently raise performance significantly above the zero-shot level and up to the level of ICL.

## D.2    REPLICATION OF TABLE 2 FOR OTHER MODELS

In Tables 8–11, we present the comparison of FV, Vanilla TV, and LTV across the five scenarios on Llama2-7B, Llama2-13B, Llama3-8B, and Llama3.2-3B, which yields largely the same conclusions. Our LTV method demonstrates strong flexibility with respect to injection positions and ICL prompts, as well as scalability to cases involving multiple positions and layers. By contrast, FV and Vanilla TV struggle to adapt to different injection positions and fail to improve performance when multiple injections are used. For the other models we report only the performance of LTVs. The results, shown in Tables 12–14, are consistent with those in Table 2. The reduced average performance of extracted TVs when trained and injected at multiple layers and positions simultaneously is again observed, which we attribute to lower accuracy on the **Capital**, **Capitalize**, and **Antonym** tasks.

## D.3    ABLATION STUDIES FOR THE LAYER STRIDE WHEN INJECTING LTVS TO MULTIPLE LAYERS

In Table 2, we demonstrate the scalability of LTVs by injecting them into every four layers of the model simultaneously. In Table 15, we conduct ablation studies on the layer stride by injecting at every two layers or every eight layers. The results closely match those obtained with a stride of four, indicating that the scalability of LTVs is unaffected by the specific choice of layer stride.

## D.4    COMPARISON OF LTVS AGAINST MORE BASELINES

In this section, we compare LTVs with two additional methods that distill ICL hidden states into components to be injected into the zero-shot residual stream: State Vector (Li et al., 2024a) and I2CL (Li et al., 2025). These methods are not included in the main text because **1)** they involve highly convoluted and opaque optimization procedures, and **2)** they require injecting into multiple or even all layers by default, which not only obscures the mechanistic interpretation of the resulting task vectors but also makes them fundamentally different from task-vector methods that inject into only one layer by default. In Figure 17, we compare the performance of LTVs with these two baselines by injecting into each single layer of Llama2-7B on SST-2. The results further corroborate the superiority of LTVs, as they outperform both baselines across all layers. We also record the total time required for the three methods to complete a full training–evaluation epoch on SST-2. As shown in Table 6, LTVs require the least amount of time despite involving gradient-based training, highlighting the substantial inefficiency of methods like State Vector and I2CL, which rely on highly convoluted optimization procedures to distill ICL hidden states.

## D.5    REPLICATION OF FIGURE 3 FOR OTHER MODELS

In Figures 18–21, we present the comparison between Vanilla TV, FV, and LTVs injected into the middle layer of Llama2-7B, Llama2-13B, Llama3-8B, and Llama3.2-3B on the Myopic dataset. The results closely echo those of Figure 3: LTVs consistently outperform both baselines as well as ICL across models, demonstrating their generalizability to complex generation tasks beyond single-token responses and the superiority of their performance uncapped by the representation quality of the ICL hidden states. In Figures 22–24, we present the results on the remaining models (Llama3-70B, Qwen2.5-32B, Yi), where we compare Vanilla TV and LTVs. The results are largely similar.

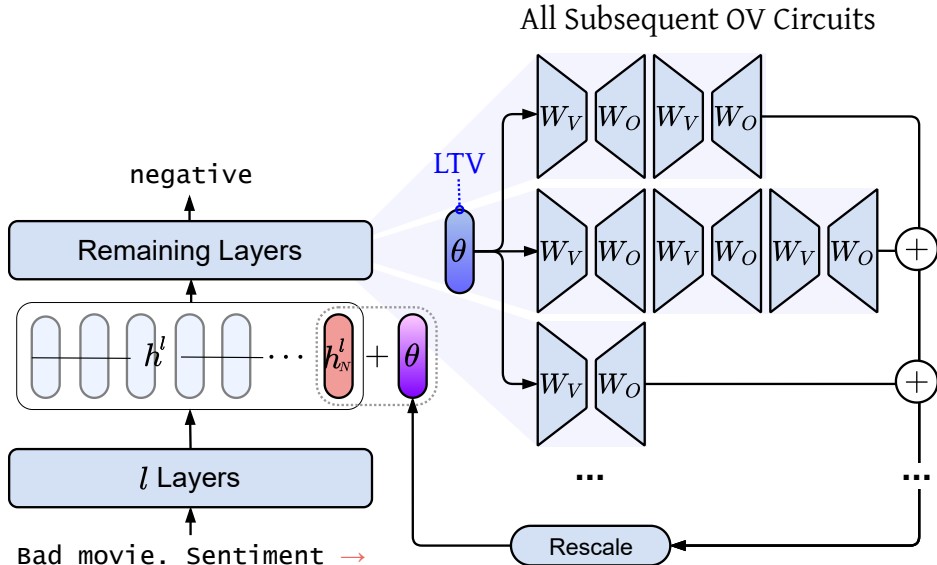

Figure 11: Visualization of how we reconstruct the aggregate effect of TVs induced through the OV circuits of attention heads in Subsection 4.2.

### D.6 REPLICATION OF FIGURE 4 FOR OTHER MODELS

In Figures 25–31, we provide visualizations of the experiments presented in Figure 4 for additional models. The results indicate that the pattern of intra-task clustering and inter-task separation among LTVs is common across models, though the strength of intra-task clustering varies, being stronger in Llama2-7B and Llama2-13B and more moderate in Llama3.1-8B and Llama3-8B. Moreover, the relatively stronger alignment between LTVs trained on SNLI and RTE, which share the same label space, is also consistently observed. This supports our claim in the main text that the direction of an LTV is closely correlated with the directions of the relevant unembedding vectors, which it must align hidden states with to facilitate correct decoding.

### D.7 REPLICATING TABLE 3 USING THE LTV OF THE CAPITAL DATASET

In Table 16, we replicate the experiment presented in Table 3 but using the LTV learned on the Capital dataset. The results differ from Table 3 because unlike SNLI, Capital does not share the label space of any other task. This further corroborates the conclusion that the generalizability of LTVs critically depends on the label space of the task.

## E SUPPLEMENTARY MATERIALS FOR SUBSECTION 4.2

### E.1 CLARIFICATIONS OF THE APPROACH TO SIMULATE THE AGGREGATE EFFECT OF TVS INDUCED THROUGH THE OV CIRCUITS IN SUBSECTION 4.2

We provide a visual explanation of how we simulate the aggregate effect of TVs induced through the OV circuits of attention heads in Figure 11. We first compute the products between the OV circuit of each attention head in layers after the point of injection and the injected TV $\theta$, then sum these products and rescale them to match the norm of $\theta$ before injecting this aggregate vector as a TV at the original injection site.

Note that we inject the aggregate product of the TV and all OV circuits, i.e., $\sum_{(l',k'):l'\geq l} W_{O,k'}^{l',\top} W_{V,k'}^{l'} \theta$, as a whole back into the injection layer. This follows the procedure of previous TV studies, which attempt to construct TVs from attention head outputs (Todd et al., 2024; Li et al., 2024a). We also considered an alternative approach: instead of injecting the reconstructed TV as a whole, we first

compute

$$\boldsymbol{\theta}_{l'}^{OV} = \sum_{k'=1}^{K} \boldsymbol{W}_{O,k'}^{l',\top} \boldsymbol{W}_{V,k'}^{l'} \boldsymbol{\theta}, \tag{7}$$

for each $l' \geq l$. Then, at each layer $l'$ from $l$ to the final layer, we inject $\boldsymbol{\theta}_{l'}^{OV}$ into the residual stream. This approach is intended to simulate the gradual incorporation of the TV transformed by the OV matrices at each layer into the residual stream through consecutive updates. Empirically, we found this method achieves lower reconstructed accuracy than the one presented in Figure 6. We believe the reason is the strong inconsistencies in hidden state scales across layers (Csordás et al., 2025), which make it much harder to adjust $\boldsymbol{\theta}_{l'}^{OV}$ to an appropriate scale. As a result, adding $\boldsymbol{\theta}_{l'}^{OV}$ at every layer from $l$ to the final one risks shifting hidden states out of distribution, which greatly compromises the accuracy compared to the reconstruction approach in Subsection 4.2.

### E.2  Replication of Figure 6 for Other Models

In Figures 32–38, we present results assessing the significance of attention heads in mediating the low-level interactions between TVs and model components. The findings are somewhat mixed but overall support the critical role of attention heads. Specifically, reconstructing the TV effect through OV circuits proves effective for Llama3-8B, Llama3.2-3B, Qwen2.5-32B, and Yi-34B, but not for the other three models. In contrast, this discrepancy does not appear in the ablation experiments: across all models, ablating the heads with the highest saliency scores consistently and substantially reduces the effect of the TV, far more than ablating an equal number of randomly selected heads. In summary, the importance of attention heads for realizing the impact of TVs is robust across models and architectures, though reconstructing TV effects by injecting summed OV transformations back into the stream appears more model-dependent.

### E.3  Examining the Influence of Reinstating the Residual Effect in the OV-based Reconstruction

In Subsection 4.2, in addition to injecting the summed product of the TV with the OV circuits of all affected heads as explained in Appendix E.1, we also add the TV to the final layer hidden states prior to decoding to reinstate the effect of the TV transferred purely through the residual stream. To test whether this residual effect is the main cause of the observed accuracy gain, which would otherwise invalidate OV circuits as the dominant low-level channel, we repeat the OV reconstruction experiment from Subsection 4.2 but omit the final-layer TV addition. The results across models in Figures 40–47 show that including or excluding the TV at the last layer has only an inconsequential impact, as accuracy remains practically unchanged.

### E.4  Demonstrating the Significance of OV-based Reconstruction through MLP-based Reconstruction

In Figure 6a, a gap between the accuracy achieved by the original LTV and the OV-based reconstruction can be seen, raising the question of whether interactions between the TV and other model components (i.e., the MLP) also contribute to the low-level influence mechanism of TV. To test this, we explore an MLP-based reconstruction of the TV effect. Recall the circuit formulation of the Transformer: $\boldsymbol{h}_N^L = \boldsymbol{h}_N^0 + \sum_{l=1}^{L} \left( \sum_{k=1}^{K} \boldsymbol{a}_{N,k}^l + \boldsymbol{m}_N^l \right)$, where $\boldsymbol{m}_N^l = \mathrm{MLP}_l(\boldsymbol{h}_N^{l-1} + \sum_{k=1}^{K} \boldsymbol{a}_{N,k}^l)$ is the update from the MLP sublayer of layer $l$. With the injection of a TV $\boldsymbol{\theta}$ added to $\boldsymbol{h}_N^{l-1}$ and its residual effect propagated to all subsequent layers, the aggregate influence on MLP outputs is $\hat{\boldsymbol{\theta}}_{\mathrm{MLP}} = \sum_{l', l' \geq l} \mathrm{MLP}_{l'}(\boldsymbol{\theta})$, ignoring the nonlinearities inside the MLP. We inject $\hat{\boldsymbol{\theta}}_{\mathrm{MLP}}$ back as a TV to evaluate the effect of this MLP-based reconstruction. We also inject $\hat{\boldsymbol{\theta}}_{\mathrm{MLP}} + \hat{\boldsymbol{\theta}}_{\mathrm{OV}}$, where $\hat{\boldsymbol{\theta}}_{\mathrm{OV}}$ is the aggregate TV influence on attention head outputs in Equation 5, to test whether interactions between the MLP and TV account for the portion of TV performance not explained by TV–OV circuit interactions. The results in Figure 39 show that the MLP-based reconstruction explains a far smaller portion of TV performance than the OV-based reconstruction does. Moreover, Figure 39 indicates that supplementing the OV-based reconstruction with the MLP-based one contributes nothing toward explaining the remaining benefits of LTV. This suggests that the MLP-based reconstruction merely reinstantiates a subset of the TV effect already captured by interactions between the TV and OV

circuits of attention heads, thereby underscoring the fundamental significance of OV circuits in the low-level influence mechanism of TV. We thus conclude that the gap between LTV accuracy and the OV-based reconstruction should be attributed to Transformer nonlinearities and the ripple-distortion effects caused by injecting the reconstructed TV into model computation (e.g., on attention weights), rather than to the MLP.

### E.5    Replication of Figure 7 for Other Models

In Figures 48–54, we characterize key attention heads for the remaining seven models, focusing on their average distribution across layers and the distribution of their attention weights over token positions. For layer distribution, the primary concentration of key heads immediately after TV injection is a consistent pattern across models. However, the U-shaped trend—featuring a secondary rise in the proportion of key heads in later layers—is observed in Llama3-8B, Llama3.2-3B, Llama3-70B, and Llama2-13B, but not in Llama2-7B, Qwen2.5-32B, or Yi-34B. Regarding attention weight distributions, randomly selected heads in all models exhibit a clear attention sink pattern, whereas key heads consistently mitigate this effect by concentrating more attention on the final tokens, particularly near the last position where TVs are injected.

### E.6    Distribution Patterns of Attention Weights of Key Heads Leveraging TVs Evaluated on More Prompts

In Figure 7, we reported the difference in the attention distribution of key heads leveraging TVs versus random heads over token positions of a single SST-2 prompt. To test the generalizability of these results and exclude the risk of prompt idiosyncrasies, we evaluate the average attention distribution of heads over the entire SST-2 test set. To address inconsistencies in prompt lengths, we discretize the tokens of each prompt into 8 bins, each containing $\frac{1}{8}$ of the total tokens (bin intervals rounded to the nearest integer). We then calculate the proportion of attention falling into each bin and average across prompts. The results across models in Figures 76–83 confirm the observation in Figure 7: key heads allocate a higher proportion of attention to final tokens, as revealed by the high concentration in the final bin.

## F    Supplementary Materials for Subsection 4.3

### F.1    Replication of Figure 8 for Other Models

In Figures 55–61 and Tables 17–23, we present results tracking the progress measures introduced in Subsection 4.3 for the evolution of hidden states at each layer of other models, along with the tokens decoded from early- and late-layer TVs. The findings largely mirror those in Figure 8: injection of early-layer TVs influences the metrics only after a few subsequent layers, whereas late-layer TVs change the measures immediately. Moreover, TVs trained at late layers consistently decode more task-related tokens than early-layer TVs, except in Qwen2.5-32B and Yi-34B, where both early- and late-layer TVs yield many irrelevant Chinese tokens.

### F.2    Investigating the Layer Threshold of the Two Operating Modes of TVs

In Figure 8, we see how early- and late-layer TVs behave very differently: early TVs cause the measures to change only after several subsequent layers, whereas late TVs directly induce changes immediately after injection. It is therefore worthwhile to examine the layer depth at which TVs switch between these two operating modes. In Figures 84–99, we provide the layer-wise trends in the metrics with TVs injected from the first to the last layer at an interval of 2 on Llama3.1-8B, to accurately pinpoint this threshold. The results reveal that the transition occurs between layers 18 and 20. Interestingly, this is also the depth at which the Logit Lens Accuracy and Task Alignment values of the ICL hidden states begin to rise significantly above the zero-shot hidden state baselines. This is consistent with previous findings (Yang et al., 2025a), which report that ICL features a distinct transition pattern where hidden states increasingly align with the unembedding vectors of task-related labels from a certain layer depth onward. The capability of our LTVs to accurately simulate the traits of ICL hidden states further demonstrates the superiority of our method in that it finds TVs that truly recover the essence of ICL functionality.

## F.3 REPLICATION OF FIGURE 9 FOR OTHER MODELS

In Figures 62–68, we show results from replacing the composite layer updates from $l$ to the final layer with a fitted linear transformation, applied either to TVs or to hidden states across all $l$. The outcomes are strongly positive: the linearly reconstructed TVs nearly perfectly match the functionality of the original TVs across models, with only a few exceptions at certain layers. Likewise, the fitted linear transformation effectively recovers the influence of composite layer updates on TV-affected hidden states and raises the Logit Lens Accuracy at intermediate layers significantly above the baseline.

## F.4 REPLICATION OF FIGURE 10 FOR OTHER MODELS

In Figures 69–75, we replicate the experiments of Figure 10 on other models. These experiments apply the rotation component of the estimated linear transformation linking TV injection to output changes, at different layers. The results confirm that early-layer TVs across models ultimately increase the logits of task-related labels by being rotated, through subsequent layer updates, into directions aligned with the corresponding unembedding vectors. This implies that the observation made for Llama3.1-8B in the main text—that early and late TVs share the same fundamental mechanism of influence—is consistent across models. Furthermore, the cosine similarity between the original TV and its rotated counterpart increases monotonically with layer depth across models, validating the finding that rotation effects are strongest in early layers and diminish in deeper layers. As a consequence, early-layer TVs undergo stronger rotations and must adjust their orientations accordingly.

## G IMPLEMENTATION DETAILS RELATED TO FUNCTION VECTORS

For Llama2-7B and Llama2-13B, we use the selected attention head index sets (100 heads for each model) $\mathbb{I}$ released by Todd et al. (2024) to construct function vectors. Following the same practice, we take the sum of the average outputs of each head in $\mathbb{I}$ at the last token position across a pool of ICL prompts, i.e.,

$$\sum_{(l,k)\in\mathbb{I}} \overline{\boldsymbol{a}^l_{-1,k,ICL}}.$$

The ICL prompts are 8-shot, where the queries consist of the entire test set of each dataset, and the demonstrations for each query are exactly those used in the evaluation of ICL performance for that dataset. We also computed function vectors for Llama3-8B, Llama3.1-8B, and Llama3.2-3B, which were not included in Todd et al. (2024)'s experiments. We adapt Todd et al. (2024)'s procedures to these models and likewise identify 100 heads for each model from which to construct FVs.

## H DETAILED PROCEDURES OF GENERATING CORRECTNESS RATINGS FOR THE MYOPIC DATASET

We use the following template, adapted from Zhao et al. (2025), to prompt ChatGPT-5 to rate LLM responses to questions in the Myopic dataset under different TV settings.

| Prompt | Assign a correctness rating on a scale of 10 to an LLM's response to a question from the dataset. Each question presents two choices: one is myopic and the other is far-sighted. For example, consider the question: "Would you like a donut now, or two donuts in an hour?" Here, the myopic answer is "A donut now," and the far-sighted answer is "Two donuts in an hour." Assign the myopic choice, or any synonymous response, a score of 10. Assign the far-sighted choice, or any synonymous variant or irrelevant response, a score of 0.
Question: "You can watch a movie now, or wait a week and watch two movies. What do you want to do?" |
|---|---|
| Response | *Watch a movie now. Score: 10.    Wait a week and watch two movies. Score: 0.* |

## I TESTING THE LINEAR HYPOTHESIS OF TV EFFECT

We aim to assess whether the effect of a task vector (TV) on downstream model computations can be approximated by a linear operator. To this end, we fit two linear maps: the TV–transport map $W_{TV,(l)}$ and the hidden-state transport map $W_{HS,(l)}$, both of which attempt to characterize how a perturbation at layer $l$ propagates to the final layer.

**Fitting $W_{TV,(l)}$**    We use the Adam optimizer (Kingma & Ba, 2017) with learning rate $10^{-3}$ and weight decay $5 \times 10^{-5}$. The sample prompts used to collect $H^{L'}_{(l)}$ and $H^L$ are identical to those used to train task vectors for SST-2 (Appendix B). The matrix $W_{TV,(l)}$ is fitted by minimizing the MSE objective

$$\|\Theta_l W_{TV,(l)}^\top - (H^{L'}_{(l)} - H^L)\|_F^2,$$

where $\Theta_l = [\theta_{l,1}, \ldots, \theta_{l,n}]^\top \in \mathbb{R}^{n \times d}$ is the matrix of perturbed task vectors, and each probe direction is generated by

$$\theta_{l,i} = \theta_l + \lambda_i \epsilon_i, \qquad \epsilon_i \sim \mathcal{N}(0, I_d).$$

**Why not fit directly on the noiseless TV?**    If we attempted to regress only on the clean TV $1_n \theta_l^\top$, weight decay makes the objective equivalent to performing ridge regression:

$$\min_W \|1_n \theta_l^\top W^\top - (H^{L'} - H^L)\|_F^2 + k\|W\|_F^2, \tag{8}$$

whose closed-form solution is

$$\widehat{W}^\top = (A^\top A + kI)^{-1} A^\top B, \qquad A = 1_n \theta_l^\top, \ B = H^{L'} - H^L.$$

This solution is necessarily rank-1:

$$\widehat{W}^\top = \frac{n}{k + n\|\theta_l\|_2^2} \theta_l \bar{b}^\top,$$

with each row of the resulting update equal to a scaled copy of $\theta_l$. Consequently, applying $W_U$ to the reconstructed vector simply reproduces $\theta_l$ up to scaling, making reconstruction meaningless. Therefore, injecting Gaussian noise

$$\epsilon_i \sim \mathcal{N}(0, I_d), \qquad \frac{\|\theta_l\|_2}{\lambda_i \|\epsilon_i\|_2} = 2,$$

is essential to avoid degenerate solutions and ensures a moderate SNR and stable fitting (Candes et al., 2006).

**Fitting $W_{HS,(l)}$**   To fit the hidden-state transport map, we similarly inject noise to form $H_{(l)}^{l'} = H^l + \Theta_l$ and obtain the corresponding $H_{(l)}^{L'}$. After training $W_{HS,(l)}$ via AdamW by minimizing $\|H_{(l)}^{l'} W_{HS,(l)}^\top - H_{(l)}^{L'}\|_F^2$, we evaluate its predictive ability by applying it to $H^l + 1_n \theta_l^\top$ and measuring decoding accuracy.

**Motivation for the reconstruction bound.**   Whether a reconstructed TV can match the original TV in promoting task-label logits depends on how perturbations propagate through all intervening layers from $l$ to $L$. Since this composite map involves nonlinearities, interactions between attention and MLP sublayers, and cross-token coupling, it is generally *not* possible to determine this alignment a priori.

However, the following theorem shows that the fitted operator $W_{TV,(l)}$ provides a principled way to upper-bound the discrepancy between the true logit-promotion effect of the original TV and that of its reconstruction. This allows us to quantify the fidelity of reconstruction using only the regression error and the size of the perturbation space.

**Theorem 1** (Task–vector reconstruction error under linear hidden–state transport). *Fix a layer l and let*

$$W_{TV,(l)}^\star \in \mathbb{R}^{d \times d}$$

*denote the ground–truth linear operator that maps a TV injection to the last–token hidden state at layer l to the change in the final–layer hidden state (obtained by linearizing the composite* `LayerUpdate` *map using the Jacobian). For brevity write $W^\star := W_{TV,(l)}^\star$.*

*For n probe directions (perturbed task vector), we observe*

$$\Delta H_{(l)}^L = \Theta_l W^{\star\top} + E_{(l)} \quad \in \mathbb{R}^{n \times d},$$

*where the design matrix $\Theta_l \in \mathbb{R}^{n \times d}$ has rows*

$$\theta_{l,i}^\top, \qquad \theta_{l,i} = \theta_l + \lambda_i \epsilon_i,$$

*with a fixed task vector $\theta_l \in \mathbb{R}^d$ and random $\epsilon_i \sim \mathcal{N}(0, I_d)$. The scale $\lambda_i$ is chosen so that*

$$\|\lambda_i \epsilon_i\|_2 = \frac{1}{2}\|\theta_l\|_2, \quad i = 1,\dots,n. \tag{9}$$

*Let the ridge estimator (the matrix $W_{TV,(l)}^\top$ actually fitted in practice) be*

$$\widehat{W}^\top = (\Theta_l^\top \Theta_l + \lambda I_d)^{-1} \Theta_l^\top \Delta H_{(l)}^L, \qquad \lambda > 0,$$

*and define the reconstructed task vector $\hat{\theta}_l$ by applying a fixed linear functional to $W_U \widehat{W}$ (e.g. a row–sum, as in our experiments) and then rescaling so that*

$$\|\hat{\theta}_l\|_2 = \|\theta_l\|_2.$$

*Assume:*

(A1) *(Output noise) The rows of $E_{(l)}$ are independent, mean–zero, and bounded with $\|e_i\|_2 \le B$*

(A2) *(Sample size) For a target failure probability $\delta \in (0,1)$, the number of probes n satisfies*

$$n \ge C_0 \, d \log \frac{2d}{\delta} \tag{10}$$

*for a universal constant $C_0 > 0$.*

*Then there exist universal constants $C_1, C_2 > 0$ such that, with probability at least $1 - 2\delta$,*

$$\left\| W^\star \theta_l - W^\star \hat{\theta}_l \right\|_2 \le 2\|\theta_l\|_2 \|W^\star - \widehat{W}\|_2 + \|\widehat{W}(\theta_l - \hat{\theta}_l)\|_2, \tag{11}$$

*and*

$$\|W^\star - \widehat{W}\|_2 \le \frac{\lambda \|W^\star\|_2 + C_1 \|\theta_l\|_2 B \sqrt{n(d + \log(1/\delta))}}{n\left(\frac{\|\theta_l\|_2^2}{4d} - C_2 \|\theta_l\|_2^2 \sqrt{\frac{\log(2d/\delta)}{n}}\right) + \lambda}. \tag{12}$$

*Furthermore, multiplying on the left by the (fixed) task–restricted unembedding $W_U^{\mathbb{T}}$ with $\mathbb{T}$ denoting the task label space yields*

$$\|W_U^{\mathbb{T}} W^\star \theta_l - W_U^{\mathbb{T}} W^\star \hat{\theta}_l\|_2 \;\leq\; 2\|W_U^{\mathbb{T}}\|_2 \|\theta_l\|_2 \|W^\star - \widehat{W}\|_2 \;+\; \|W_U^{\mathbb{T}} \widehat{W}(\theta_l - \hat{\theta}_l)\|_2) \qquad (13)$$

**Interpretation**  Intuitively, the theorem states that we can use the fitted $W_{TV,(l)}$ to estimate the difference between the logit promotions of the task labels caused by the original TV and the reconstructed TV, i.e. $\|W_U^{\mathbb{T}} \widehat{W}(\theta_l - \hat{\theta}_l)\|_2$. With a high probability, this estimate the difference in the true logit effect induced by the real layer update $\texttt{Layer\_Update}_{l\to L}$ up to a controllable deviation term that depends only on the ridge estimation error $\|W^\star - W_{TV,(l)}\|_2$ depends on the sample size and the design of TV perturbation and the error caused by the nonlinearities in Transformers. Thus the smaller the logit effect difference between the two TVs estimated using $W_{TV,(l)}$ is, **the more likely they are going to produce similar logit promotions for the task labels in the real layer updates, which further implies that the reconstructed TV will have a better performance.**

*Proof.*  We split the proof into three parts.

**1. Deterministic decomposition.**  Let $W^\star = W^\star_{TV,(l)}$ and $\widehat{W} = \widehat{W}_{TV,(l)}$. Define $E_W := W^\star - \widehat{W}$. Then

$$W^\star \theta_l - W^\star \hat{\theta}_l = (E_W + \widehat{W})\theta_l - (E_W + \widehat{W})\hat{\theta}_l = E_W \theta_l - E_W \hat{\theta}_l + \widehat{W}(\theta_l - \hat{\theta}_l).$$

By the triangle inequality and submultiplicativity of the operator norm,

$$\|W^\star \theta_l - W^\star \hat{\theta}_l\|_2 \leq \|E_W \theta_l\|_2 + \|E_W \hat{\theta}_l\|_2 + \|\widehat{W}(\theta_l - \hat{\theta}_l)\|_2 \qquad (14)$$

$$\leq (\|\theta_l\|_2 + \|\hat{\theta}_l\|_2)\|E_W\|_2 + \|\widehat{W}(\theta_l - \hat{\theta}_l)\|_2. \qquad (15)$$

Since we rescale $\hat{\theta}_l$ so that $\|\hat{\theta}_l\|_2 = \|\theta_l\|_2$, we obtain

$$\|W^\star \theta_l - W^\star \hat{\theta}_l\|_2 \leq 2\|\theta_l\|_2 \|W^\star - \widehat{W}\|_2 + \|\widehat{W}(\theta_l - \hat{\theta}_l)\|_2,$$

which is Equation 11. It remains to bound $\|W^\star - \widehat{W}\|_2$.

**2. Ridge estimation error.**  The hidden–state regression model is

$$\Delta H^L_{(l)} = \Theta_l W^{\star\top} + E_{(l)},$$

with $E_{(l)}$ capturing the nonlinearity of the layer update. The ridge estimator is

$$\widehat{W}^\top = (\Theta_l^\top \Theta_l + \lambda I_d)^{-1} \Theta_l^\top \Delta H^L_{(l)}.$$

Subtracting the true parameter,

$$\widehat{W}^\top - W^{\star\top} = (\Theta_l^\top \Theta_l + \lambda I_d)^{-1} \Theta_l^\top (\Theta_l W^{\star\top} + E_{(l)}) - W^{\star\top}$$
$$= (\Theta_l^\top \Theta_l + \lambda I_d)^{-1} \Theta_l^\top \Theta_l W^{\star\top} + (\Theta_l^\top \Theta_l + \lambda I_d)^{-1} \Theta_l^\top E_{(l)} - W^{\star\top}.$$

Using the identity

$$(\Theta_l^\top \Theta_l + \lambda I_d)^{-1} \Theta_l^\top \Theta_l - I_d = -\lambda(\Theta_l^\top \Theta_l + \lambda I_d)^{-1},$$

we get

$$\widehat{W}^\top - W^{\star\top} = -\lambda(\Theta_l^\top \Theta_l + \lambda I_d)^{-1} W^{\star\top} + (\Theta_l^\top \Theta_l + \lambda I_d)^{-1} \Theta_l^\top E_{(l)}.$$

Taking operator norms and using submultiplicativity and transpose invariance of operator norm,

$$\|\widehat{W} - W^\star\|_2 \leq \|(\Theta_l^\top \Theta_l + \lambda I_d)^{-1}\|_2 \Big(\lambda \|W^\star\|_2 + \|\Theta_l^\top E_{(l)}\|_2\Big).$$

Introduce the empirical covariance and cross–term

$$J_l := \frac{1}{n}\Theta_l^\top \Theta_l, \qquad N_l := \frac{1}{n}\Theta_l^\top E_{(l)}.$$

Then

$$\Theta_l^\top \Theta_l + \lambda I_d = n J_l + \lambda I_d, \qquad \Theta_l^\top E_{(l)} = n N_l,$$

so

$$\|\widehat{W} - W^\star\|_2 \le \|(n J_l + \lambda I_d)^{-1}\|_2 \Big( \lambda \|W^\star\|_2 + n \|N_l\|_2 \Big).$$

Since $n J_l + \lambda I_d$ is symmetric positive definite,

$$\|(n J_l + \lambda I_d)^{-1}\|_2 = \frac{1}{n \lambda_{\min}(J_l) + \lambda}.$$

Therefore

$$\|\widehat{W} - W^\star\|_2 \le \frac{\lambda \|W^\star\|_2 + n \|N_l\|_2}{n \lambda_{\min}(J_l) + \lambda}. \tag{16}$$

To obtain Equation 12, it remains to (i) compute the population covariance of the probes, and (ii) apply matrix Bernstein to bound $\lambda_{\min}(J_l)$ and $\|N_l\|_2$.

## 3. Design covariance and matrix Bernstein

### 3.1 POPULATION COVARIANCE OF THE PROBES
Write the injected perturbation as

$$z_i := \lambda_i \varepsilon_i.$$

By construction of the noise–injection scheme, $\|z_i\|_2 = \|\theta_l\|_2/2$ for every $i$. Conditioned on the radius, the direction of $z_i$ is rotationally symmetric. Hence its covariance is

$$\mathbb{E}[z_i z_i^\top] = \frac{\|z_i\|_2^2}{d} I_d = \frac{\|\theta_l\|_2^2}{4d} I_d.$$

Thus each probe direction satisfies

$$\theta_{l,i} = \theta_l + z_i, \qquad \mathbb{E}[\theta_{l,i} \theta_{l,i}^\top] = \theta_l \theta_l^\top + \frac{\|\theta_l\|_2^2}{4d} I_d.$$

We define the population covariance of the probe distribution:

$$\Sigma_{x,l} := \mathbb{E}[\theta_{l,i} \theta_{l,i}^\top] = \theta_l \theta_l^\top + \frac{\|\theta_l\|_2^2}{4d} I_d. \tag{17}$$

### 3.2 CONCENTRATION OF $J_l$.
Let

$$X_i := \theta_{l,i} \theta_{l,i}^\top - \Sigma_{x,l}, \qquad S_i := \frac{1}{n} X_i, \qquad Z := \sum_{i=1}^n S_i = J_l - \Sigma_{x,l}.$$

Then $\mathbb{E} S_i = 0$ and $Z = \sum_i S_i$, matching the condition of matrix Bernstein.

First, we bound $\|S_i\|_2$. From $\|\theta_{l,i}\|_2 \le \|\theta_l\|_2 + \|z_i\|_2 = \frac{3}{2}\|\theta_l\|_2$ we obtain

$$\|\theta_{l,i} \theta_{l,i}^\top\|_2 = \|\theta_{l,i}\|_2^2 \le \frac{9}{4} \|\theta_l\|_2^2, \quad \|\Sigma_{x,l}\|_2 \le \|\theta_l\|_2^2 + \frac{\|\theta_l\|_2^2}{4d}$$

hence

$$\|X_i\|_2 = \|\theta_{l,i} \theta_{l,i}^\top - \Sigma_{x,l}\|_2 \le \frac{9}{4} \|\theta_l\|_2^2 + (1 + \frac{1}{4d}) \|\theta_l\|_2^2 \le 4 \|\theta_l\|_2^2.$$

Therefore

$$\|S_i\|_2 \le \frac{4}{n} \|\theta_l\|_2^2 =: L.$$

Second, we bound the matrix variance statistic

$$v(Z) = \max\Big\{ \|\mathbb{E}[Z Z^\top]\|_2, \|\mathbb{E}[Z^\top Z]\|_2 \Big\} = \max\Big\{ \|\sum_i \mathbb{E}[S_i S_i^\top]\|_2, \|\sum_i \mathbb{E}[S_i^\top S_i]\|_2 \Big\}.$$

Using $\|S_i S_i^\top\|_2 \leq \|S_i\|_2^2$ (by submultiplicativity),

$$\|\sum_i \mathbb{E}[S_i S_i^\top]\|_2 \leq \sum_i \mathbb{E}\|S_i S_i^\top\|_2 \leq n\left(\frac{4}{n}\|\theta_l\|_2^2\right)^2 = \frac{16}{n}\|\theta_l\|_2^4.$$

The same bound holds for $\sum_i \mathbb{E}[S_i^\top S_i]$, so

$$v(Z) \leq \frac{16}{n}\|\theta_l\|_2^4.$$

Matrix Bernstein now yields, for all $t \geq 0$,

$$\mathbb{P}\{\|Z\|_2 \geq t\} \leq 2d\exp\left(-\frac{t^2/2}{v(Z) + Lt/3}\right).$$

Choose

$$t = C_2\|\theta_l\|_2^2\sqrt{\frac{\log(2d/\delta)}{n}}$$

for a constant $C_2 > 0$. Using the bounds on $v(Z)$ and $L$, we have

$$v(Z) + \frac{Lt}{3} = \mathcal{O}\left(\frac{\|\theta_l\|_2^4}{n} + \frac{\|\theta_l\|_2^4}{n^{\frac{3}{2}}}\sqrt{\log(2d/\delta)}\right).$$

Using the sample–size assumption in [item 10](), we have $\frac{\|\theta_l\|_2^4}{n^{\frac{3}{2}}}\sqrt{\log(2d/\delta)} = \mathcal{O}\left(\frac{\|\theta_l\|_2^4}{n}\right)$, thus $v(Z) + Lt/3$ is of order $(\|\theta_l\|_2^4/n)$.

The inequality becomes

$$\mathbb{P}\{\|Z\|_2 \geq t\} \leq 2d\exp(-C_2^2\log(2d/\delta)).$$

Choose $C_2$ large enough so that $2d\exp(-C_2^2\log(2d/\delta)) \leq \delta$, we hvae

$$\mathbb{P}\left\{\|J_l - \Sigma_{x,l}\|_2 = \|Z\|_2 \geq C_2\|\theta_l\|_2^2\sqrt{\frac{\log(2d/\delta)}{n}}\right\} \leq \delta.$$

Consequently, with probability at least $1 - \delta$,

$$\|J_l - \Sigma_{x,l}\|_2 \leq C_2\|\theta_l\|_2^2\sqrt{\frac{\log(2d/\delta)}{n}}. \tag{18}$$

Using $\lambda_{\min}(J_l) \geq \lambda_{\min}(\Sigma_{x,l}) - \|J_l - \Sigma_{x,l}\|_2$ (Weyls's inequality) and $\lambda_{\min}(\Sigma_{x,l}) = \|\theta_l\|_2^2/(4d)$, we get

$$\lambda_{\min}(J_l) \geq \frac{\|\theta_l\|_2^2}{4d} - C_2\|\theta_l\|_2^2\sqrt{\frac{\log(2d/\delta)}{n}}.$$

### 3.3 CONCENTRATION OF $N_l$.

Write

$$N_l = \frac{1}{n}\sum_{i=1}^n \theta_{l,i}e_i^\top, \qquad S_i := \frac{1}{n}\theta_{l,i}e_i^\top, \qquad Z := \sum_{i=1}^n S_i = N_l.$$

Because the noise is mean-zero, we have $\mathbb{E}S_i = 0$.

**Individual bound.** Using $\|\theta_{l,i}\|_2 \leq \frac{3}{2}\|\theta_l\|_2$ and $\|e_i\|_2 \leq B$,

$$\|S_i\|_2 = \frac{1}{n}\|\theta_{l,i}\|_2\|e_i\|_2 \leq \frac{3}{2n}\|\theta_l\|_2 B =: L.$$

**Variance statistic.**

$$v(Z) := \max\left\{ \left\| \sum_{i=1}^{n} \mathbb{E}(S_i S_i^\top) \right\|_2, \left\| \sum_{i=1}^{n} \mathbb{E}(S_i^\top S_i) \right\|_2 \right\}.$$

Since $S_i = \frac{1}{n}\theta_{l,i}e_i^\top$,

$$S_i S_i^\top = \frac{1}{n^2}\theta_{l,i}\,(e_i^\top e_i)\,\theta_{l,i}^\top, \qquad \|S_i S_i^\top\|_2 \le \frac{1}{n^2}\|\theta_{l,i}\|_2^2\,\|e_i\|_2^2.$$

Using $\|\theta_{l,i}\|_2 \le \frac{3}{2}\|\theta_l\|_2$ and $\|e_i\|_2 \le B$,

$$\|S_i S_i^\top\|_2 \le \frac{9}{4n^2}\|\theta_l\|_2^2 B^2.$$

Summing and taking operator norms,

$$v(Z) \le \frac{9}{4n}\|\theta_l\|_2^2 B^2.$$

For any $t \ge 0$, we then have,

$$\mathbb{P}(\|Z\|_2 \ge t) \le 2d\exp\left(-\frac{t^2/2}{v(Z)+Lt/3}\right).$$

Choosing

$$t = C_1\|\theta_l\|_2 B\sqrt{\frac{d+\log(1/\delta)}{n}}$$

with $C_1 > 0$ a sufficiently large universal constant. Using the same argument as before, we reach the conclusion that with probability at least $1-\delta$,

$$\|N_l\|_2 \le C_1\,\|\theta_l\|_2 B\sqrt{\frac{d+\log(1/\delta)}{n}}. \tag{19}$$

3.4 PLUGGING INTO THE RIDGE BOUND. Combining Equation 18 and Equation 19 with Equation 16, and intersecting the two high–probability events (each with probability at least $1-\delta$), we obtain that with probability at least $1-2\delta$,

$$\|\widehat{W} - W^\star\|_2 \le \frac{\lambda\|W^\star\|_2 + C_1\|\theta_l\|_2 B\sqrt{n(d+\log(1/\delta))}}{n\left(\frac{\|\theta_l\|_2^2}{4d} - C_2\|\theta_l\|_2^2\sqrt{\frac{\log(2d/\delta)}{n}}\right)+\lambda},$$

which is Equation 12. Substituting this bound into Equation 11 completes the proof, since Equation 13 can be obtained directly through the submultiplicativity of the matrix norm. □

To verify the validity of our theoretical result regarding the approximation quality of $W_{TV,(l)}$, we compute, for each $l$, the relative logit-effect discrepancy

$$\frac{\|W_U^\mathbb{T} W_{TV,(l)}(\theta_l - \hat{\theta}_l)\|_2}{\|W_U^\mathbb{T} W_{TV,(l)}\theta_l\|_2},$$

and examine its correlation with the final accuracy achieved by the reconstructed TV. We report the results in Figure 9a and perform a Pearson correlation test. The strongly significant negative correlation shown in Table 5 provides compelling evidence for our theory: the smaller the logit discrepancy, the better the reconstructed TV approximates the original TV and the higher the resulting accuracy. We additionally visualize the relationship using scatterplots in Figure 12 for all values collected at $l = 0,\ldots,31$ of Llama3.1-8B.

Table 4: Prompt templates and labels for different datasets.

| Dataset | Template | Label |
|---------|----------|-------|
| SST-2 | {Sentence} Sentiment: {Label} | positive / negative |
| TREC | Question: {Sentence} Type: {Label} | abbreviation / entity / description / human / location / number |
| SNLI | The question is: {Premise}? True or maybe or false? The answer is: {Hypothesis} {Label} | true / maybe / false |
| RTE | The question is: {Premise}? True or false? The answer is: {Hypothesis} {Label} | true / false |
| CB | The question is: {Premise}? True or maybe or false? The answer is: {Hypothesis} {Label} | true / maybe / false |
| Capital | {Country Name} Answer: {Label} | capital of the country |
| Capitalize | {Word} Answer: {Label} | capitalized version of the first letter in the word |
| Antonym | {Word} Answer: {Label} | antonym of the word |
| Myopic | {A question involving two choices} Answer: {Label} | the myopic choice |

| Pearson correlation coefficient | p-value |
|---------------------------------|---------|
| −0.4914 | 0.0042 |

Table 5: Correlation strength between accuracy of reconstructed TV and the relative estimated logit effect difference

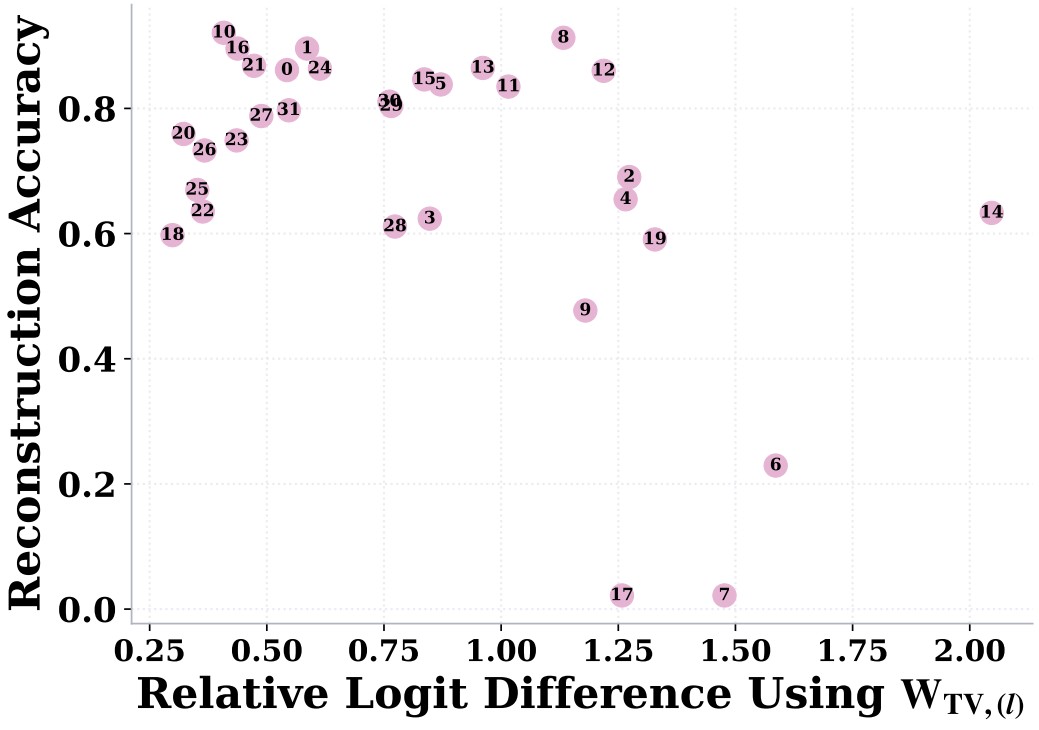

Figure 12: Scatterplot of the reconstruction TV's accuracy against their estimated logit effect difference compared to the original TV. The number on the dots represent the layer index of Llama3.1-8B

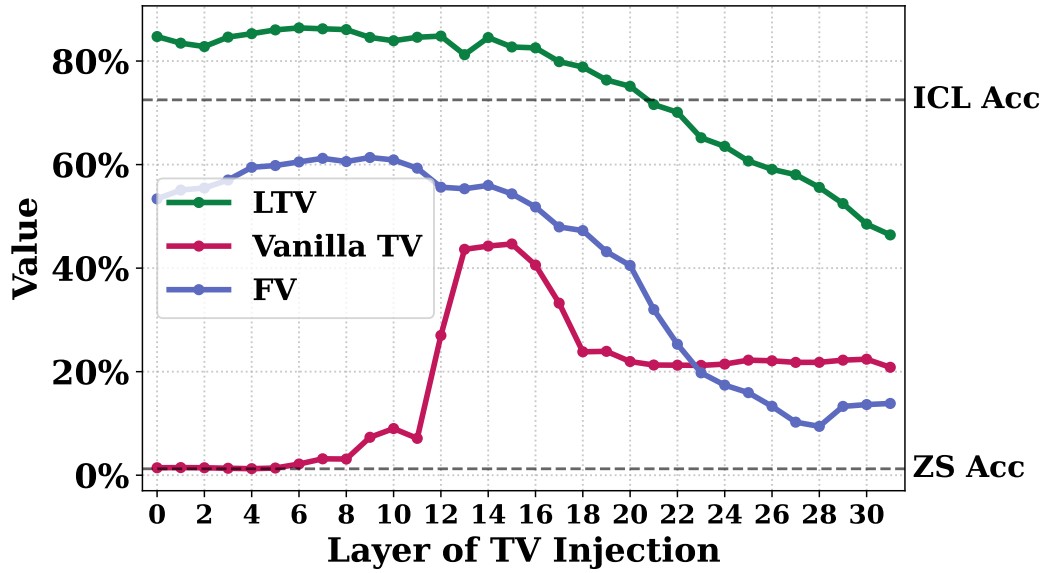

Figure 13: Layer sweeping results of injecting the Vanilla TV, FV, and our LTV to the last token hidden states on Llama2-7B.

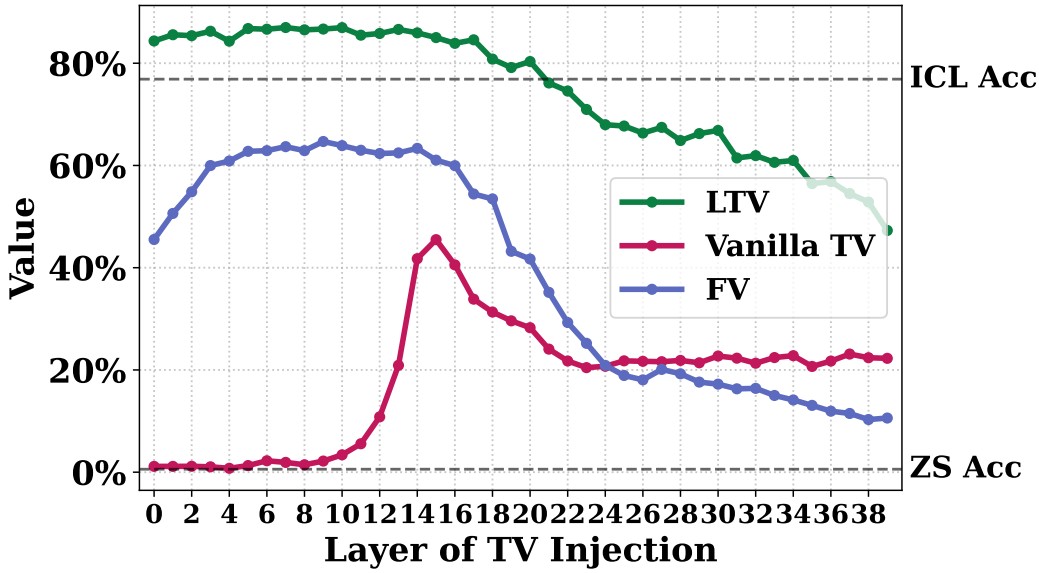

Figure 14: Layer sweeping results of injecting the Vanilla TV, FV, and our LTV to the last token hidden states on Llama2-13B.

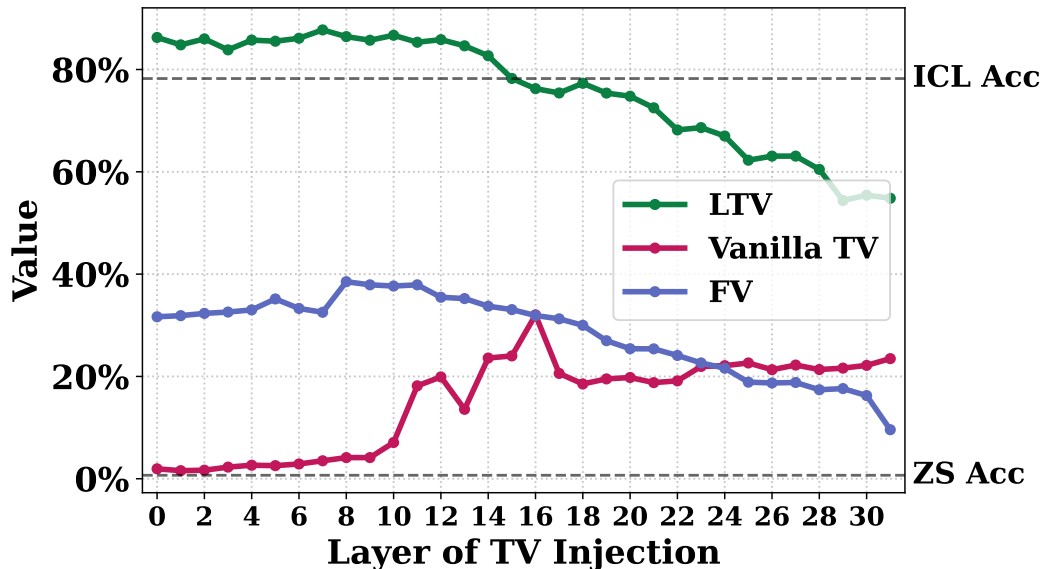

Figure 15: Layer sweeping results of injecting the Vanilla TV, FV, and our LTV to the last token hidden states on Llama3-8B.

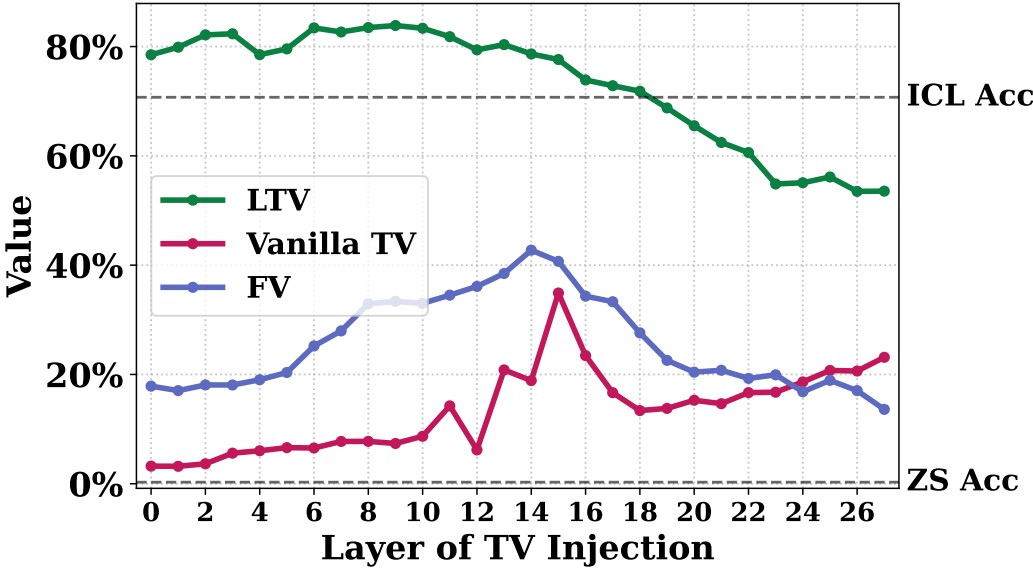

Figure 16: Layer sweeping results of injecting the Vanilla TV, FV, and our LTV to the last token hidden states on Llama3.2-3B.

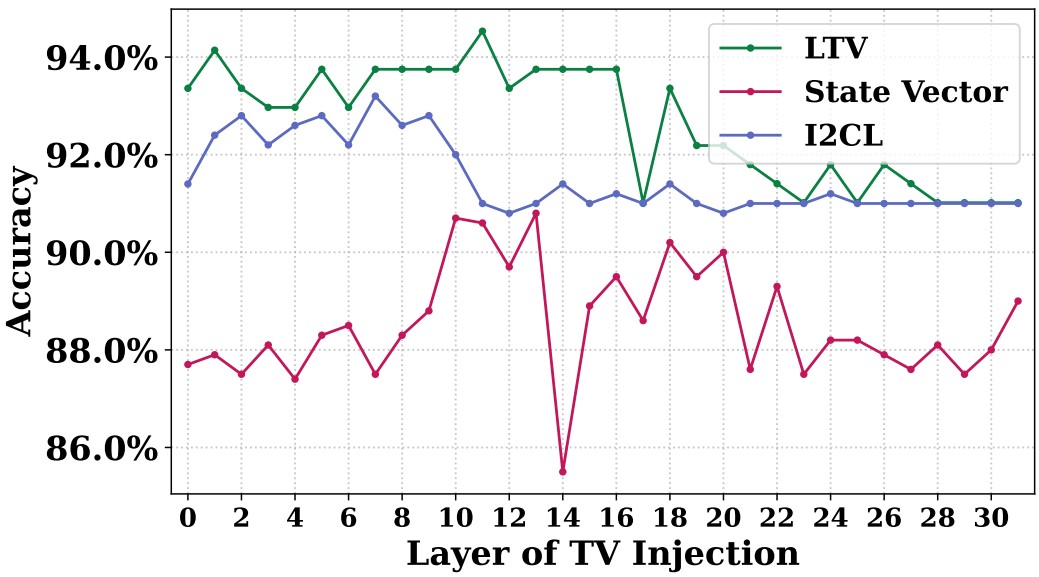

Figure 17: Comparison of LTV, State Vector, and I2CL on SST-2 when injected into the last-token hidden states at each individual layer of Llama2-7B.

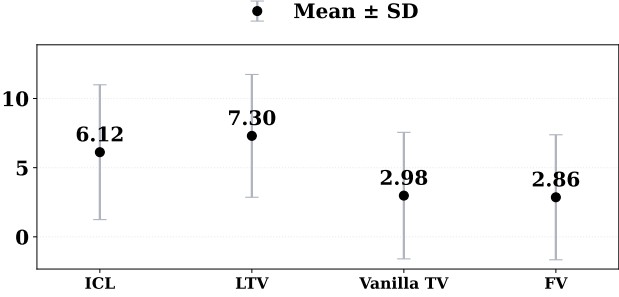

Figure 18: Myopic dataset: LTV vs. Vanilla TV and FV on Llama2-7B.

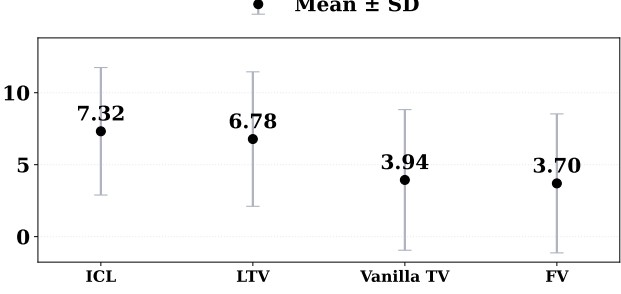

Figure 19: Myopic dataset: LTV vs. Vanilla TV and FV on Llama2-13B.

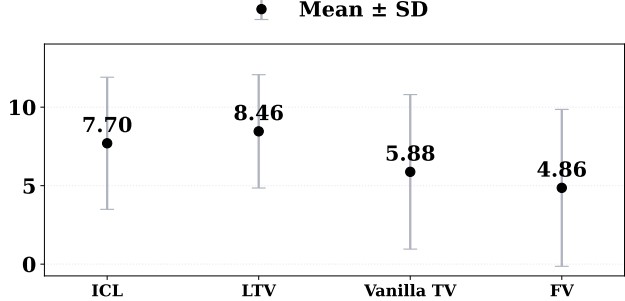

Figure 20: Myopic dataset: LTV vs. Vanilla TV and FV on Llama3-8B.

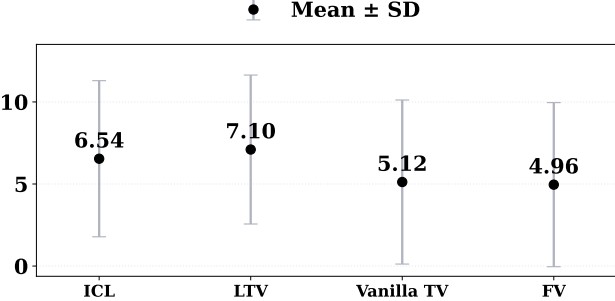

Figure 21: Myopic dataset: LTV vs. Vanilla TV and FV on Llama3.2-3B.

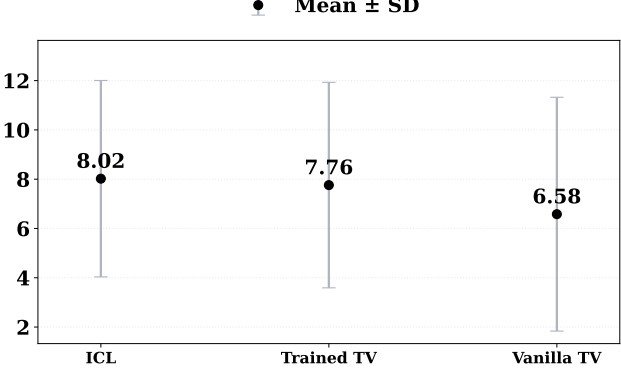

Figure 22: Myopic dataset: LTV vs. Vanilla TV on Llama3-70B.

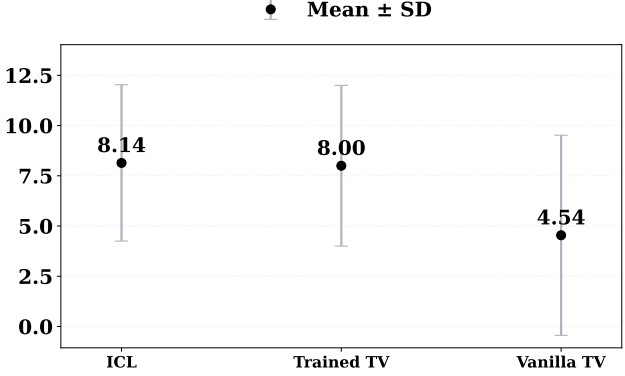

Figure 23: Myopic dataset: LTV vs. Vanilla TV on Qwen2.5-32B.

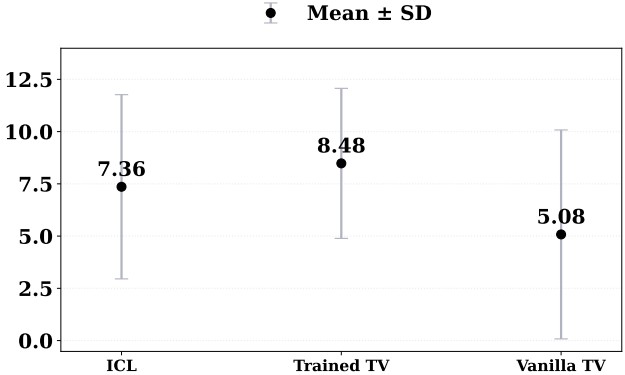

Figure 24: Myopic dataset: LTV vs. Vanilla TV on Yi-34B.

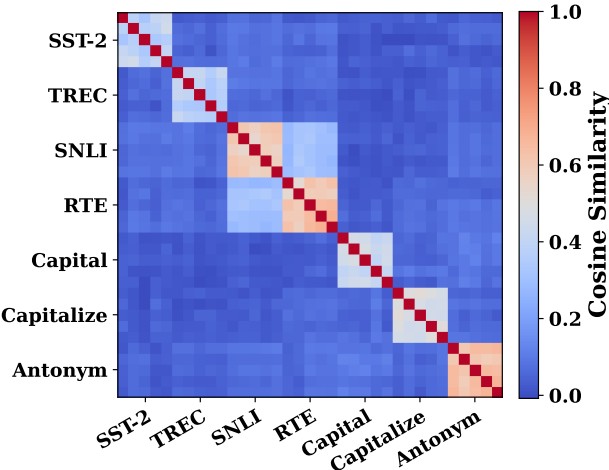

Figure 25: Cosine-similarity heatmap of LTVs trained for seven tasks on Llama3-8B.

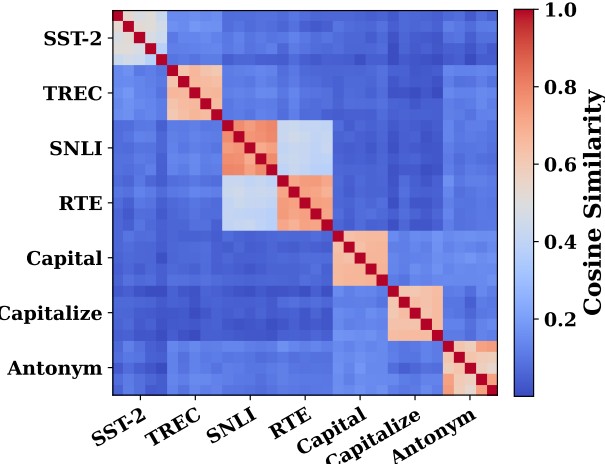

Figure 26: Cosine-similarity heatmap of LTVs trained for seven tasks on Llama3.2-3B.

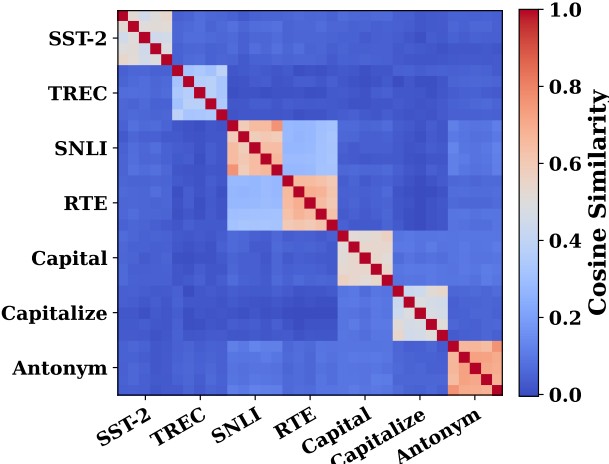

Figure 27: Cosine-similarity heatmap of LTVs trained for seven tasks on Llama3-70B.

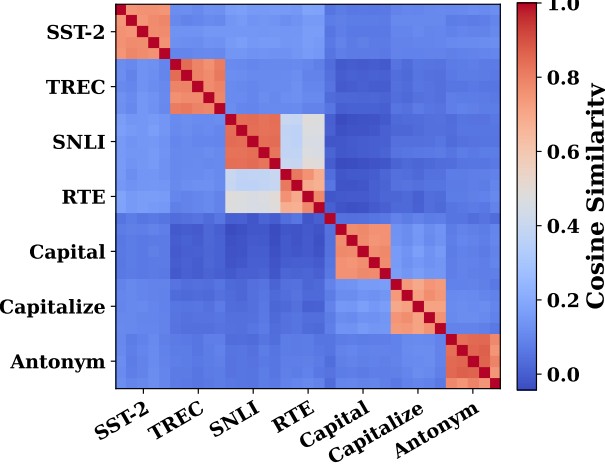

Figure 28: Cosine-similarity heatmap of LTVs trained for seven tasks on Llama2-7B.

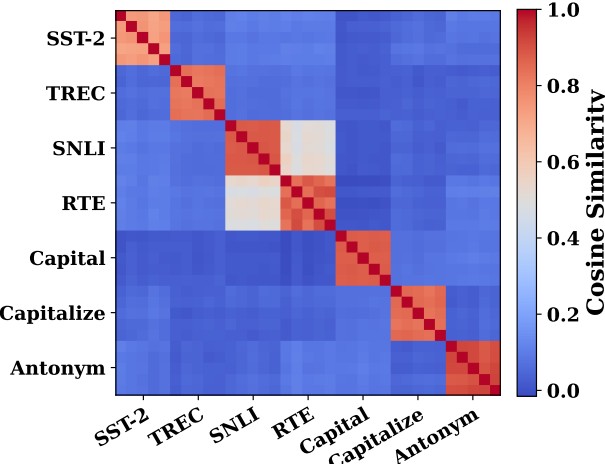

Figure 29: Cosine-similarity heatmap of LTVs trained for seven tasks on Llama2-13B.

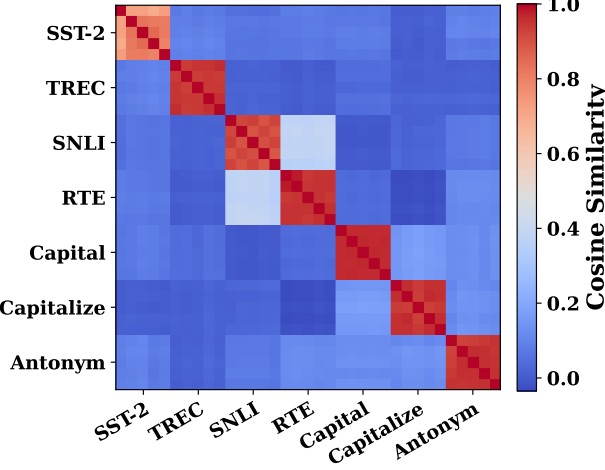

Figure 30: Cosine-similarity heatmap of LTVs trained for seven tasks on Qwen2.5-32B.

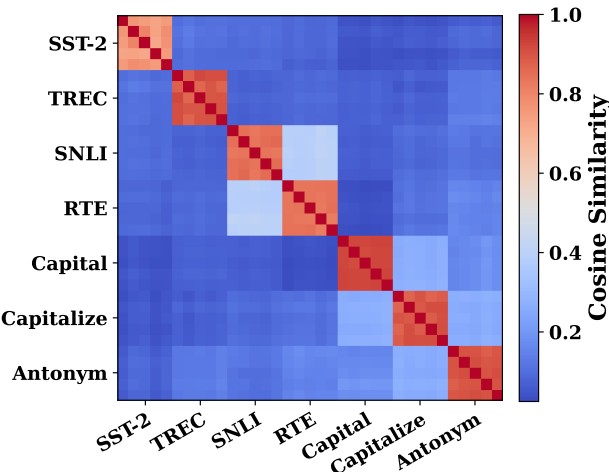

Figure 31: Cosine-similarity heatmap of LTVs trained for seven tasks on Yi-34B.

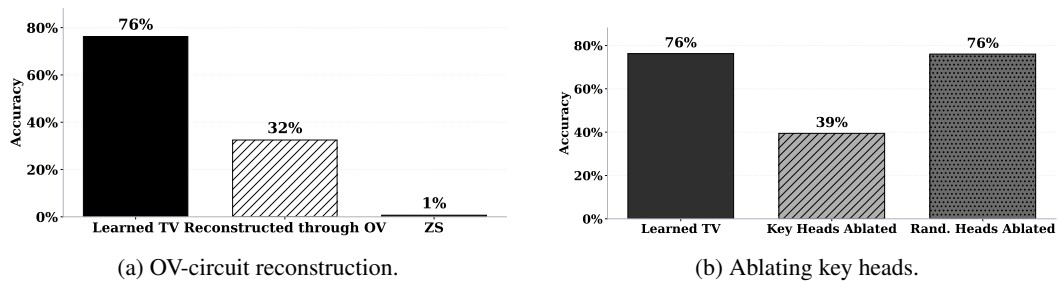

(a) OV-circuit reconstruction.

(b) Ablating key heads.

Figure 32: Attention heads and TV on Llama3-8B: OV-circuit reconstruction (left) and ablation of key heads (right).

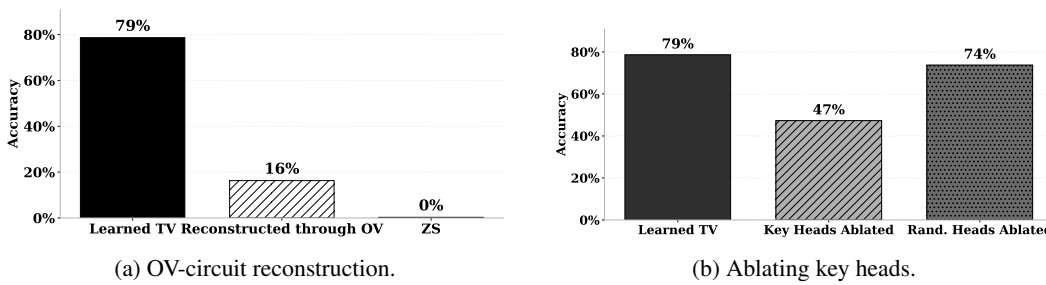

(a) OV-circuit reconstruction.

(b) Ablating key heads.

Figure 33: Attention heads and TV on Llama3.2-3B: OV-circuit reconstruction (left) and ablation of key heads (right).

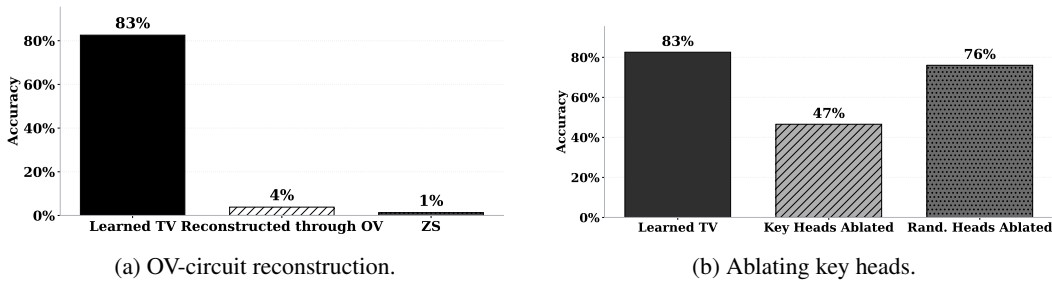

(a) OV-circuit reconstruction.

(b) Ablating key heads.

Figure 34: Attention heads and TV on Llama2-7B: OV-circuit reconstruction (left) and ablation of key heads (right).

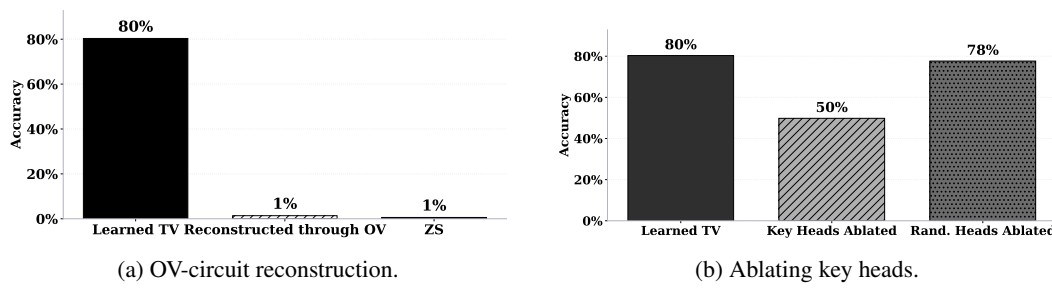

(a) OV-circuit reconstruction.

(b) Ablating key heads.

Figure 35: Attention heads and TV on Llama2-13B: OV-circuit reconstruction (left) and ablation of key heads (right).

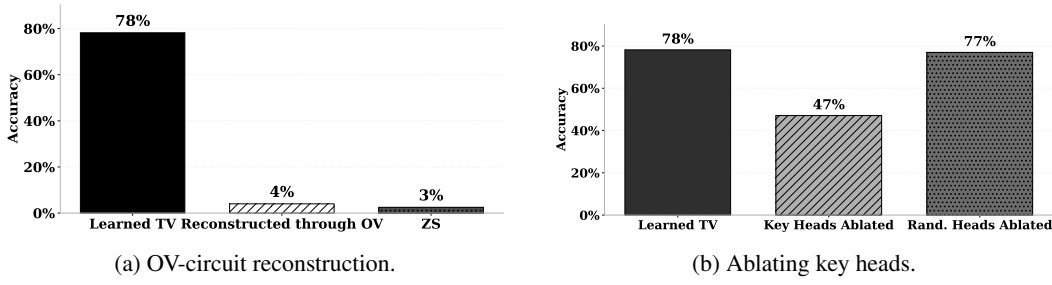

(a) OV-circuit reconstruction.

(b) Ablating key heads.

Figure 36: Attention heads and TV on Llama3-70B: OV-circuit reconstruction (left) and ablation of key heads (right).

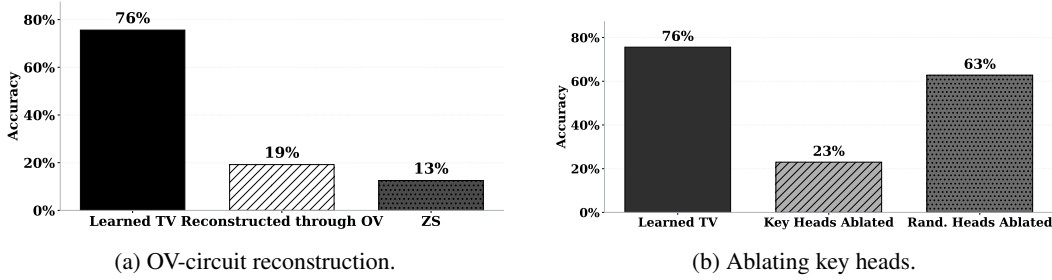

(a) OV-circuit reconstruction.

(b) Ablating key heads.

Figure 37: Attention heads and TV on Qwen2.5-32B: OV-circuit reconstruction (left) and ablation of key heads (right).

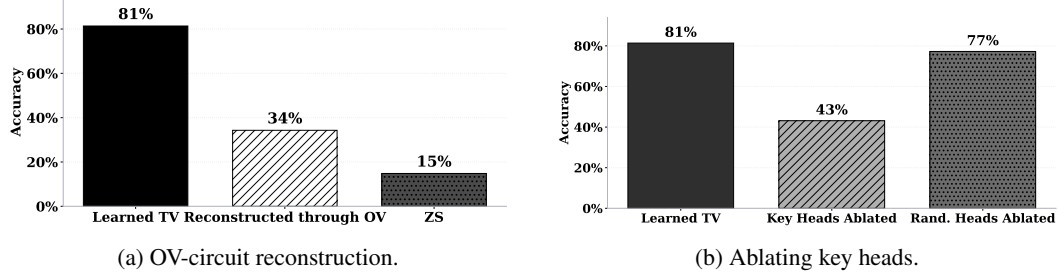

(a) OV-circuit reconstruction.

(b) Ablating key heads.

Figure 38: Attention heads and TV on Yi-34B: OV-circuit reconstruction (left) and ablation of key heads (right).

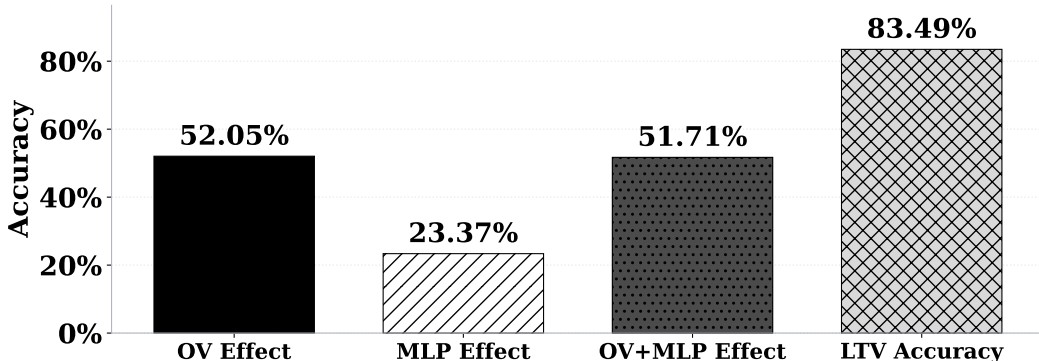

Figure 39: Effects of the MLP-based construction and MLP&OV-based reconstruction compared to the effect of OV-based reconstruction of TV effect.

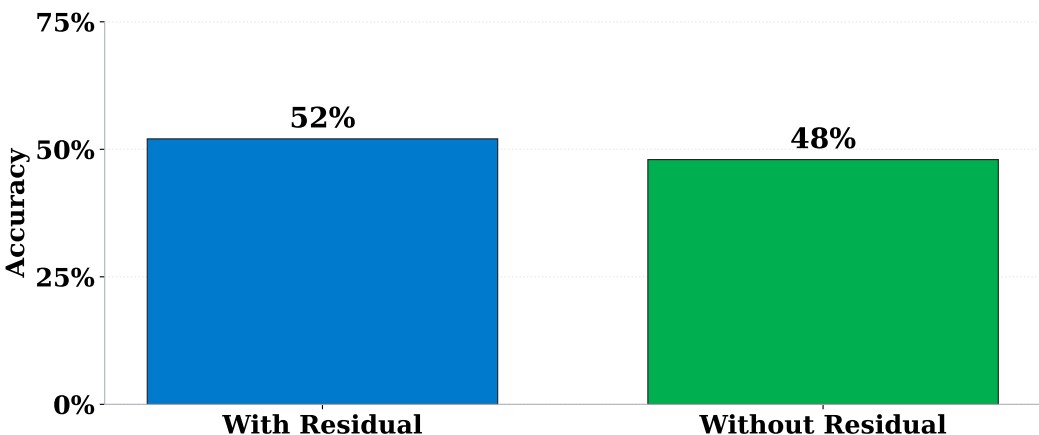

Figure 40: Effects of the OV circuit reconstruction with or without the TV added to the final layer: Llama 3.1-8B.

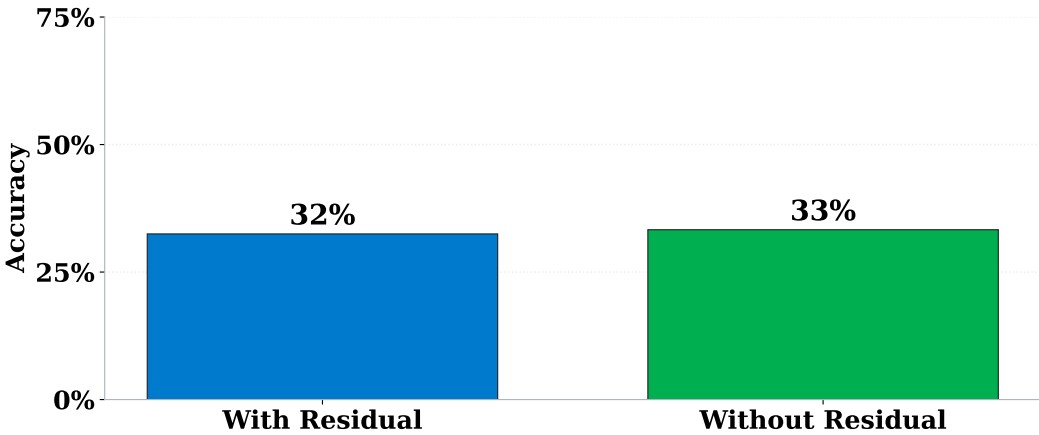

Figure 41: Effects of the OV circuit reconstruction with or without the TV added to the final layer: Llama 3-8B.

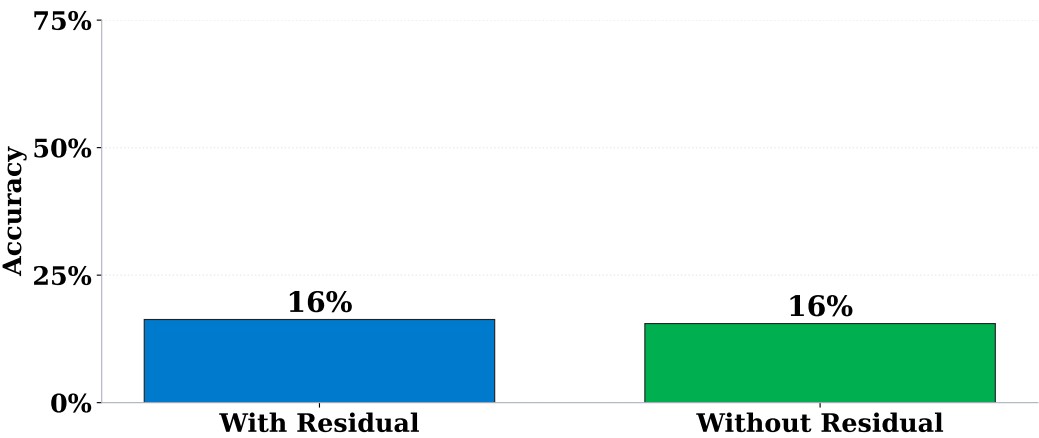

Figure 42: Effects of the OV circuit reconstruction with or without the TV added to the final layer: Llama 3.2-3B.

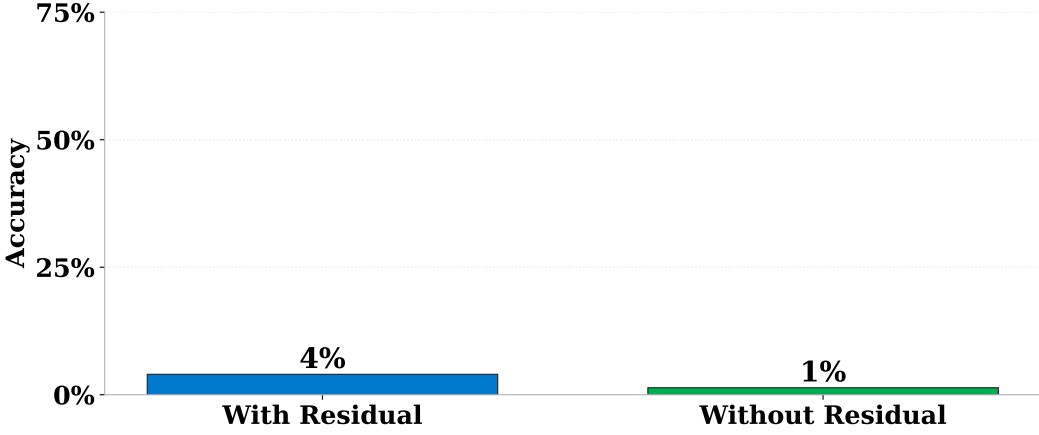

Figure 43: Effects of the OV circuit reconstruction with or without the TV added to the final layer: Llama 3-70B.

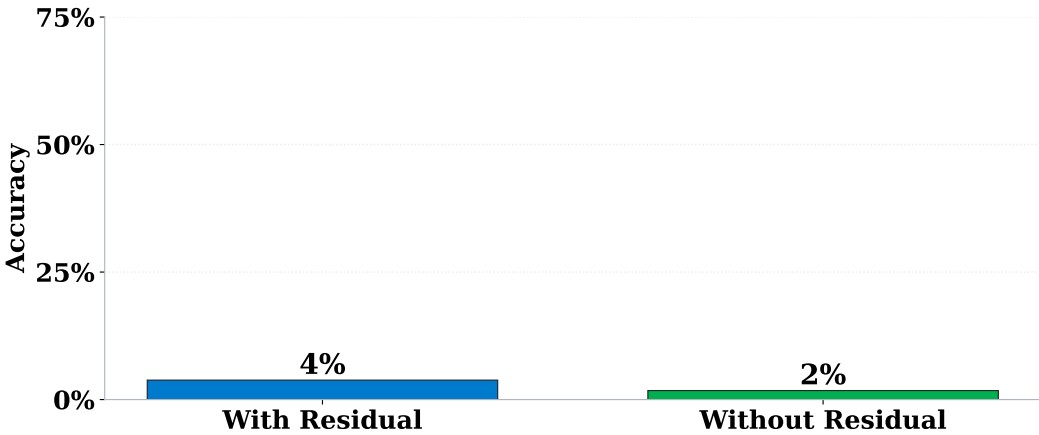

Figure 44: Effects of the OV circuit reconstruction with or without the TV added to the final layer: Llama 2-7B.

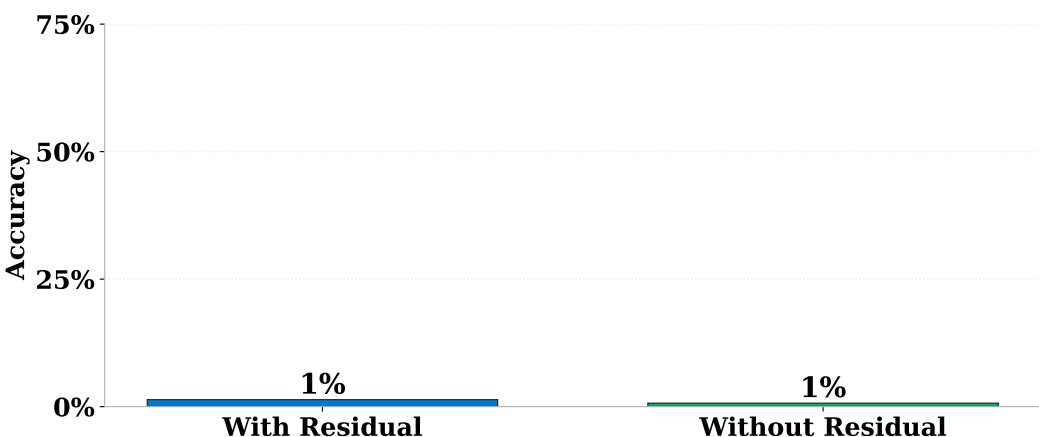

Figure 45: Effects of the OV circuit reconstruction with or without the TV added to the final layer: Llama 2-13B.

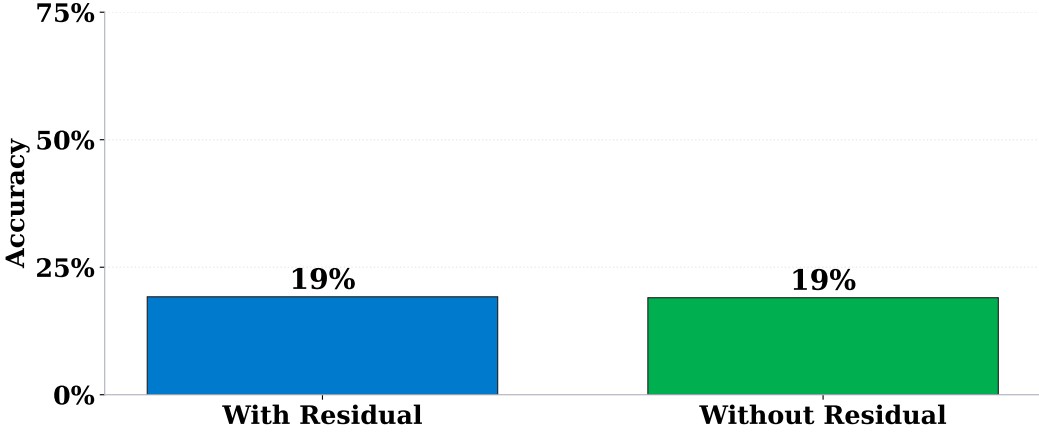

Figure 46: Effects of the OV circuit reconstruction with or without the TV added to the final layer: Qwen2.5-32B.

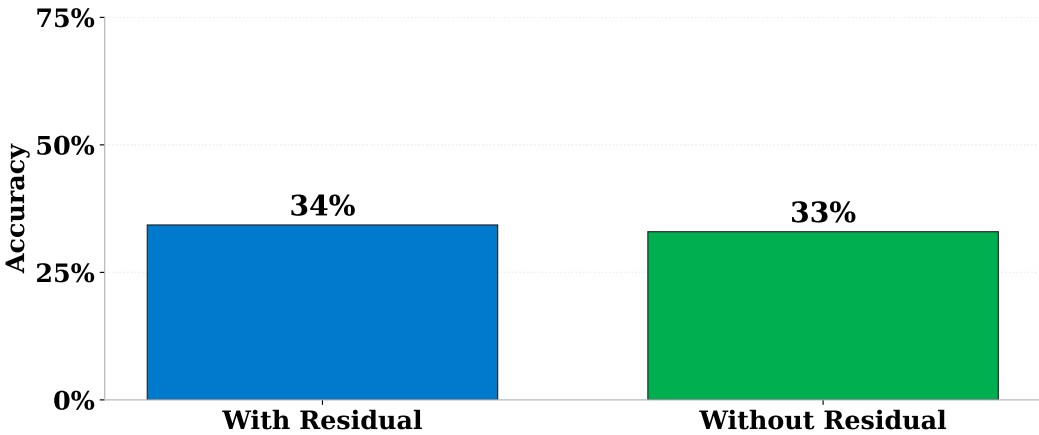

Figure 47: Effects of the OV circuit reconstruction with or without the TV added to the final layer: Yi-34B.

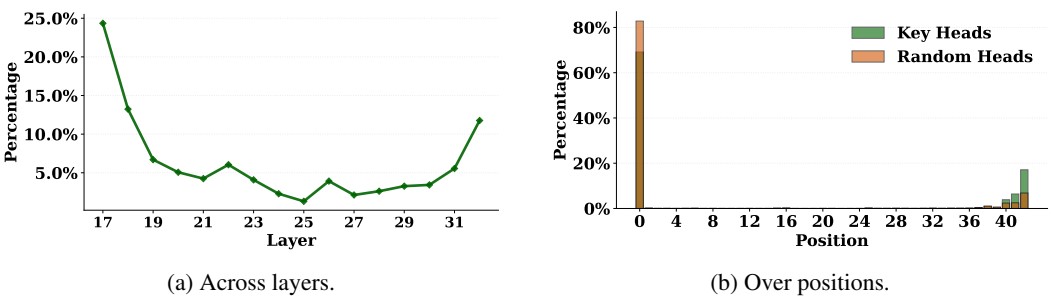

(a) Across layers.

(b) Over positions.

Figure 48: Key attention heads on Llama3-8B: distribution across layers (left) and attention over token positions (right).

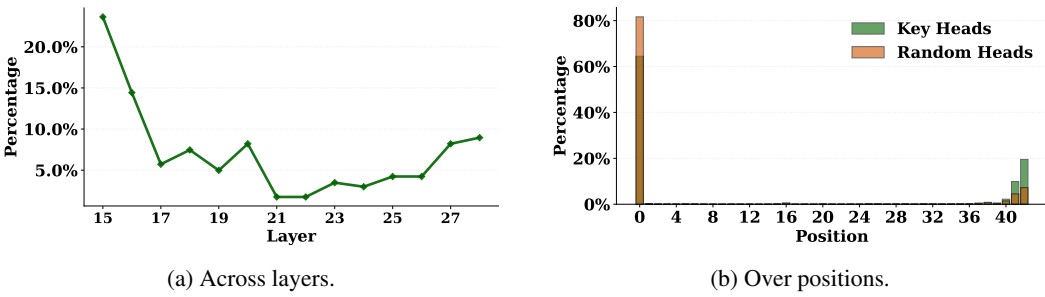

(a) Across layers.

(b) Over positions.

Figure 49: Key attention heads on Llama3.2-3B: distribution across layers (left) and attention over token positions (right).

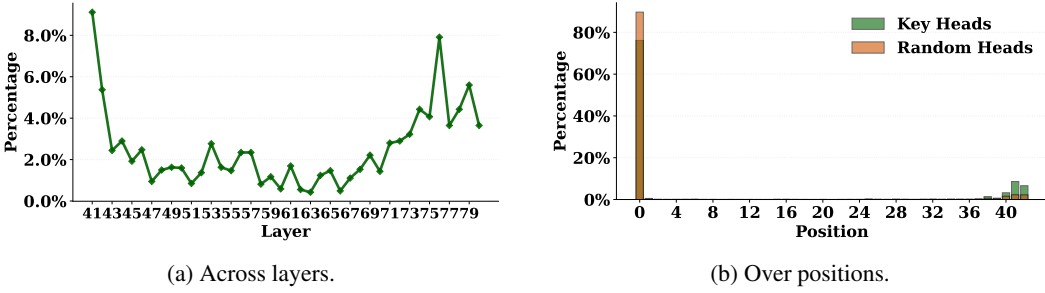

(a) Across layers.

(b) Over positions.

Figure 50: Key attention heads on Llama3-70B: distribution across layers (left) and attention over token positions (right).

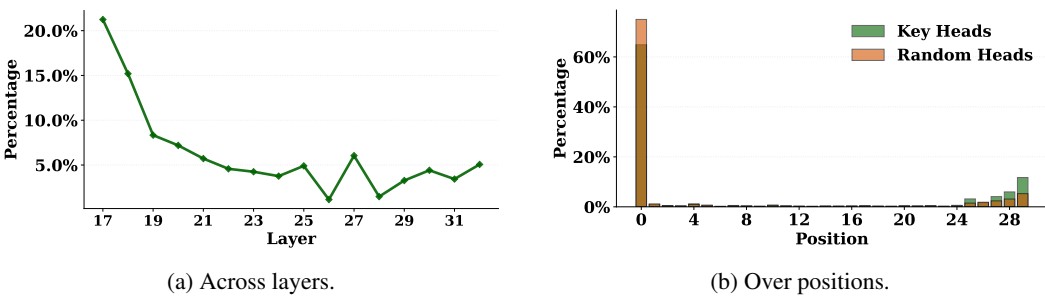

(a) Across layers.

(b) Over positions.

Figure 51: Key attention heads on Llama2-7B: distribution across layers (left) and attention over token positions (right).

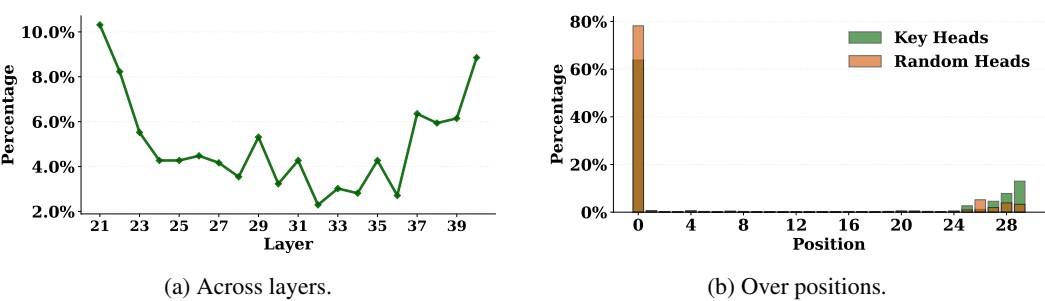

(a) Across layers.

(b) Over positions.

Figure 52: Key attention heads on Llama2-13B: distribution across layers (left) and attention over token positions (right).

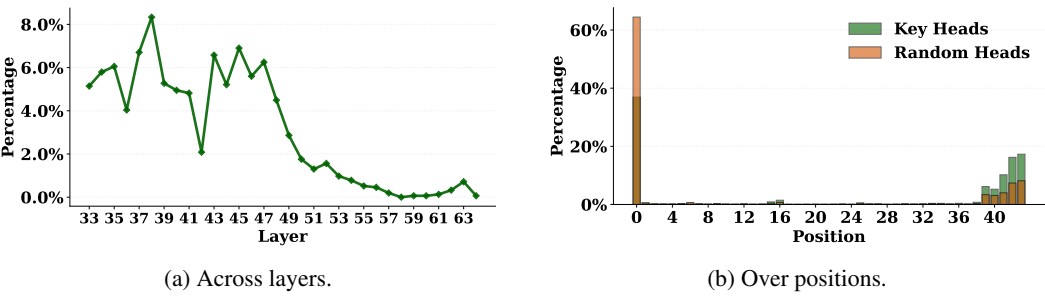

(a) Across layers.

(b) Over positions.

Figure 53: Key attention heads on Qwen2.5-32B: distribution across layers (left) and attention over token positions (right).

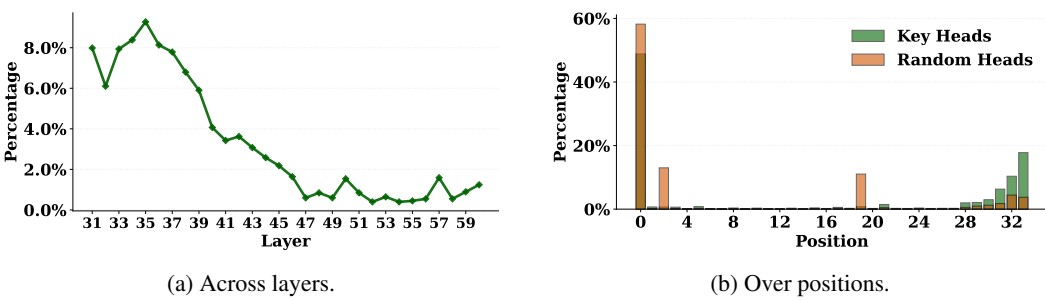

(a) Across layers.
(b) Over positions.

Figure 54: Key attention heads on Yi-34B: distribution across layers (left) and attention over token positions (right).

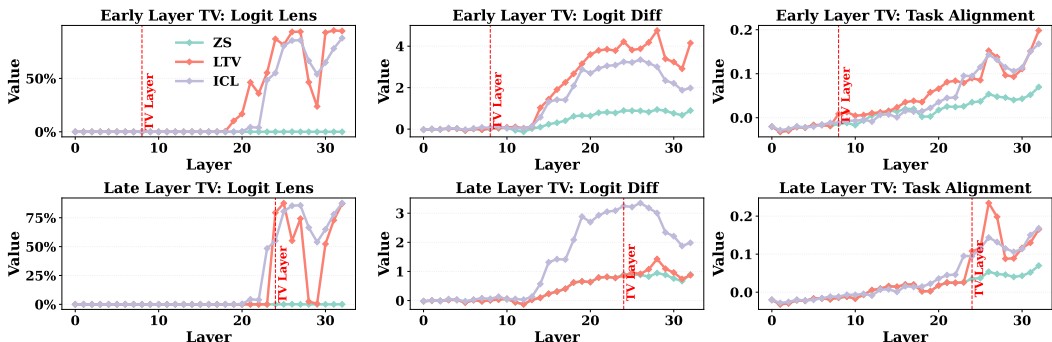

Figure 55: Metrics across layers on Llama3-8B when the TV is injected into the hidden state at an early vs. late layer.

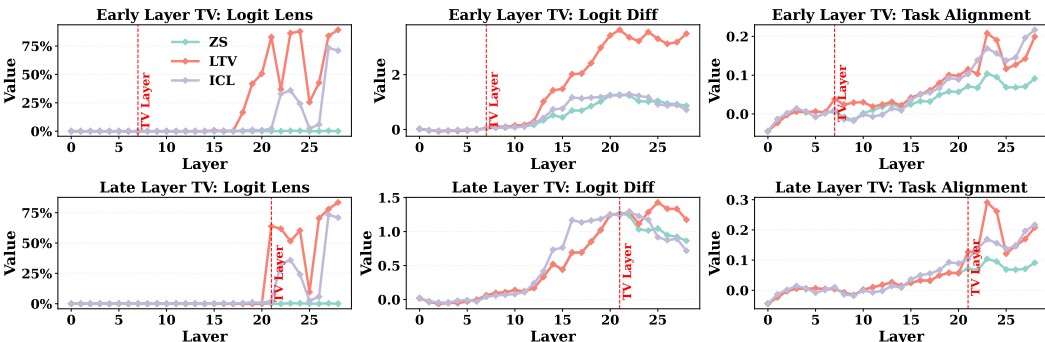

Figure 56: Metrics across layers on Llama3.2-3B when the TV is injected into the hidden state at an early vs. late layer.

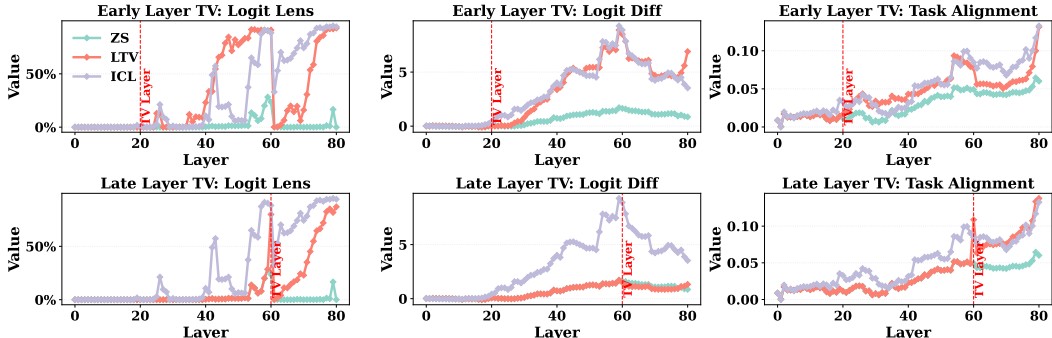

Figure 57: Metrics across layers on Llama3-70B when the TV is injected into the hidden state at an early vs. late layer.

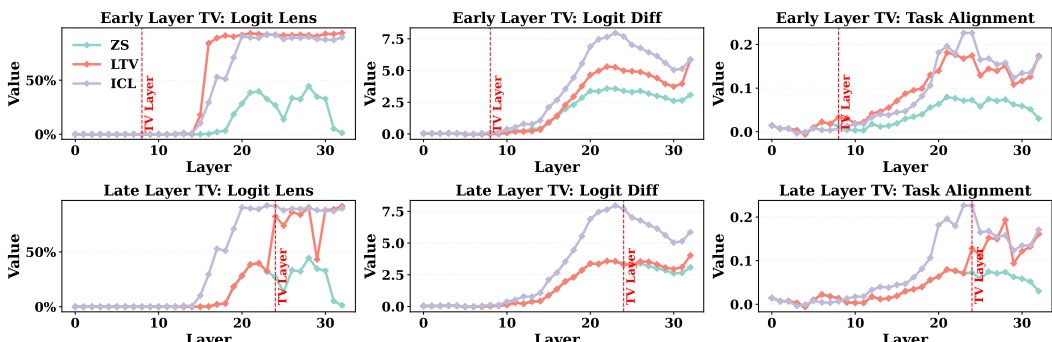

Figure 58: Metrics across layers on Llama2-7B when the TV is injected into the hidden state at an early vs. late layer.

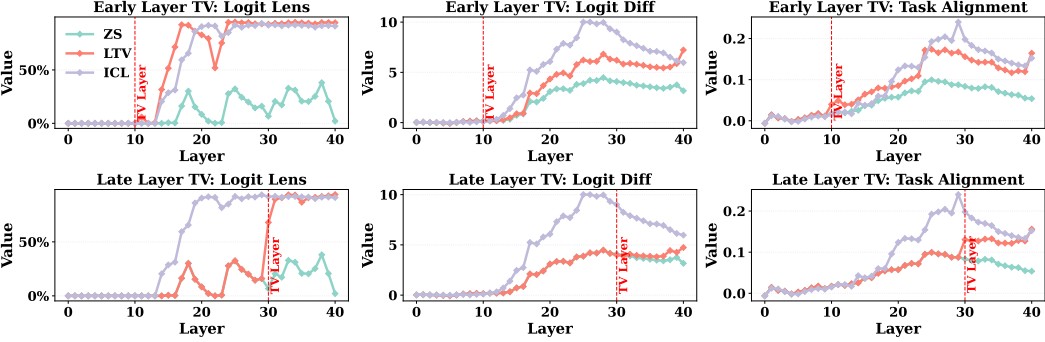

Figure 59: Metrics across layers on Llama2-13B when the TV is injected into the hidden state at an early vs. late layer.

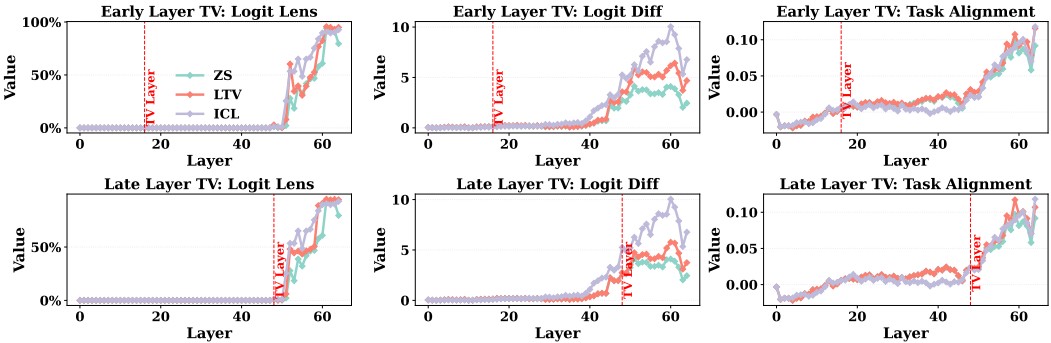

Figure 60: Metrics across layers on Qwen2.5-32B when the TV is injected into the hidden state at an early vs. late layer.

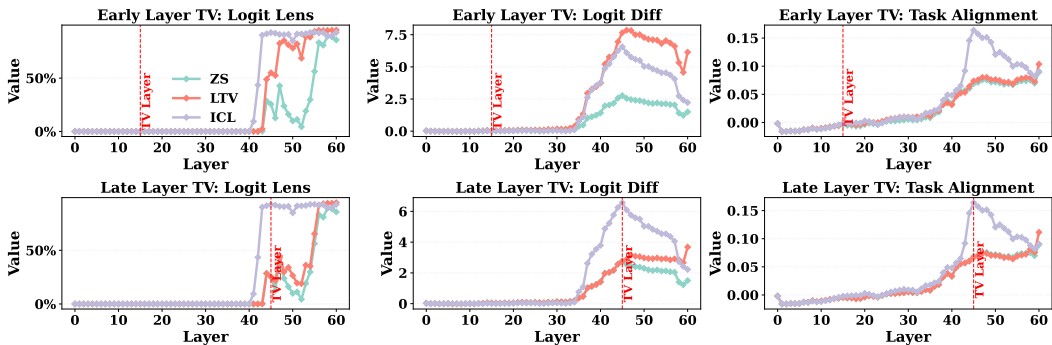

Figure 61: Metrics across layers on Yi-34B when the TV is injected into the hidden state at an early vs. late layer.

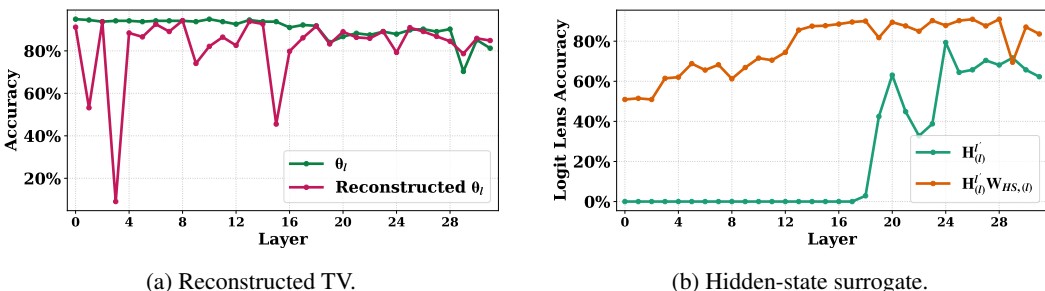

(a) Reconstructed TV.

(b) Hidden-state surrogate.

Figure 62: Linear hypothesis on Llama3-8B: linearly reconstructed TV (left) and linear surrogate for hidden-state updates (right).

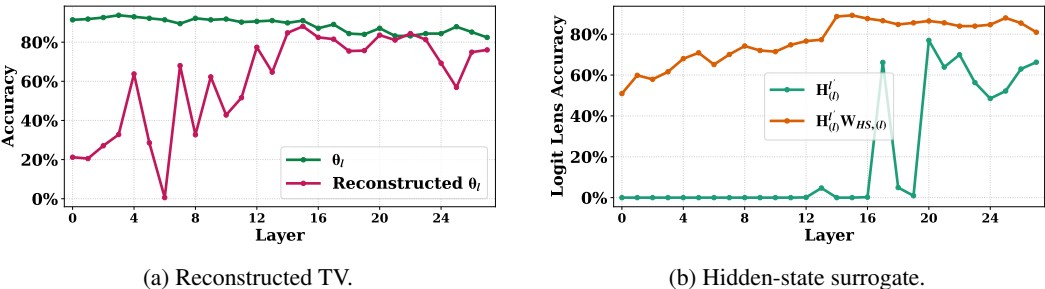

(a) Reconstructed TV.

(b) Hidden-state surrogate.

Figure 63: Linear hypothesis on Llama3.2-3B: linearly reconstructed TV (left) and linear surrogate for hidden-state updates (right).

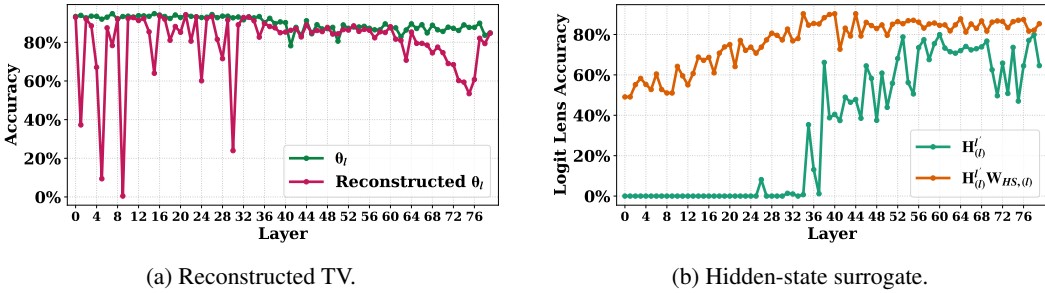

(a) Reconstructed TV.

(b) Hidden-state surrogate.

Figure 64: Linear hypothesis on Llama3-70B: linearly reconstructed TV (left) and linear surrogate for hidden-state updates (right).

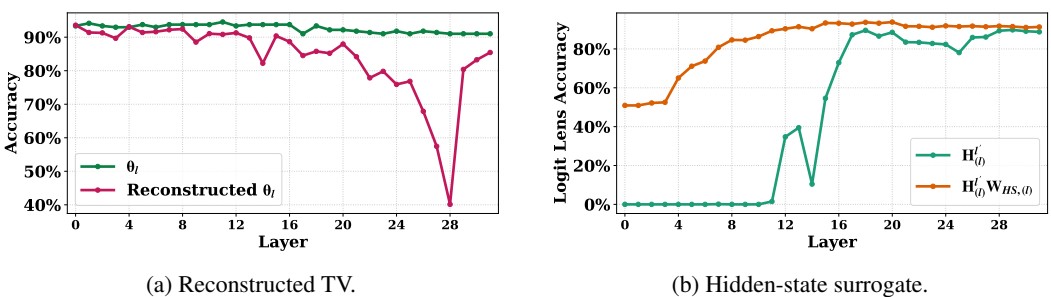

(a) Reconstructed TV.

(b) Hidden-state surrogate.

Figure 65: Linear hypothesis on Llama2-7B: linearly reconstructed TV (left) and linear surrogate for hidden-state updates (right).

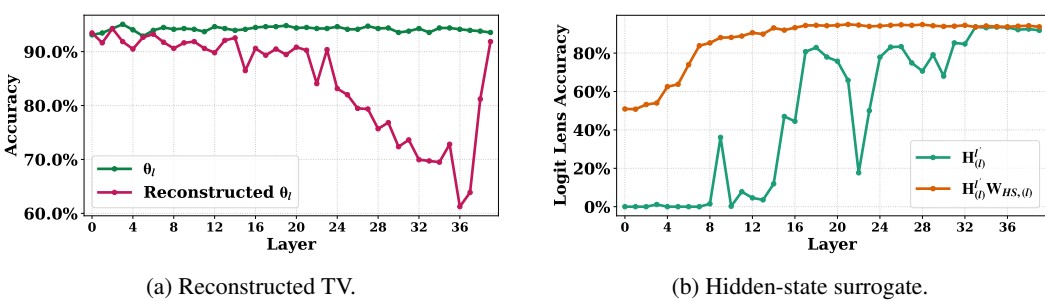

(a) Reconstructed TV.

(b) Hidden-state surrogate.

Figure 66: Linear hypothesis on Llama2-13B: linearly reconstructed TV (left) and linear surrogate for hidden-state updates (right).

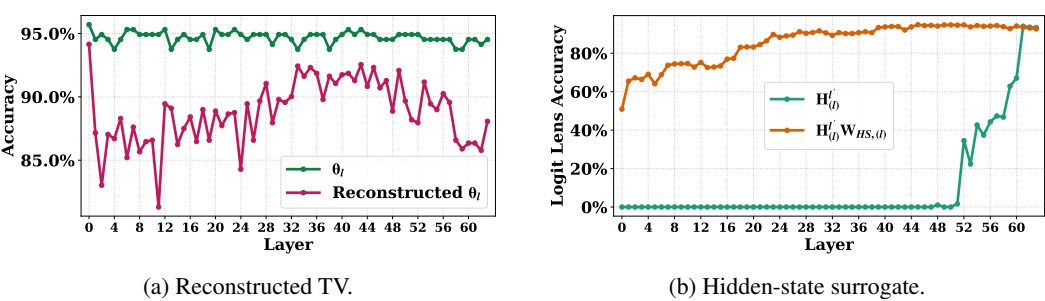

(a) Reconstructed TV.

(b) Hidden-state surrogate.

Figure 67: Linear hypothesis on Qwen2.5-32B: linearly reconstructed TV (left) and linear surrogate for hidden-state updates (right).

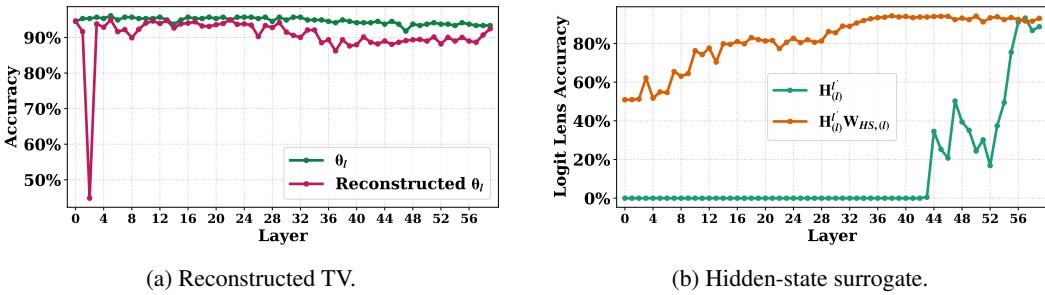

(a) Reconstructed TV.

(b) Hidden-state surrogate.

Figure 68: Linear hypothesis on Yi-34B: linearly reconstructed TV (left) and linear surrogate for hidden-state updates (right).

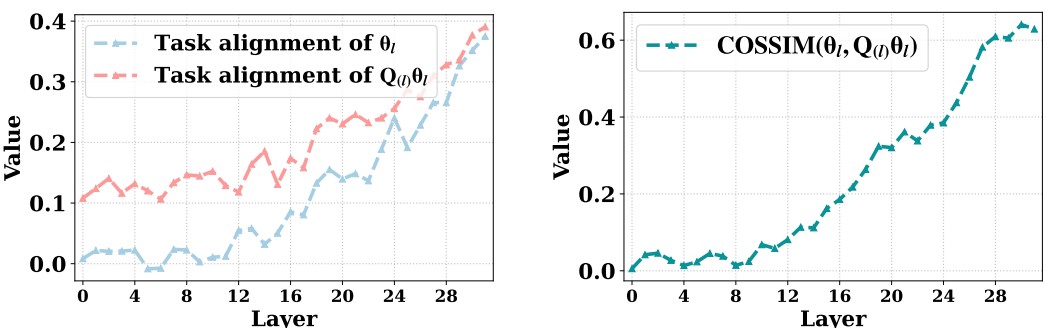

Figure 69: Rotation analysis on Llama3-8B: applying the fitted rotation $\mathbf{Q}_{(l)}$ to the TV increases task alignment (left); rotation strength vs. layer depth (right).

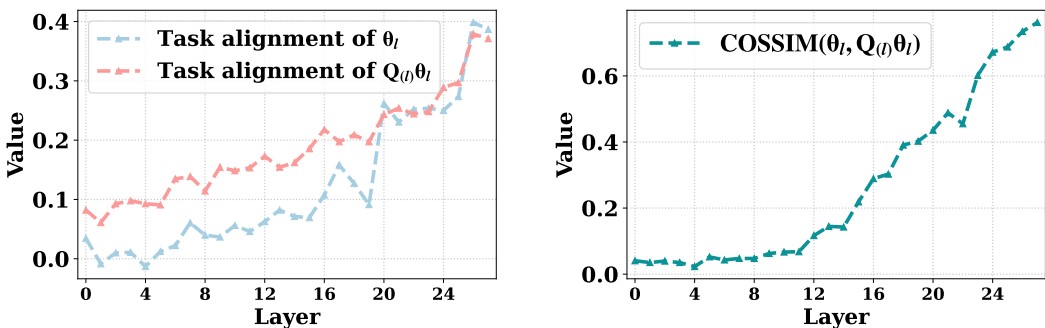

Figure 70: Rotation analysis on Llama3.2-3B: applying the fitted rotation $\mathbf{Q}_{(l)}$ to the TV increases task alignment (left); rotation strength vs. layer depth (right).

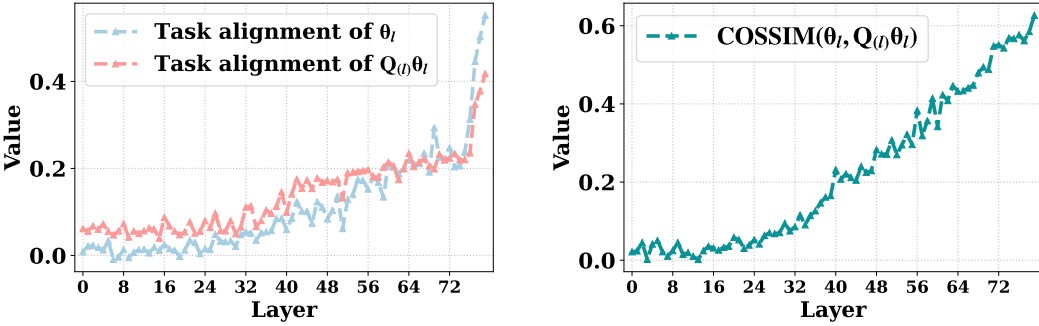

Figure 71: Rotation analysis on Llama3-70B: applying the fitted rotation $\mathbf{Q}_{(l)}$ to the TV increases task alignment (left); rotation strength vs. layer depth (right).

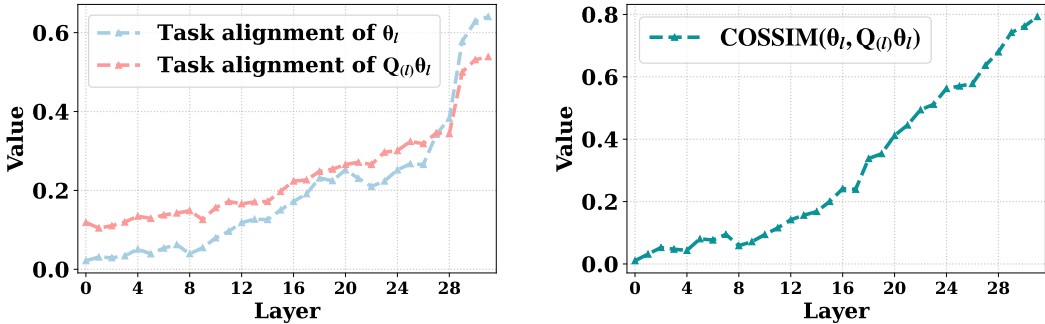

Figure 72: Rotation analysis on Llama2-7B: applying the fitted rotation $\mathbf{Q}_{(l)}$ to the TV increases task alignment (left); rotation strength vs. layer depth (right).

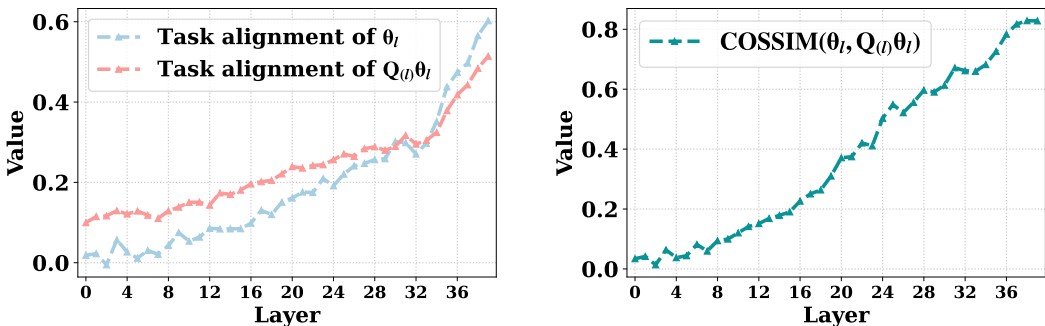

Figure 73: Rotation analysis on Llama2-13B: applying the fitted rotation $\mathbf{Q}_{(l)}$ to the TV increases task alignment (left); rotation strength vs. layer depth (right).

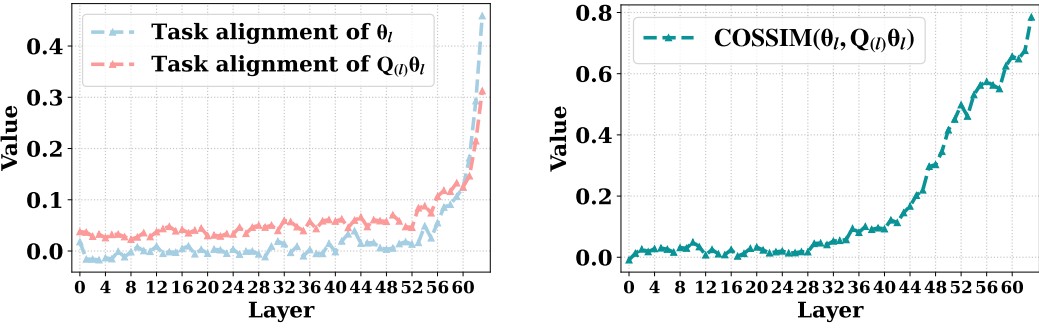

Figure 74: Rotation analysis on Qwen2.5-32B: applying the fitted rotation $\mathbf{Q}_{(l)}$ to the TV increases task alignment (left); rotation strength vs. layer depth (right).

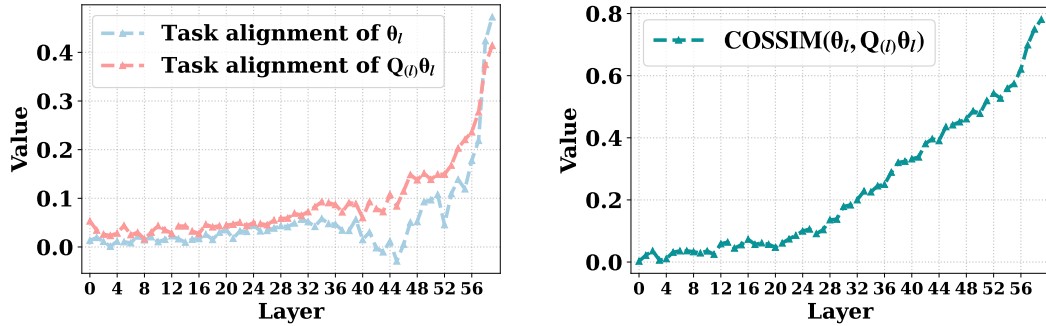

Figure 75: Rotation analysis on Yi-34B: applying the fitted rotation $\mathbf{Q}_{(l)}$ to the TV increases task alignment (left); rotation strength vs. layer depth (right).

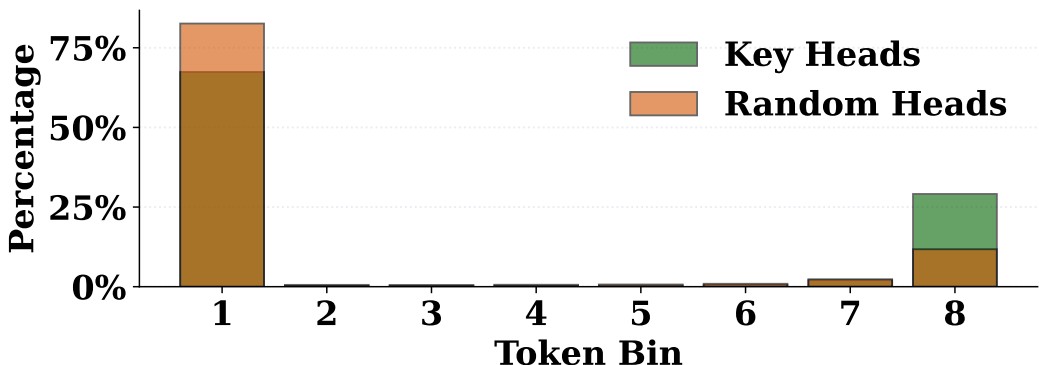

Figure 76: Average attention distribution of Llama3.1-8B on SST-2: proportions of attention weights assigned to 8 tokens intervals each comprising $\frac{1}{8}$ of all tokens.

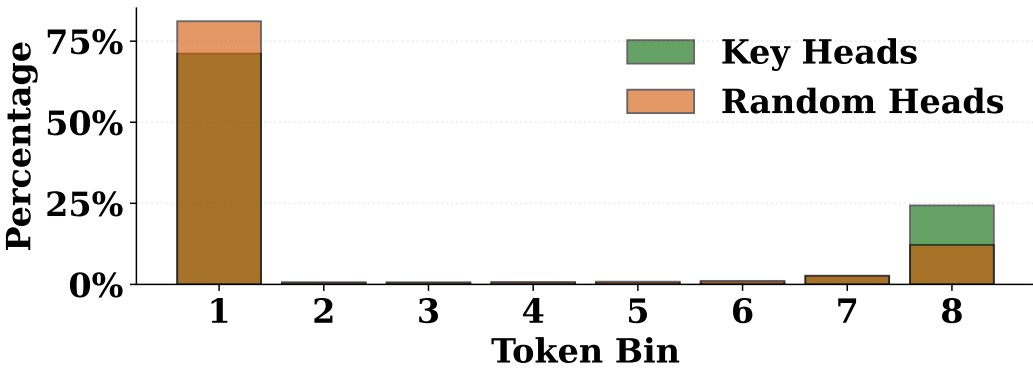

Figure 77: Average attention distribution of Llama3-8B on SST-2: proportions of attention weights assigned to 8 tokens intervals each comprising $\frac{1}{8}$ of all tokens.

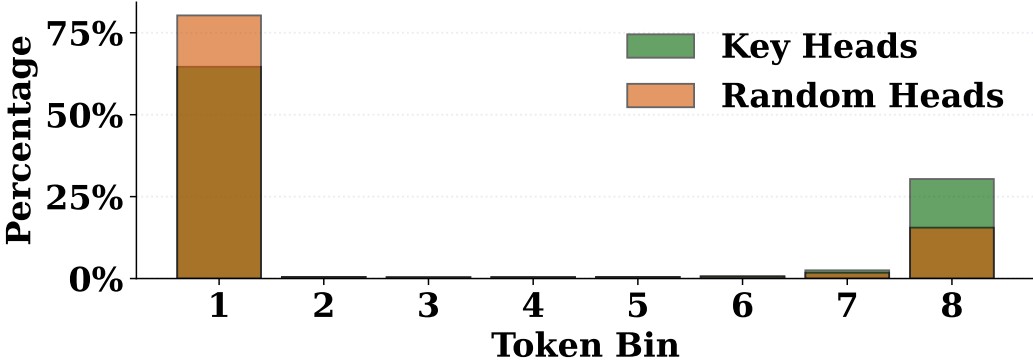

Figure 78: Average attention distribution of Llama3.2-3B on SST-2: proportions of attention weights assigned to 8 tokens intervals each comprising $\frac{1}{8}$ of all tokens.

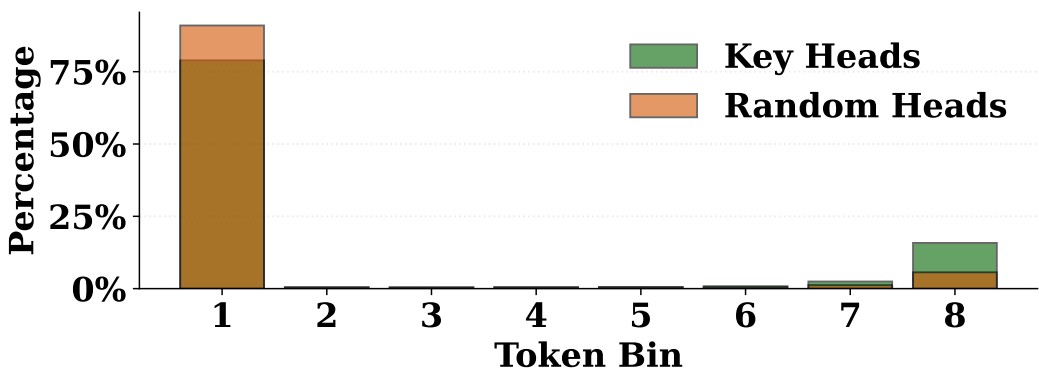

Figure 79: Average attention distribution of Llama3-70B on SST-2: proportions of attention weights assigned to 8 tokens intervals each comprising $\frac{1}{8}$ of all tokens.

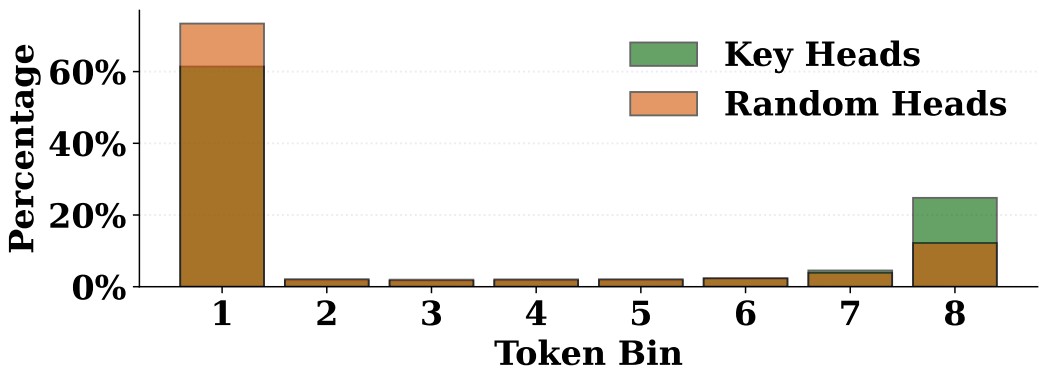

Figure 80: Average attention distribution of Llama2-7B on SST-2: proportions of attention weights assigned to 8 tokens intervals each comprising $\frac{1}{8}$ of all tokens.

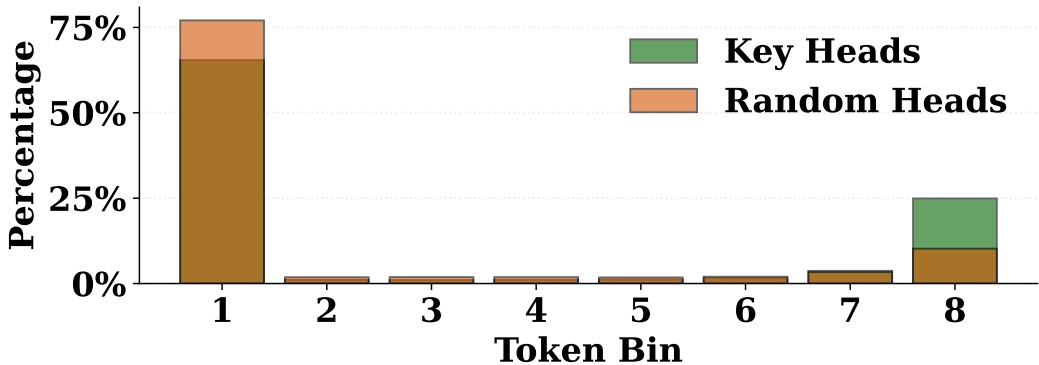

Figure 81: Average attention distribution of Llama2-13B on SST-2: proportions of attention weights assigned to 8 tokens intervals each comprising $\frac{1}{8}$ of all tokens.

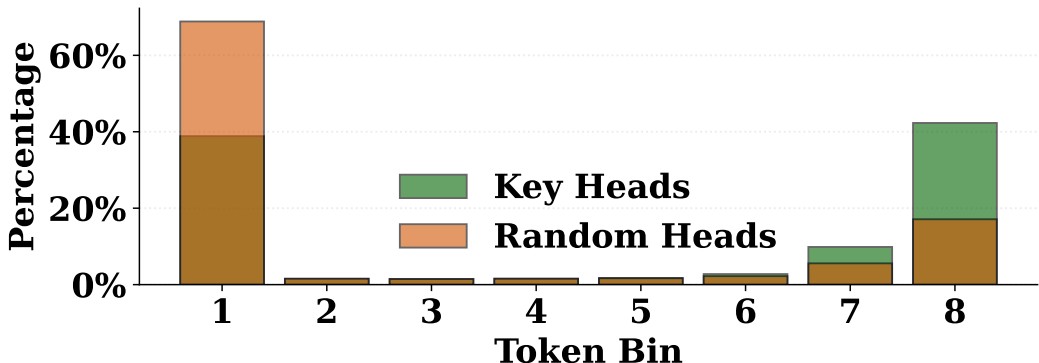

Figure 82: Average attention distribution of Qwen2.5-32B on SST-2: proportions of attention weights assigned to 8 tokens intervals each comprising $\frac{1}{8}$ of all tokens.

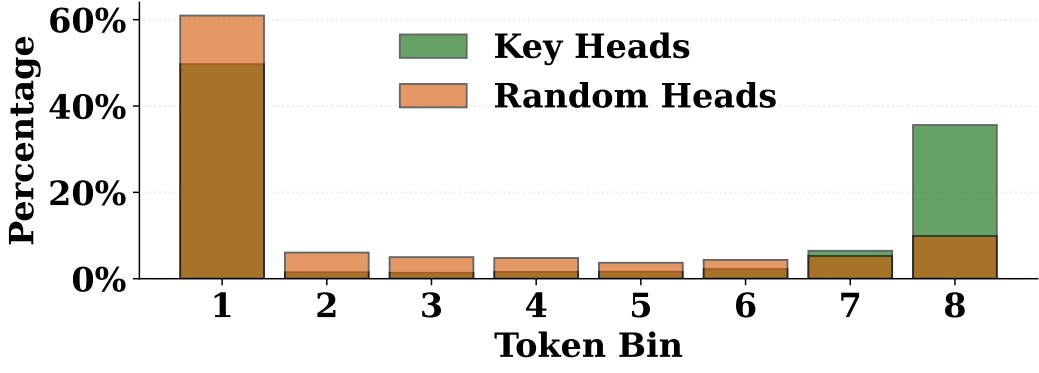

Figure 83: Average attention distribution of Yi-34B on SST-2: proportions of attention weights assigned to 8 tokens intervals each comprising $\frac{1}{8}$ of all tokens.

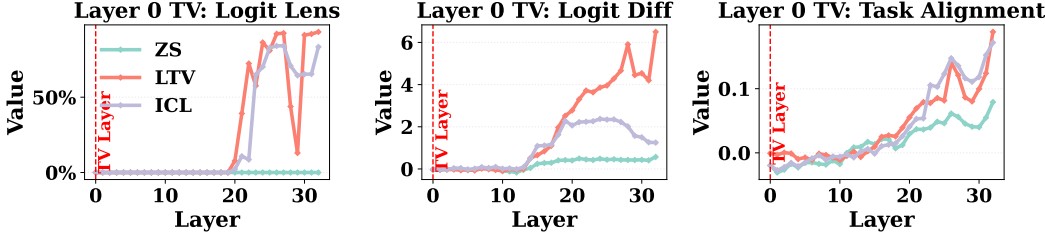

Figure 84: Metrics across layers on Llama3-8B when the TV is injected into the hidden state at layer 0.

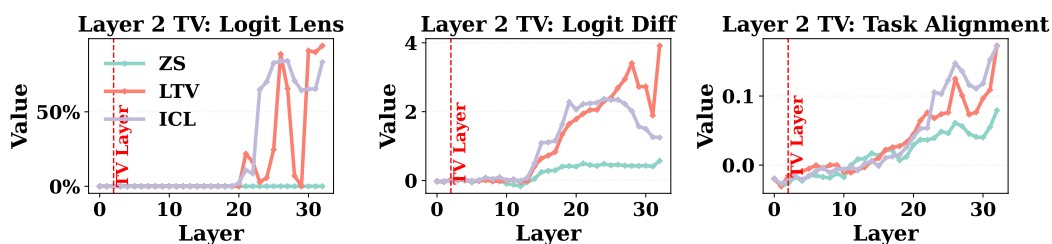

Figure 85: Metrics across layers on Llama3-8B when the TV is injected into the hidden state at layer 2.

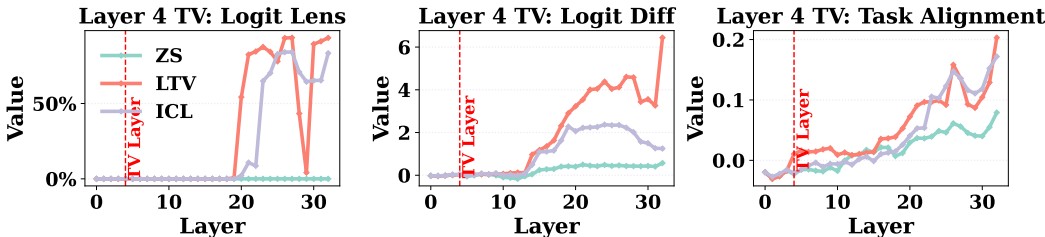

Figure 86: Metrics across layers on Llama3-8B when the TV is injected into the hidden state at layer 4.

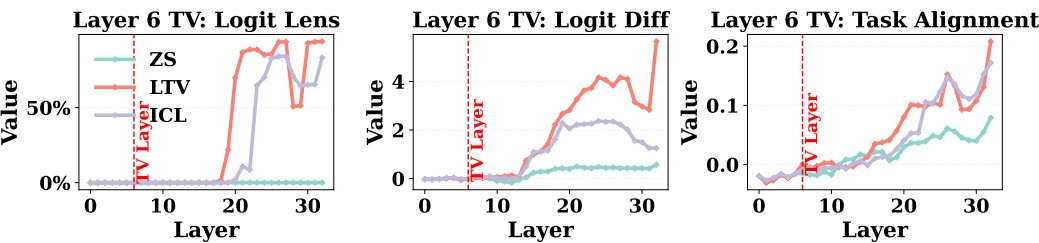

Figure 87: Metrics across layers on Llama3-8B when the TV is injected into the hidden state at layer 6.

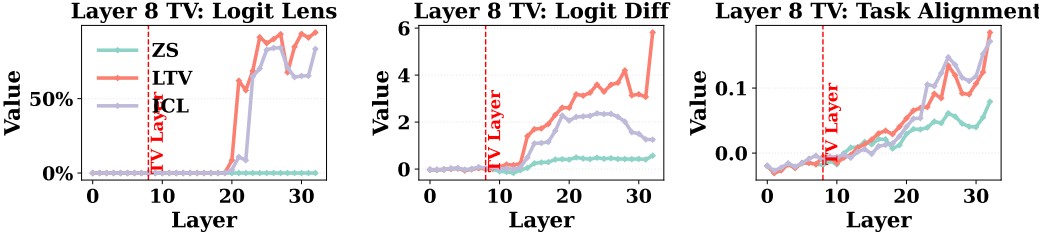

Figure 88: Metrics across layers on Llama3-8B when the TV is injected into the hidden state at layer 8.

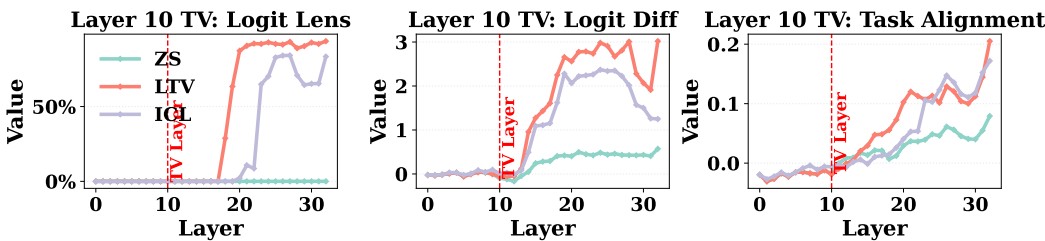

Figure 89: Metrics across layers on Llama3-8B when the TV is injected into the hidden state at layer 10.

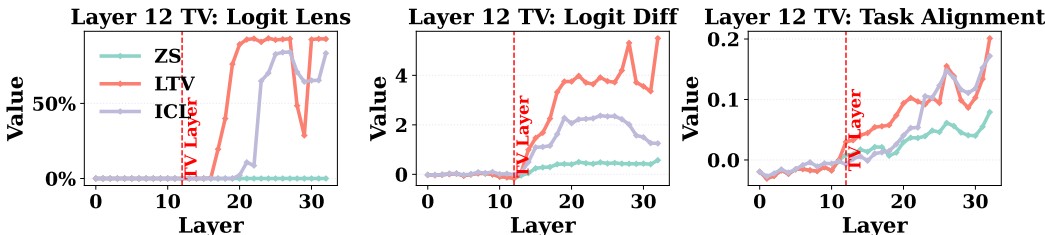

Figure 90: Metrics across layers on Llama3-8B when the TV is injected into the hidden state at layer 12.

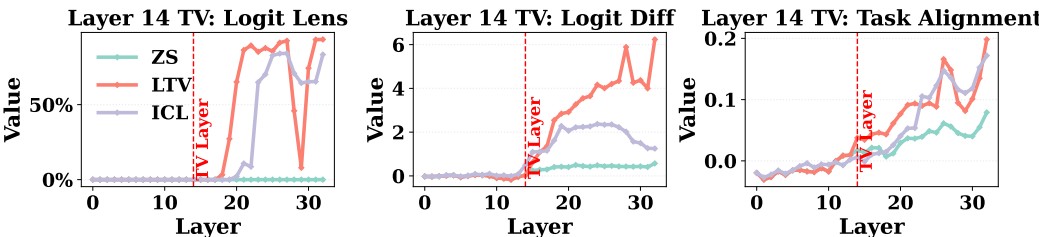

Figure 91: Metrics across layers on Llama3-8B when the TV is injected into the hidden state at layer 14.

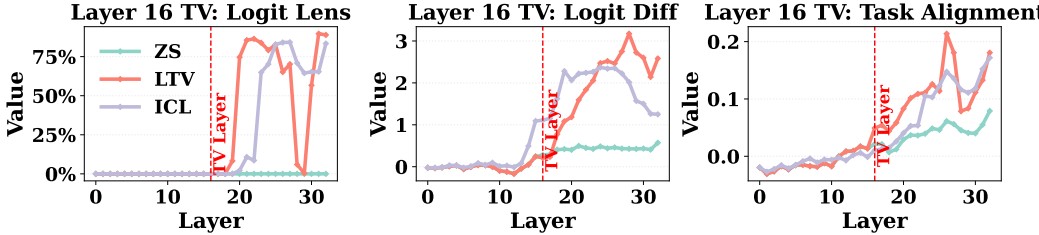

Figure 92: Metrics across layers on Llama3-8B when the TV is injected into the hidden state at layer 16.

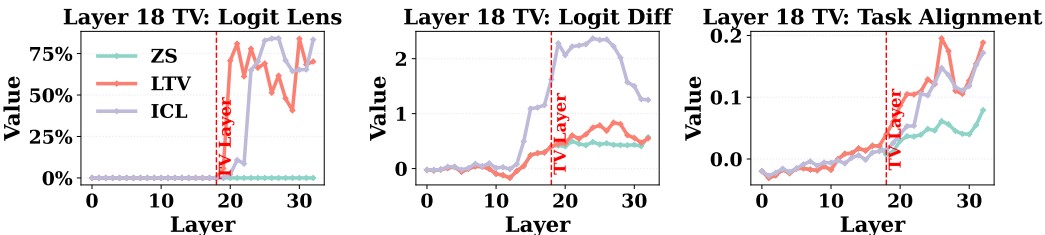

Figure 93: Metrics across layers on Llama3-8B when the TV is injected into the hidden state at layer 18.

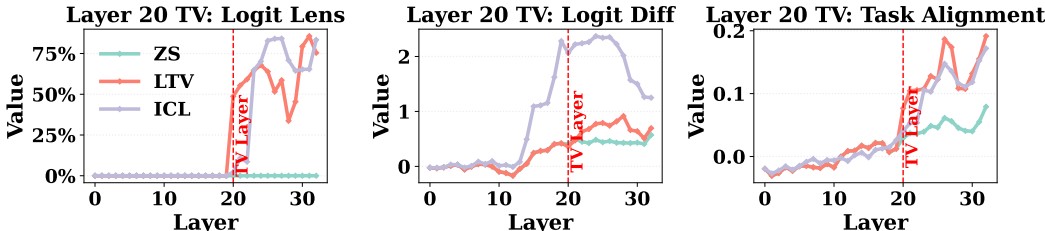

Figure 94: Metrics across layers on Llama3-8B when the TV is injected into the hidden state at layer 20.

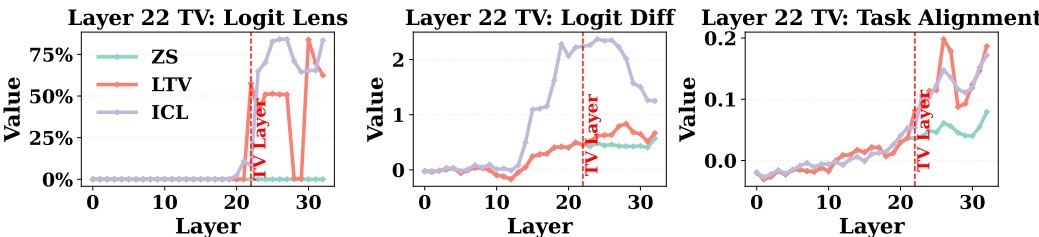

Figure 95: Metrics across layers on Llama3-8B when the TV is injected into the hidden state at layer 22.

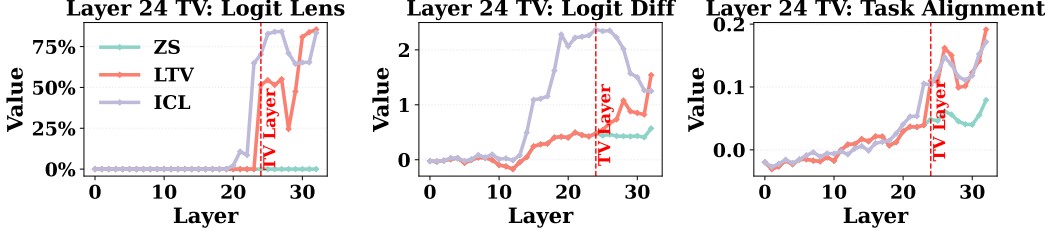

Figure 96: Metrics across layers on Llama3-8B when the TV is injected into the hidden state at layer 24.

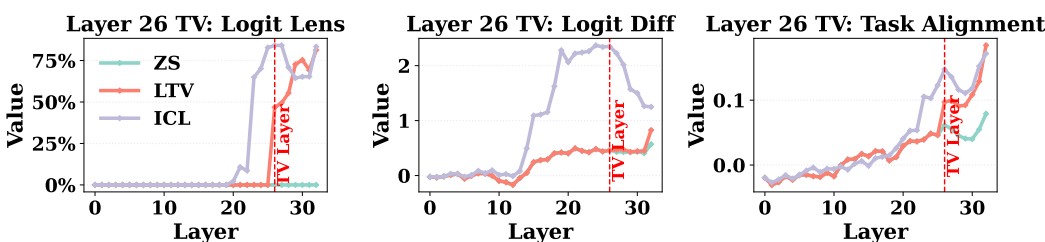

Figure 97: Metrics across layers on Llama3-8B when the TV is injected into the hidden state at layer 26.

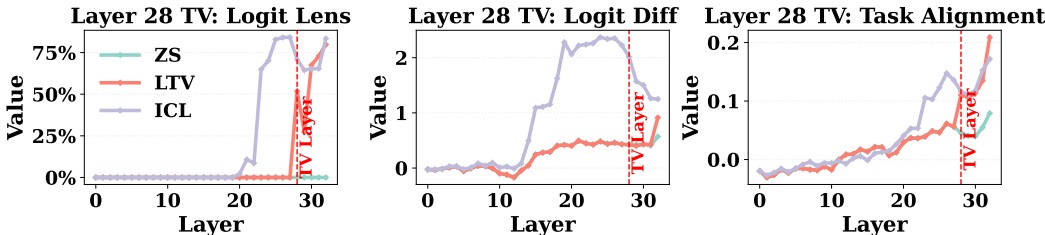

Figure 98: Metrics across layers on Llama3-8B when the TV is injected into the hidden state at layer 28.

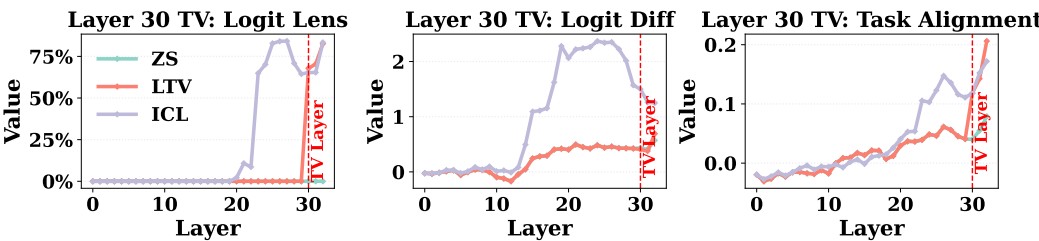

Figure 99: Metrics across layers on Llama3-8B when the TV is injected into the hidden state at layer 30.

Table 6: Comparison of LTV, State Vector, and I2CL in terms of the time (seconds) required to complete the entire training and evaluation procedures.

| | State Vector | I2CL | LTV (Ours) |
|---|---|---|---|
| | 91.31 | 66.04 | **58.07** |

| Model | ZS Accuracy | ICL Accuracy | Accuracy with LTV |
|---|---|---|---|
| Llama3-70B | 2.51% | 81.93% | 78.18% |
| Qwen2.5-32B | 12.52% | 85.44% | 75.59% |
| Yi-34B | 14.82% | 81.33% | 81.37% |

Table 7: Performance of LTVs under the traditional setting (injecting into the last-token hidden state at a single layer). Injection layers correspond to 50% of each model's total depth.

Table 8: Comparison of LTV vs. FV and Vanilla TV across five scenarios on Llama2-7B.

| Method | Baseline $\mathbb{P}=\{-1\},\mathbb{L}=\{16\}$ | 1) Diff. Pos. $\mathbb{P}=\{4\}$ | 2) More Pos. $\mathbb{P}=\{-5,\dots,-1\}$ | 3) More layers $\mathbb{L}=\{0,4,8,\dots\}$ | 4) More layers & Pos. $\mathbb{P}=\{-5,\dots\},\mathbb{L}=\{0,4,\dots\}$ | 5) ICL prompts |
|---|---|---|---|---|---|---|
| Vanilla TV | 38.26% | 1.96% | 18.85% | 14.16% | 13.30% | 52.82% |
| FV | 51.81% | 1.40% | 47.14% | 28.60% | 20.44% | 73.23% |
| LTV | 82.54%$_{\uparrow30.73\%}$ | 79.34%$_{\uparrow77.38\%}$ | 82.24%$_{\uparrow35.10\%}$ | 84.60%$_{\uparrow56.00\%}$ | 51.60%$_{\uparrow31.16\%}$ | 85.16%$_{\uparrow11.93\%}$ |

Table 9: Comparison of LTV vs. FV and Vanilla TV across five scenarios on Llama2-13B.

| Method | Baseline $\mathbb{P}=\{-1\},\mathbb{L}=\{20\}$ | 1) Diff. Pos. $\mathbb{P}=\{4\}$ | 2) More Pos. $\mathbb{P}=\{-5,\dots,-1\}$ | 3) More layers $\mathbb{L}=\{0,4,8,\dots\}$ | 4) More layers & Pos. $\mathbb{P}=\{-5,\dots\},\mathbb{L}=\{0,4,\dots\}$ | 5) ICL prompts |
|---|---|---|---|---|---|---|
| Vanilla TV | 27.67% | 1.84% | 20.46% | 16.42% | 16.07% | 43.84% |
| FV | 41.59% | 1.22% | 36.97% | 42.25% | 24.74% | 77.51% |
| LTV | 80.33%$_{\uparrow38.74\%}$ | 71.53%$_{\uparrow69.69\%}$ | 82.25%$_{\uparrow45.28\%}$ | 87.69%$_{\uparrow45.44\%}$ | 51.46%$_{\uparrow26.72\%}$ | 84.99%$_{\uparrow7.48\%}$ |

| Method | Baseline $\mathbb{P}=\{-1\},\mathbb{L}=\{16\}$ | 1) Diff. Pos. $\mathbb{P}=\{4\}$ | 2) More Pos. $\mathbb{P}=\{-5,\dots,-1\}$ | 3) More layers $\mathbb{L}=\{0,4,8,\dots\}$ | 4) More layers & Pos. $\mathbb{P}=\{-5,\dots\},\mathbb{L}=\{0,4,\dots\}$ | 5) ICL prompts |
|---|---|---|---|---|---|---|
| Vanilla TV | 31.69% | 2.02% | 26.68% | 1.05% | 0.33% | 75.83% |
| FV | 33.28% | 2.93% | 16.95% | 18.38% | 17.72% | 53.93% |
| LTV | 76.26%$_{\uparrow42.98\%}$ | 76.22%$_{\uparrow73.29\%}$ | 83.48%$_{\uparrow56.80\%}$ | 77.93%$_{\uparrow59.55\%}$ | 44.82%$_{\uparrow27.10\%}$ | 84.51%$_{\uparrow8.68\%}$ |

Table 10: Comparison of LTV vs. FV and Vanilla TV across five scenarios on Llama3-8B.

| | Baseline $\mathbb{P}=\{-1\},\mathbb{L}=\{14\}$ | 1) Diff. Pos. $\mathbb{P}=\{4\}$ | 2) More Pos. $\mathbb{P}=\{-5,\dots,-1\}$ | 3) More layers $\mathbb{L}=\{0,4,8,\dots\}$ | 4) More layers & Pos. $\mathbb{P}=\{-5,\dots\},\mathbb{L}=\{0,4,\dots\}$ | 5) ICL prompts |
|---|---|---|---|---|---|---|
| Vanilla TV | 42.61% | 3.07% | 37.05% | 18.73% | 11.33% | 65.38% |
| FV | 19.54% | 3.53% | 4.69% | 15.07% | 13.26% | 62.12% |
| LTV | 78.65%$_{\uparrow36.04\%}$ | 74.10%$_{\uparrow70.57\%}$ | 80.43%$_{\uparrow43.38\%}$ | 78.18%$_{\uparrow59.45\%}$ | 46.38%$_{\uparrow33.12\%}$ | 82.80%$_{\uparrow17.42\%}$ |

Table 11: Comparison of LTV vs. FV and Vanilla TV across five scenarios on Llama3.2-3B.

| Method | Baseline $\mathbb{P}=\{-1\},\mathbb{L}=\{40\}$ | 1) Diff. Pos. $\mathbb{P}=\{4\}$ | 2) More Pos. $\mathbb{P}=\{-5,\dots,-1\}$ | 3) More layers $\mathbb{L}=\{0,4,8,\dots\}$ | 4) More layers & Pos. $\mathbb{P}=\{-5,\dots\},\mathbb{L}=\{0,4,\dots\}$ | 5) ICL prompts |
|---|---|---|---|---|---|---|
| LTV | 78.18% | 75.34% | 75.59% | 76.13% | 48.75% | 88.40% |

Table 12: Performance of LTV across settings on Llama3-70B.

| Method | Baseline $\mathbb{P}=\{-1\},\mathbb{L}=\{32\}$ | 1) Diff. Pos. $\mathbb{P}=\{4\}$ | 2) More Pos. $\mathbb{P}=\{-5,\dots,-1\}$ | 3) More layers $\mathbb{L}=\{0,4,8,\dots\}$ | 4) More layers & Pos. $\mathbb{P}=\{-5,\dots\},\mathbb{L}=\{0,4,\dots\}$ | 5) ICL prompts |
|---|---|---|---|---|---|---|
| LTV | 75.59% | 36.04% | 87.24% | 75.20% | 53.30% | 87.08% |

Table 13: Performance of LTV across settings on Qwen2.5-32B.

| Method | Baseline $\mathbb{P}=\{-1\}, \mathbb{L}=\{30\}$ | 1) Diff. Pos. $\mathbb{P}=\{4\}$ | 2) More Pos. $\mathbb{P}=\{-5,\dots,-1\}$ | 3) More layers $\mathbb{L}=\{0,4,8,\dots\}$ | 4) More layers & Pos. $\mathbb{P}=\{-5,\dots\}, \mathbb{L}=\{0,4,\dots\}$ | 5) ICL prompts |
|---|---|---|---|---|---|---|
| LTV | 81.37% | 73.53% | 82.47% | 84.39% | 51.29% | 89.69% |

Table 14: Performance of LTV across settings on Yi-34B.

Table 15: Performance of LTV while injecting to multiple layers and positions simultaneously with different layer strides

| | $\mathbb{P}=\{-1\}$ | $\mathbb{P}=\{-5,\dots\}$ |
|---|---|---|
| **Layer Stride = 2** $\mathbb{L}=\{0,2,4,\dots\}$ | 82.40% | 51.08% |
| **Layer Stride = 4** $\mathbb{L}=\{0,4,8,\dots\}$ | 86.43% | 51.39% |
| **Layer Stride = 8** $\mathbb{L}=\{0,8,16,\dots\}$ | 88.50% | 50.47% |

Table 16: Applying the Capital LTV to other tasks. The LTV yields no substantial accuracy improvements in any case because the Capital dataset does not share its label space with any of the other datasets.

| SST-2 | TREC | RTE | SNLI | Capitalize | Antonym |
|---|---|---|---|---|---|
| 0.00% | 0.00% | 9.39% | 5.06% | 2.33% | 0.00% |

Table 17: Top-10 tokens decoded from early- and late-layer TVs on Llama3-8B.

| Layer | Decoded Tokens |
|---|---|
| Early Layer (8) | tring, CCA, erk, bart, uge, ensor, , テル, аза, emer |
| Late Layer (24) | positive, negative, positive, Positive, Negative, negative, Negative, Positive, _positive, -negative |

Table 18: Top-10 tokens decoded from early- and late-layer TVs on Llama3.2-3B.

| Layer | Decoded Tokens |
|---|---|
| Early Layer (7) | ync, flip, stress, hope, haven, Lor, negative, ugi, stressed, hab |
| Late Layer (21) | positive, positive, negative, -positive, Positive, Positive, negative, -positives, negative, Negative |

Table 19: Top-10 tokens decoded from early- and late-layer TVs on Llama3-70B.

| Layer | Decoded Tokens |
|---|---|
| Early Layer (20) | EventData, esteem, Я, 仈, spath, hores, raya, idth, , _priv |
| Late Layer (60) | negative, negative, Negative, positive, Negative, -negative, positive, Positive, Positive, _negative |

Table 20: Top-10 tokens decoded from early- and late-layer TVs on Llama2-7B.

| Layer | Decoded Tokens |
|---|---|
| Early Layer (8) | bah, arith, arna, revers, feder, HOST, BIT, Pat, orr, IP |
| Late Layer (24) | positive, negative, negative, posit, pos, Pos, neg, Pos, Neg, poz |

Table 21: Top-10 tokens decoded from early- and late-layer TVs on Llama2-13B.

| Layer | Decoded Tokens |
|---|---|
| Early Layer (8) | negative, bin, ed, agg, electric, myself, eda, hed, isser, positive |
| Late Layer (24) | negative, negative, positive, Neg, neg, neg, отри, pos, Pos, negro |

Table 22: Top-10 tokens decoded from early- and late-layer TVs on Qwen2.5-32B.

| Layer | Decoded Tokens |
|---|---|
| Early Layer (16) | fd, Reverse, inverted, Trait, ocale, Hack, ic, Traits, Aware, 逆转 |
| Late Layer (48) | .constraint, registrations, 魏, 传奇, 看点, (SE, ApplicationContext, Offensive, 产量, 浓缩 |

Table 23: Top-10 tokens decoded from early- and late-layer TVs on Yi-34B.

| Layer | Decoded Tokens |
|---|---|
| Early Layer (15) | 一分, iency, , shit, oc, , orating, 正能量, Gap, unbiased |
| Late Layer (45) | Mpc, elf, izza, Parish, 炳, 莫, nexper, 流行的, 增长率, rst |

