# OpenReview forum: "Task Vectors, Learned Not Extracted: Performance Gains and Mechanistic Insights"
_ICLR.cc/2026/Conference — ICLR 2026 Poster_

### Official Review · Reviewer_7qZh · 2025-10-27

**Soundness:** 2
**Presentation:** 3
**Contribution:** 3
**Rating:** 6
**Confidence:** 3

**Summary:**

This paper introduces learned task vectors (LTVs), a method of directly training task vectors via gradient descent, rather than existing approaches of extracting the vectors from model activations. Additionally, the paper provides a comprehensive mechanistic interpretation of how TVs affect the model's computation. The investigation shows that, at a low level, TVs influence model computation mainly through attention-head OV circuits, and at a high level, the TVs are linearly propagated through the layers of the network.

**Strengths:**

- The idea of training TVs instead of extracting them from model activations is interesting and intuitive. The formulation is easy to understand and effective
- There are extensive experiments offering insights into the mechanistic understanding of TVs.
- I like the part about separating the interpretation of TVs into low-level and high-level, with different mechanisms.

**Weaknesses:**

A major limitation in my opinion, is that the proposed LTV is more like a PEFT method, rather than an ICL method, since extra parameters need to be finetuned on a training dataset. While it is interesting to find a "middle ground" between them, I hope to see an experimental comparison with other PEFT methods to adequately demonstrate the effectiveness of the proposed method.

**Questions:**

- According to Eq.3, there seems to be no need to involve ICL demonstrations for training LTVs. Is it true that LTVs are "demonstration-agnostic"?
- What is the biggest advantage of LTV compared to existing PEFT methods, say LoRA? Is it because of the placce that the extra parameters are injected? Is there an analysis about that?

---

> ### Author Response · Authors · 2025-11-23
>
> **Dear Reviewer 7qZh,**
>
> Thank you for taking the time to provide such an insightful evaluation of our paper. We greatly appreciate the care you put into your comments and the constructive guidance you offered, which will strengthen our study. Below, we provide point-by-point responses to your remarks.
>
> **W1 & Q2: Comparison with PEFT baselines regarding performance, efficiency, and mechanistic insights**
>
> We fully agree that benchmarking PEFT baselines is important for positioning LTV as a credible PEFT alternative. As detailed in Revision Note 1, we compare LTV with Prefix Tuning and LoRA on SST-2, reporting both accuracy and efficiency metrics (training/inference latency, FLOPs per sample, and peak memory usage during training and inference). For fair parameter matching, we apply prefix tuning to all KV heads at layer 1 of Llama3.1-8B with prefix length 2, yielding d parameters, and LoRA with r=1 to the first-layer output projection, yielding the minimal 2d parameters. Against these baselines, LTV injection into the layer-0 last-token hidden states (d parameters) achieves higher final accuracy, better training-time efficiency, and lower inference latency than both LoRA variants, and is only slightly worse in inference FLOPs and memory usage. Overall, these results show that LTV is both effective and efficient as a PEFT method.
>
> Beyond performance and efficiency, a crucial advantage of LTV over standard PEFT baselines is that we provide a detailed mechanistic account of how it works, making LTV a “middle ground” between a PEFT strategy and a tool for mechanistic interpretability. Indeed, roughly two thirds of our experimental section is devoted to analyzing LTV’s mechanism, spanning both its low-level interactions with model components and its high-level effects on layer-wise transformations. This transparency is a major practical benefit over more opaque PEFT approaches, especially in settings where understanding *why* a method works is important for reducing trial-and-error and mitigating potential risks.
>
> **Q1: Is LTV demonstration-agnostic?**
>
> Yes, you are correct. While we also consider training LTV using ICL prompts (Column 5 of Table 2), which would make LTV demonstration-dependent, the baseline LTV is optimized using zero-shot prompts. This is a key advantage over traditional TV methods, since LTV’s effectiveness is not influenced by demonstration quality. In contrast, traditional TV methods rely on extraction from ICL hidden states, and their performance can be unstable due to the well-documented sensitivity of ICL to demonstration choice and quality [1,2].
>
> Despite being demonstration-agnostic, we find (Figure 8) that injecting LTV induces layer-wise geometric evolution patterns in hidden states that closely resemble those produced by ICL. This suggests that LTV captures core aspects of ICL dynamics and can promote similar hidden-state transformations, further strengthening the mechanistic value of our findings.
>
> Thank you again for your careful review and the time you invested in it. We appreciate the ongoing dialogue. If our clarifications have adequately addressed your points, we would be grateful if you could take this into account when updating your assessment.
>
> **References**
>
> [1] Wang, Huazheng, et al. "General Quality Evaluation Metric Towards Demonstrations for In-Context Learning."
>
> [2] Liu, Jiachang, et al. "What makes good in-context examples for GPT-3?."

---

> ### Author Response · Authors · 2025-11-27
>
> **Dear Reviewer 7qZh,**
>
> Thank you for your generous time and insightful evaluation of our paper. With the discussion period nearing its end, we trust that our responses have satisfactorily addressed your inquiries. If you have any remaining questions, we are more than happy to elaborate. We truly value your constructive feedback.
> Have a wonderful day!

---

### Official Review · Reviewer_ALnq · 2025-11-01

**Soundness:** 3
**Presentation:** 3
**Contribution:** 4
**Rating:** 4
**Confidence:** 3

**Summary:**

This paper proposes learning a task vector that is injected into a model’s hidden states to induce task behavior. Instead of extracting a vector from demonstrations, the vector is optimized directly on labeled training examples and can be placed at chosen layers and positions, with or without demonstrations. Empirically, the learned vector outperforms extraction-based baselines and often matches few-shot prompting while touching only a tiny number of parameters. The analysis traces how the vector acts through attention head output paths and shows that its effect propagates in a largely linear way, behaving like rotation toward a task subspace early in the network and magnitude scaling in later layers.

**Strengths:**

- The method is simple and effective. Optimize a single hidden-state vector per task and inject it where needed. This yields strong accuracy with minimal trainable parameters and integrates cleanly with or without demonstrations.

- The mechanistic account is a real contribution. The work shows that task vectors influence predictions primarily via attention-head OV circuits, reconstructs their effect by aggregating OV transforms across layers, and validates the role of a small set of key heads through ablations.

- The high-level propagation story is clear and useful. Layer-wise diagnostics with logit lens, logit difference, and task alignment reveal that early injections mainly rotate representations toward label directions, while later injections mostly amplify task-aligned components. This gives actionable intuition for where to inject.

- Results are consistent across multiple datasets and models, and the method remains flexible across layers and positions. The narrative that a single learned vector can rival few-shot prompting is well supported by the reported experiments.

**Weaknesses:**

- Writing clarity needs a pass. There are mismatches between figure and table information and the main text. Aligning the Figure 2 caption with the plotted trend and clarifying the layer sets in Table 1 would improve trust and readability.

- Please quantify efficiency and position it with matched baselines. It would help to report FLOPs, latency, and peak memory for training and inference, and include matched comparisons to common PEFT methods like LoRA, prefix tuning, or adapters at similar parameter budgets. This would make the practical efficiency gap clear.

- The OV-only reconstruction under-recovers the trained vector’s effect. The gap is visible but not analyzed component-wise, leaving open which pathways beyond OV account for the missing performance.

- The interpretation in section 4.2–4.3 is interesting, but it would become a more complete account with explicit theoretical grounding for when the linear approximation holds.

**Questions:**

- Please state the exact protocol for Figure 2. is it a single-layer injection sweep where the vector is added at one layer at a time, or a simultaneous multi-layer injection?
- In Table 1 item four the layer set differs from the main text L={0,4,.. } in main vs L={0,2,..} in table. Which schedule was actually used, and how sensitive are results to the stride of the layer set.
- Can you report training and inference FLOPs, latency, and peak memory for LTV and matched PEFT baselines like LoRA or prefix tuning at equal parameter budgets?
- What accounts for the gap between the trained vector and the OV reconstruction (Figure 5)?
- Could you give a theoretical account of when the linear propagation approximation should hold?
If several of these weaknesses are resolved, I would consider increasing my rating.

---

> ### Author Response · Authors · 2025-11-23
>
> **Dear Reviewer ALnq,**
>
> We sincerely appreciate your detailed and perceptive comments on our work. Your thorough review and helpful recommendations will greatly enhance the manuscript. Below, we address the concerns and questions you highlighted.
>
> **W1 & Q1 & Q2: Fixing Table 1, Figure 2 description and Ablation for Layer Stride**
>
> Thank you for identifying clarity issues in our paper. Please refer to Revision Note 4 and 5, where we revised Table 1 and Figure 2 per your comment: the Table 1 header now correctly indicates the layer stride of 4 (the previous stride of 2 was a typo) used in our multi-layer injection experiments, and the Figure 2 caption notes that we loop over all layers while injecting at only one layer per run. To assess whether stride choice affects our findings, we added ablations in Appendix D.3 comparing strides of 2, 4, and 8 under both injecting to the last token and to the last five tokens (as in the third and fourth columns of Table 2). Results show all three strides yield similar effects, supporting robustness to stride choice.
>
> **W2 & Q3: Comparison with PEFT Methods with Comparable Parameter Budgets**
>
> We fully concur that comparisons with PEFT methods help establish LTV as a viable PEFT approach. We refer you to Revision Note 1, where we evaluate LTV against Prefix Tuning and LoRA and report SST-2 performance together with efficiency metrics (training/inference latency, FLOPs per sample, and peak memory usage). To ensure comparable parameter budgets, we apply prefix tuning to all KV heads at the first layer of Llama3.1-8B with prefix length 2, leading to d (model dimension) parameters, and apply LoRA with r=1 to the first-layer output projection matrix, leading to the smallest possible 2d parameters. Comparing both to LTV injection into the layer-0 last-token hidden states (d parameters) shows that LTV outperforms LoRA and prefix tuning in final accuracy, training-time efficiency, and inference latency, while being only marginally behind in inference FLOPs and memory usage. These results demonstrate that LTV is a performant and efficient PEFT method.
>
> **W3 & Q4 Performance gap in the OV-based reconstruction and reconstrution with other components**
> We understand the concern that the gap between original LTV performance and OV-based reconstruction could suggest other significant components beyond the OV account. To examine this, we refer you to Revision Note 6, where we attempt reconstruction with the MLP component. Experiments show that MLP-based reconstruction significantly **underperforms** OV-based reconstruction. Moreover, supplementing OV reconstruction with MLP reconstruction by adding the two reconstructed TV and injecting the sum yields no gain (slightly lower accuracy than OV alone). This indicates that MLP reconstruction restores only a small fraction already captured by OV circuits, further establishing OV circuits as the dominant low-level pathway. The remaining gap is likely due to Transformer nonlinearities not captured by linear reconstruction, and/or distortion in forward computation caused by injecting reconstructed TVs.
>
>
> **W4&Q5 Theoretical Grounding for the Linear Hypothesis of TV's effect and its Reconstruction**
> We agree that theoretical support would strengthen our empirical study of TV-effect linearity and reconstruction. We refer you to Revision Note 7 and Appendix J. Let the original TV be $\boldsymbol{\theta}_ {l}$ and the linearly reconstructed one be $\hat{\boldsymbol{\theta}}_ {l}$. Our theorem states that we can estimate the gap between the output logit changes caused by $\boldsymbol{\theta}_ {l}$ and $\hat{\boldsymbol{\theta}}_ {l}$ through the real layer update process using the fitted layer update $\boldsymbol{W}_ {TV,(l)}$ matrix. Concretely, the estimated logit-gap magnitude is $\||\boldsymbol{W}_ {U}^{\mathcal{T}}\boldsymbol{W}_ {TV,(l)}(\boldsymbol{\theta}_ {l}-\hat{\boldsymbol{\theta}}_ {l})\||_ {2}$, where $\boldsymbol{W}_ {U}^{\mathcal{T}}$ is the unembedding restricted to the task label space $\mathcal{T}$ (e.g., positive/negative). When this quantity is small, the reconstructed TV is predicted to closely match the original TV’s function and attain stronger performance. Empirically, we compute the relative gap:
>
> $$
> \frac{\||\boldsymbol{W}_ {U}^{\mathcal{T}}\boldsymbol{W}_ {TV,(l)}(\boldsymbol{\theta}_ {l}-\hat{\boldsymbol{\theta}}_ {l})\||_ {2}}{\||\boldsymbol{W}_ {U}^{\mathcal{T}}\boldsymbol{W}_ {TV,(l)}\boldsymbol{\theta}_ {l}\||_ {2}}
> $$
>
> We correlate this with the layer-wise accuracy in Figure 9(a). We observe a strong negative association (small logit gap implies better accuracy), matching the theoretical prediction.
>
> Thank you once more for the time and attention you have devoted to reviewing our manuscript. We value the chance to continue this discussion. If you feel our responses have satisfactorily addressed your concerns, we would appreciate your consideration of this in your revised evaluation.

---

> ### Author Response · Authors · 2025-11-27
>
> **Dear Reviewer ALnq,**
>
> We deeply appreciate the thoughtful review and the time you have invested in our manuscript. As the author–reviewer dialogue concludes, we hope we have fully resolved your questions. Please let us know if there is anything else we can clarify. Thank you once again for your helpful comments.
> Wishing you a great day!

---

### Official Review · Reviewer_bLeS · 2025-11-01

**Soundness:** 3
**Presentation:** 3
**Contribution:** 3
**Rating:** 6
**Confidence:** 3

**Summary:**

This paper presents a method for learning task vectors (LTVs) via direct gradient-based optimization, providing an alternative to extraction-based approaches in in-context learning. The proposed LTVs demonstrate improved performance and greater flexibility across model layers, positions, and prompt formats, as shown through evaluations on multiple language models and benchmark tasks. The authors further conduct a mechanistic analysis, identifying specific attention heads as the primary components through which task vectors affect model predictions. The study offers both a practical technique and insight into the underlying mechanisms of task representation in large language models.

**Strengths:**

1. The paper offers a structured mechanistic analysis of in-context learning by distinguishing between low-level effects and high-level behaviors. This decomposition is supported by empirical results and contributes to a clearer understanding of how injected task vectors influence model predictions.

2. The proposed method is extensively evaluated across eight backbone language models and seven benchmark datasets covering classification and reasoning tasks. Ablation studies further examine the impact of injection layer, token position, and multi-vector configurations, providing a comprehensive empirical characterization of LTV’s behavior.

3. The use of a directly optimized task vector in LTVs via gradient descent simplifies the construction process compared to prior extraction-based methods. Despite its simplicity, the approach demonstrates consistent improvements over baseline techniques across injection layers.

**Weaknesses:**

1. The evaluation primarily compares LTV against baseline task-vector methods such as Vanilla TV and Function Vector. However, several recent approaches have explored improved task-vector formulations or parameter-efficient tuning methods (e.g., LoRA), which are not included in the comparison and could provide a more comprehensive baseline context.

2. Since each task is associated with a single learned vector, the approach may face limitations in generalizing across related tasks or domains. The paper provides limited analysis regarding the transferability and compositionality of learned task vectors.

**Questions:**

1. The paper proposes a gradient-based approach for learning task vectors but does not discuss its computational cost. Could the authors clarify the training cost of LTVs compared to extraction-based methods or parameter-efficient tuning approaches such as LoRA?

2. The evaluation focuses on Vanilla TV and FV, without comparisons to more recent task vector methods or PEFT baselines. Have the authors considered evaluating LTV against improved task-vector variants or methods like LoRA?

3. As one vector is learned per task, it is unclear whether LTVs can generalize beyond the training distribution. Do the authors have any analysis on the generalization of LTVs across tasks or under distribution shifts?

---

> ### Author Response · Authors · 2025-11-23
>
> **Dear Reviewer bLeS,**
>
> Thank you for your thoughtful and constructive review. We are grateful for the depth of your feedback and the valuable suggestions, which will undoubtedly improve our paper. Below, we respond to each of the points you raised.
>
> ---
>
> **W1, W2 & Q1, Q2: Comparison with PEFT Methods and Recent TV Baselines**
>
> We fully concur with your remark that comparisons with PEFT methods and more recent TV baselines are essential for demonstrating (i) LTV’s potential as a viable PEFT approach and (ii) its advantages over traditional extraction- or optimization-based TV methods. We first refer you to Revision Note 1, where we evaluate LTV against Prefix Tuning and LoRA. We report their performance on SST-2 as well as efficiency metrics, including training and inference latency and FLOPs per sample, and peak memory usage during both training and inference epochs. The results show that LTV outperforms both LoRA variants in final accuracy, training-time efficiency, and inference latency, and is only marginally behind in inference FLOPs and memory usage. These findings collectively demonstrate that LTV is a performant and efficient PEFT method.
>
> We refer you to Revision Note 2 for comparisons against recent TV baselines where we consider State Vector and I2CL. We would like to clarify that we did not include them in the main experiments because, by default, both methods rely on more complex optimization procedures and require simultaneous multi-layer injection and optimization, as reported in their original papers. These properties distinguish them from mechanistically cleaner vanilla TV/FV baselines and make them less suitable for the paper’s primary goal of mechanistic interpretation of TV effects. Nevertheless, in Appendix D.4 we compare LTV to State Vector and I2CL in both performance and efficiency. The results show that LTV outperforms both baselines across injections at all layers of Llama3-8B. Moreover, LTV requires less time to complete the full training and evaluation cycle on SST-2, highlighting the inefficiency introduced by the heavier optimization procedures used by traditional TV methods and underscoring LTV’s comparative efficiency.
>
> ---
>
> **W3 & Q3: Generalizability and Compositionality of LTV**
>
> We appreciate your suggestion that testing LTV’s generalizability and compositionality is important for understanding how it internalizes task essence and relationships between tasks. Please refer to Revision Note 3 for the corresponding experiments and findings.
>
> For generalizability, we first apply an LTV trained on SNLI to other datasets. The LTV achieves non-trivial accuracy only on RTE, which shares the same label space. This echoes our finding of an unusually high cosine similarity between SNLI and RTE LTVs in Figure 4. This experiment confirms that LTVs are tightly coupled to the task label tokens they are optimized to align with, and thus cross-task generalization critically depends on overlap in the label space. In Appendix D.7, we further apply the LTV trained on the Capital dataset (which shares labels with no other dataset) across tasks, and observe no meaningful transfer. This additional result reinforces the same conclusion.
>
> For compositionality, we train LTVs on two constituent tasks: **(1)** English–French translation (e.g., *dog → chien*) and **(2)** Masculine–Feminine transformation (e.g., *actor → actress*). We then add these two LTVs and evaluate the resulting vector on the composite English Masculine–French Feminine task (e.g., *actor → actrice*). This setup parallels classic word2vec-style composition and provides a clean testbed for LTV compositionality. The results show that the composed LTV achieves higher accuracy than 8-shot ICL on the composite task, demonstrating that LTVs can capture meaningful semantic associations between tasks.
>
> ---
>
> We thank you again for the thoughtful effort you invested in assessing our paper. We welcome further discussion, and if you find that our responses have resolved your concerns, we would be grateful if you could reflect this in your updated review.

---

> ### Author Response · Authors · 2025-11-27
>
> **Dear Reviewer bLeS,**
>
> Thank you very much for the time and insight you have dedicated to evaluating our work. With the discussion window drawing to a close, we hope our responses have comprehensively addressed your concerns. If any further clarification is required, we remain happy to provide it. Your feedback has been greatly appreciated.
> Hope you have a nice day!

---

### Author Response · Authors · 2025-11-23
**Revision Note**

Dear PCs, SACs, ACs, and Reviewers,

Thank you for your valuable suggestions. **To address the comments raised by the reviewers, we have made the following adjustments to the paper.** (Revisions are highlighted in orange.)

1. **Reviewer bLeS, ALnq, 7qZh: Comparison with PEFT methods regarding performance and efficiency (Section 4.1)**
   We agree with the reviewers’ suggestion that comparisons with PEFT methods are essential given our claim about the potential of LTV as a PEFT approach, and that such comparisons can further strengthen our results. Following this suggestion, we present comparisons between LTV and LoRA/Prefix Tuning with comparable parameter budgets in Section 4.1 (Table 1), in terms of final accuracy and training/inference efficiency metrics. The results demonstrate the advantages of our LTV method in both performance and efficiency.

2. **Reviewer bLeS: Comparing with more recent task vector methods (Appendix D.4)**
   We concur that comparisons with more recent task vector methods better demonstrate the superiority of LTV relative to extraction- and optimization-based TV approaches. In Appendix D.4, we evaluate LTV against two new baselines, State Vector and I2CL. The results show that LTV outperforms them across all model layers and requires less time to complete the training and evaluation procedure.

3. **Reviewer bLeS: Testing the generalizability and compositionality of LTV (Section 4.1, Appendix D.7)**
   We share the reviewer’s view that testing the generalizability and compositionality of LTV deepens our understanding of how it internalizes task essence and relationships between tasks. We have added the corresponding experiments in Section 4.1 (Table 3, Figure 5) and Appendix D.7. The results show that LTV trained on one task can generalize to other tasks that share the same label space, aligning with our observations of cross-dataset cosine similarities of LTVs and showing that the generalizbility of LTV depends on the task label space. We further observe that composing (adding) LTVs trained on constituent tasks (e.g., English–French, Masculine–Feminine) yields an effective LTV for the composite task (English Masculine–French Feminine). This demonstrates that LTVs satisfy word2vec-style algebraic properties and can represent semantic associations between tasks.

4. **Reviewer ALnq: Fixing Table 1 header and Figure 2 caption**
   We have fixed the mentioned table and figure per the reviewer’s suggestion to improve presentation clarity. Table 1’s header now correctly reflects the layer stride we use (4) in the multi-layer injection experiments, and the caption of Figure 2 clearly states that we loop over all layers and inject at only one layer each time.

5. **Reviewer ALnq: Ablation studies for the layer stride in multi-layer injection (Appendix D.3)**
   Following the reviewer’s suggestion, in Appendix D.3 we test how the choice of layer stride in multi-layer LTV injection impacts results, by experimenting with alternative strides of 2 and 8 alongside our original choice (4). The experiments show that these three strides yield largely similar results, increasing the robustness of our conclusions.

6. **Reviewer ALnq: Reconstructing TV through the MLP component (Appendix F.4)**
   We understand the reviewer’s concern that the performance gap between the original LTV and OV-based reconstruction may suggest unaccounted importance of other model components (i.e., MLPs) in LTV’s low-level mechanism. We therefore examine MLP-based reconstruction in Appendix F.4. The results show that MLP-based reconstruction recovers a much smaller proportion of LTV effectiveness, suggesting that the OV circuit (and thus attention heads) is indeed the main component modulating LTV’s effect.

7. **Reviewer ALnq: Theoretical grounding for the linear assumption of TV’s influence pathway and its linear reconstruction (Appendix J)**
   We concur that supplementing our empirical investigations of the linearity of TV effects and their linear reconstruction with theoretical justification would substantially strengthen the paper. We provide such a justification in Appendix J. The theoretical result implies that we can use the fitted $\mathbf{W}_{TV,(l)}$ matrix to estimate the difference in logit changes caused by the original TV and the linearly reconstructed TV. If this estimated logit difference is small, then the reconstructed TV closely approximates the original TV’s function and can achieve better performance. We also provide accompanying empirical validations confirming that this prediction holds in practice.

We sincerely appreciate the time and consideration you devoted to reviewing our manuscript. Your thoughtful feedback has substantially strengthened the paper.

Sincerely,
The Authors

---

### Author Response · Authors · 2025-12-02
**Summary of Reviewer-Author Discussions: Part 1**

**Dear PCs, SACs, ACs, and Reviewers,**

Thank you very much for your valuable contributions to our work. Given the unconventional review dynamics this year and the substantial length of our detailed rebuttal and revision, we provide below a concise summary of the key points from the initial reviews and the discussion period to assist the newly assigned AC in decision making.

# **Part 1 — Strengths and Positive Reviewer Feedback**

Across the initial reviews, all three reviewers agreed that the paper offers meaningful contributions to both the practical use and mechanistic understanding of task vectors. Two reviewers assigned positive scores (6 and 6), and the third reviewer (ALnq) noted that they would **consider raising their score if their concerns were resolved**. While no reviewer posted follow-up comments during the discussion period, **no new issues were raised**, and the revisions directly address all concerns highlighted as score-relevant, which we believe signals a positive post-rebuttal trajectory for the paper.

---

## **1. A simple, effective, and flexible method for learning task vectors**

Reviewers consistently emphasized that the proposed Learned Task Vector (LTV) method is conceptually simple, easy to implement, and surprisingly powerful. Unlike extraction-based approaches that depend on demonstration selection and hidden-state variability, LTV is directly optimized via gradient descent and requires training only a single O(d)-sized vector per task.

- Reviewer **ALnq** highlighted that LTV often matches few-shot prompting while touching only a tiny number of parameters, underscoring its practical appeal.
- Reviewer **7qZh** emphasized the intuitive formulation and noted that LTV offers a clean alternative to activation extraction.
- Reviewer **bLeS** underscored LTV’s flexibility across layers, positions, and prompt formats, demonstrating broad applicability.

Together, reviewers viewed LTV as an elegant and scalable formulation that unifies ICL-style task vectors with the efficiency and controllability of PEFT methods.

---

## **2. A substantial mechanistic contribution**

All reviewers identified the mechanistic analysis as a major strength of the paper. They found the decomposition into low-level and high-level mechanisms particularly illuminating:

- **Low-level:** LTVs exert most of their influence through attention-head OV circuits, identifying specific heads whose OV paths mediate the effect of LTV injection.
- **High-level:** The layer-wise propagation of LTV effects follows a largely linear trajectory, manifesting as rotations toward task-subspace directions in early layers and scalings of task-aligned components in later layers.

Reviewer **ALnq** described this mechanistic account as a meaningful contribution, offering insight not only into LTVs but into task-vector mechanisms more broadly. Reviewer **7qZh** praised the clear separation between low-level component pathways and high-level representational behavior, and reviewer **bLeS** noted that this decomposition clarifies how injected vectors influence final predictions.

---

## **3. Extensive, consistent experiments across models and datasets**

Reviewers unanimously viewed the empirical evaluation as broad in scope and carefully executed. They highlighted several aspects:

- Coverage across **eight language models** (ranging from 2B to 70B scale) and **seven datasets**, including reasoning, classification, and translation.
- Rich ablation settings that probe injection layers, token positions, stride schedules, and multi-vector setups.
- Diagnostic analyses — including logit lens, task alignment, logit difference curves, and attention-head ablations — that enable a fine-grained interpretation of LTV behavior.
- Consistency of trends across architectures and tasks, reinforcing the stability and generality of the findings.

Reviewer **ALnq** called the experimental narrative clear and useful, **bLeS** described the empirical study as comprehensive, and **7qZh** emphasized the depth of mechanistic insight provided by the experiments.

---

## **4. Clear potential as a PEFT method, pending comparisons**

Multiple reviewers independently pointed out that LTV, by design, operates in the regime of parameter-efficient finetuning: it optimizes only a single vector and modifies none of the pretrained weights. They agreed that LTV appears promising as a lightweight PEFT method and could offer advantages over standard approaches.

- Reviewer **bLeS** encouraged comparisons to LoRA and other improved task-vector formulations to contextualize performance.
- Reviewer **7qZh** noted that PEFT-style evaluation is needed to fairly position LTV.
- Reviewer **ALnq** emphasized that reporting FLOPs, memory usage, and latency relative to PEFT baselines would significantly strengthen the paper, and explicitly noted that addressing these concerns could justify a **score increase**.

---

> ### Author Response · Authors · 2025-12-02
> **Summary of Reviewer-Author Discussions: Part 2**
>
> # **Part 2 — How We Addressed Reviewers’ Concerns**
>
> We revised the paper extensively in response to all reviewer feedback, adding new experiments, comparisons, ablations, and theoretical analysis. Below, we summarize how each major concern was addressed.
>
> ---
>
> ## **1. Comparison with PEFT methods (bLeS, ALnq, 7qZh)**
> To address all reviewers’ requests for PEFT comparisons, we added evaluations in **Section 4.1 (Table 1)** comparing LTV to **LoRA** and **Prefix Tuning** under matched parameter budgets. We report:
>
> - final accuracy,
> - training and inference latency,
> - FLOPs per sample,
> - peak memory usage.
>
> LTV achieves **higher final accuracy**, **faster training**, and **lower inference latency** than both LoRA variants, while remaining comparable in FLOPs and memory. These results establish LTV as an efficient and competitive PEFT approach.
>
> ---
>
> ## **2. Comparisons with recent task-vector baselines (bLeS)**
> We added comparisons in **Appendix D.4** to two recent methods—**State Vector** and **I2CL**—which use more complex optimization or multi-layer extraction. LTV:
>
> - outperforms both methods across all tested injection layers,
> - requires less training/evaluation time,
> - maintains cleaner mechanistic interpretability.
>
> This strengthens the case that LTV improves upon both extraction- and optimization-based task vector methods.
>
> ---
>
> ## **3. Generalizability and compositionality of LTV (bLeS)**
> We added two new sets of experiments (Section 4.1, Appendix D.7):
>
> ### **Generalization:**
> - An LTV trained on **SNLI** transfers meaningfully only to **RTE**, which shares the same label space.
> - Capital-dataset LTV does *not* transfer to unrelated tasks.
>
> These results confirm that LTV’s generalization hinges on **shared label tokens**, consistent with cross-dataset cosine similarity in Figure 4.
>
> ### **Compositionality:**
> We train LTVs on:
> - English→French translation, and
> - Masculine→Feminine transformation.
>
> Adding the two vectors produces an effective composite LTV for English Masculine→French Feminine translation, outperforming 8-shot ICL.
> This demonstrates that LTVs exhibit **word2vec-style algebraic structure**.
>
> ---
>
> ## **4. Fixing figure/table mismatches and clarifying protocols (ALnq)**
> We corrected:
>
> - Table 1 header (layer stride),
> - Figure 2 caption (one-layer-at-a-time protocol).
>
> We also added **layer-stride ablations** (stride 2, 4, 8) in Appendix D.3, showing broadly similar results and confirming robustness.
>
> ---
>
> ## **5. Explaining the gap between LTV and OV-only reconstruction (ALnq)**
> Appendix F.4 now includes:
>
> - reconstruction using **MLP** components, which recovers far less of LTV’s effect,
> - combined OV+MLP reconstruction, which does not improve over OV alone.
>
> This demonstrates that **OV circuits are the dominant low-level pathway**, and the remaining gap arises from nonlinearities or injection-induced distortion.
>
> ---
>
> ## **6. Theoretical grounding for linear propagation and reconstruction (ALnq)**
> We added a theoretical justification in Appendix J showing that the logit difference between original and reconstructed TVs achieved in the true layer update can be estimated using the fitted layer update linear matrix. Empirical evidence shows that the estimated logit difference correlates strongly with performance, thereby validating the theory and providing principled support for the linear approximation.
>
> ---
>
> ## **7. Clarifying demonstration-agnostic training (7qZh)**
> We clarified that **baseline LTV training is zero-shot** and does *not* require demonstrations.
> LTV remains effective and mechanistically interpretable while avoiding the known instability of extraction-based TVs dependent on ICL demonstration quality.
>
> ---
>
> ## **8. Overall improvements in clarity, consistency, and organization (ALnq, 7qZh)**
> We improved:
>
> - consistency between text, tables, and figures,
> - alignment of layer indexing and protocols,
> - explanations of LTV mechanisms,
> - narrative flow across sections.
>
> ---
>
> We sincerely thank all reviewers for their thoughtful and constructive feedback. Their comments directly guided substantial additions and improvements that strengthened both the empirical and mechanistic contributions of the paper.
>
> Sincerely,
> **The Authors**

---

### Meta-Review · Area_Chair_UF3t · 2025-12-30

**Summary:**

This paper proposes a way to learn task vectors directly instead of extracting task vectors from prompts. The learned task vector outperforms extracted task vectors, and the paper also gives some insight on how task vectors mechanistically change the behavior of the models. Overall the reviewers find the new idea of learning task vectors to be interesting and simple to implement, and the mechanistic interpretation a solid contribution. The main concern from the reviewers focuses on comparisons to several existing methods and baselines, which the authors have addressed in the updated version. There were also concerns about quality of writing and theoretical justification for some of the mechanistic interpretation. These are partially addressed by the author response. Overall the paper gives an interesting new way of learning task vectors and a better understanding of how they work.

**Reviewer Concerns:**

Main concern: comparison to many previous methods and baselines, in terms of metrics such as FLOPS in addition to accuracy. This has been addressed in the updated version.
Theoretical justification for some of the mechanistic interpretation (Questions 4-5 from ALnq): some intuition explained.
Quality of writing: some improvements in the updated version.

**Reviewer Scores:**

bLeS: concerns mostly addressed. may +1 or stay
ALnq: some important concerns addressed while others are partially addressed. Original score a bit lower than others. Likely +1, may even +2.
7qZh: concerns mostly addressed. may +1 or stay

---

### Decision · Program_Chairs · 2026-01-26

Accept (Poster)